# GENERALIZATION OF SCALED DEEP RESNETS IN THE MEAN-FIELD REGIME

**Yihang Chen**
EPFL
`yihang.chen@epfl.ch`

**Fanghui Liu**
University of Warwick
`fanghui.liu@warwick.ac.uk`

**Yiping Lu**
New York University
`yplu@nyu.edu`

**Grigorios G. Chrysos**
University of Wisconsin-Madison
`chrysos@wisc.edu`

**Volkan Cevher**
EPFL
`volkan.cevher@epfl.ch`

## ABSTRACT

Despite the widespread empirical success of ResNet, the generalization properties of deep ResNet are rarely explored beyond the lazy training regime. In this work, we investigate *scaled* ResNet in the limit of infinitely deep and wide neural networks, of which the gradient flow is described by a partial differential equation in the large-neural network limit, i.e., the *mean-field* regime. To derive the generalization bounds under this setting, our analysis necessitates a shift from the conventional time-invariant Gram matrix employed in the lazy training regime to a time-variant, distribution-dependent version. To this end, we provide a global lower bound on the minimum eigenvalue of the Gram matrix under the mean-field regime. Besides, for the traceability of the dynamic of Kullback-Leibler (KL) divergence, we establish the linear convergence of the empirical error and estimate the upper bound of the KL divergence over parameters distribution. Finally, we build the uniform convergence for generalization bound via Rademacher complexity. Our results offer new insights into the generalization ability of deep ResNet beyond the lazy training regime and contribute to advancing the understanding of the fundamental properties of deep neural networks.

## 1 INTRODUCTION

Deep neural networks (DNNs) have achieved great success empirically, a notable illustration of which is ResNet (He et al., 2016), a groundbreaking network architecture with skip connections. One typical way to theoretically understand ResNet (e.g., optimization, generalization), is based on the neural tangent kernel (NTK) tool (Jacot et al., 2018). Concretely, under proper assumptions, the training dynamics of ResNet can be described by a fixed kernel function (NTK). Hence, the global convergence and generalization guarantees can be given via NTK and the benefits of residual connection can be further demonstrated by spectral properties of NTK (Hayou et al., 2019; Huang et al., 2020; Hayou et al., 2021; Tirer et al., 2022). However, the NTK analysis requires the parameters of ResNet to not move much during training (which is called *lazy training* or kernel regime (Chizat et al., 2019; Woodworth et al., 2020; Barzilai et al., 2022)). Accordingly, the NTK analysis fails to describe the true non-linearity of ResNet. Beyond the NTK analysis thus received great attention in the deep learning theory community.

One typical approach is, *mean-field* based analysis, which allows for unrestricted movement of the parameters of DNNs during training, (Wei et al., 2019; Woodworth et al., 2020; Ghorbani et al., 2020; Yang & Hu, 2021; Akiyama & Suzuki, 2022; Ba et al., 2022; Mahankali et al., 2023; Chen et al., 2022; Mei et al., 2018; Rotskoff & Vanden-Eijnden, 2018; Nguyen, 2019; Sirignano & Spiliopoulos, 2020b). The training dynamics can be formulated as an optimization problem over the weight space of probability measures by studying suitable scaling limits. For deep ResNet, taking the limit of infinite depth is naturally connected to a continuous neural ordinary differential equation (ODE) (Sonoda & Murata, 2019; Weinan, 2017; Lu et al., 2018; Haber & Ruthotto, 2017; Chen et al., 2018), which makes the optimization guarantees of deep ResNets feasible under the mean-field regime (Lu et al.,

2020; Ma et al., 2020; Ding et al., 2021; 2022; Barboni et al., 2022). The obtained optimization results indicate that infinitely deep and wide ResNets can easily fit the training data with random labels, i.e., *global convergence*. While previous works have obtained promising optimization results for deep ResNets, there is a notable absence of generalization analysis, which is essential for theoretically understanding why deep ResNet can generalize well *beyond the lazy training regime*. Accordingly, this naturally raises the following question:

*Can we build a generalization analysis of trained Deep ResNets in the mean-field setting?*

We answer this question by providing a generalization analysis framework of *scaled* deep ResNet in the mean-field regime, where the word *scaled* denotes a scaling factor on deep ResNet. To this end, we consider an infinite width and infinite depth ResNet parameterized by two measures: $\nu$ over the feature encoder and $\tau$ over the output layer, respectively. By proving the condition number of the optimization dynamics of deep ResNets parameterized by $\tau$ and $\nu$ is lower bounded, we obtain the global linear convergence guarantees, aiming to derive the Kullback–Leibler (KL) divergence of such two measures between initialization and after training. Based on the KL divergence results, we, therefore, build the uniform convergence result for generalization via Rademacher complexity under the mean-field regime, obtaining the convergence rate at $\mathcal{O}(1/\sqrt{n})$ given $n$ training data.

Our contributions are summarized as below:

- The paper provides the minimum eigenvalue estimation (lower bound) of the Gram matrix of the gradients for deep ResNet parameterized by the ResNet encoder's parameters and MLP predictor's parameters.
- The paper proves that the KL divergence of feature encoder $\tau$ and output layer $\nu$ can be bounded by a constant (depending only on network architecture parameters) during the training.
- This paper builds the connection between the Rademacher complexity result and KL divergence, and then derives the convergence rate $\mathcal{O}(1/\sqrt{n})$ for generalization.

Our theoretical analysis provides an in-depth understanding of the global convergence under minimal assumptions, sheds light on the KL divergence of network measures before and after training, and builds the generalization guarantees under the mean-field regime, matching classical results in the lazy training regime (Allen-Zhu et al., 2019; Du et al., 2019b). We expect that our analysis opens the door to generalization analysis for feature learning and look forward to which adaptive features can be learned from the data under the mean-field regime.

## 2 RELATED WORK

In this section, we briefly introduce the large width/depth ResNets in an ODE formulation, NTK analysis, and mean-field analysis for ResNets.

### 2.1 INFINITE-WIDTH, INFINITE-DEPTH RESNET, ODE

The limiting model of deep and wide ResNets can be categorized into three class, by either taking the width or depth to infinity: a) the infinite depth limit to the ODE/SDE model (Sonoda & Murata, 2019; Weinan, 2017; Lu et al., 2018; Chen et al., 2018; Haber & Ruthotto, 2017; Marion et al., 2023); b) infinite width limit, (Hayou et al., 2021; Hayou & Yang, 2023; Frei et al., 2019); c) infinite depth width ResNets, mean-field ODE framework under the infinite depth width limit (Li et al., 2021; Lu et al., 2020; Ding et al., 2021; 2022; Barboni et al., 2022)

In this work, we are particularly interested in mean-field ODE formulation. The deep ResNets modeling by mean-field ODE formulation stems from (Lu et al., 2020), in which every residual block is regarded as a particle and the target is changed to optimize over the empirical distribution of particles. Sander et al. (2022) discusses the rationale behind such equivalence between discrete dynamics and continuous ODE for ResNet under certain cases. The global convergence is built under a modified cost function (Ding et al., 2021), and further improved by removing the regularization term on the cost function (Ding et al., 2022). However, the analysis in (Ding et al., 2022) requires more technical assumptions about the limiting distribution. Barboni et al. (2022) show a local linear convergence by parameterizing the network with RKHS. However, the radius of the ball containing

parameters relies on the $N$-universality and is difficult to estimate. Our work requires minimal assumptions under a proper scaling of the network parameters and the design of architecture, and hence foresters optimization and generalization analyses. There is a concurrent works (Marion et al., 2023) that studies the implicit regularization of ResNets converging to ODEs, but the employed technique is different from ours and the generalization analysis in their work is missing.

## 2.2 NTK ANALYSIS FOR DEEP RESNET

Jacot et al. (2018) demonstrate that the training process of wide neural networks under gradient flow can be effectively described by the Neural Tangent Kernel (NTK) as the network's width (denoted as 'M') tends towards infinity under the NTK scaling (Du et al., 2019b). During the training, the NTK remains unchanged and thus the theoretical analyses of neural networks can be transformed to those of kernel methods. In this case, the optimization and generalization properties of neural networks can be well controlled by the minimum eigenvalue of NTK (Cao & Gu, 2019; Nguyen et al., 2021). Regarding ResNets, the architecture we are interested in, the NTK analysis is also valid (Tirer et al., 2022; Huang et al., 2020; Belfer et al., 2021), as well as an algorithm-dependent bound (Frei et al., 2019) in the lazy training regime. Compared with kernelized analysis of the wide neural network, we do not rely on the convergence to a fixed kernel as the width approaches infinity to perform the convergence and generalization analysis. Instead, under the mean field regime, the so-called kernel falls into a time-varying, measure-dependent version. We also remark that, for a ResNet with an infinite depth but constant width, the global convergence is given by Cont et al. (2022) beyond the lazy training regime by studying the evolution of gradient norm. To our knowledge, this is the first work that analyzes the (varying) kernel eigenvalue of infinite-width/depth ResNet beyond the NTK regime in terms of optimization and generalization.

## 2.3 MEAN-FIELD ANALYSIS

Under suitable scaling limits (Mei et al., 2018; Rotskoff & Vanden-Eijnden, 2018; Sirignano & Spiliopoulos, 2020a), as the number of neurons goes to infinity, *i.e.* $M \to \infty$, neural networks work in the *mean-field limit*. In this setting, the training dynamics of neural networks can be formulated as an optimization problem over the distribution of neurons. A notable benefit of the mean-field approach is that, after deriving a formula for the gradient flow, conventional PDE methods can be utilized to characterize convergence behavior, which enables both *nonlinear feature learning* and global convergence (Araújo et al., 2019; Fang et al., 2019; Nguyen, 2019; Du et al., 2019a; Chatterji et al., 2021; Chizat & Bach, 2018; Mei et al., 2018; Wojtowytsch, 2020; Lu et al., 2020; Sirignano & Spiliopoulos, 2021; 2020b; E et al., 2020; Jabir et al., 2021).

In the case of the two-layer neural network, Chizat & Bach (2018); Mei et al. (2018); Wojtowytsch (2020); Chen et al. (2020); Barboni et al. (2022) justify the mean-field approach and demonstrate the convergence of the gradient flow process, achieving the zero loss. For the wide shallow neural network, Chen et al. (2022) proves the linear convergence of the training loss by virtue of the Gram matrix. In the multi-layer case, Lu et al. (2020); Ding et al. (2021; 2022) translate the training process of ResNet to a gradient-flow partial differential equation (PDE) and showed that with depth and width depending algebraically on the accuracy and confidence levels, first-order optimization methods can be guaranteed to find global minimizers that fit the training data.

In terms of the generalization of a trained neural network under the mean-field regime, current results are limited to two-layer neural networks. For example, Chen et al. (2020) provides a generalized NTK framework for two-layer neural networks, which exhibits a "kernel-like" behavior. Chizat & Bach (2020) demonstrate that the limits of the gradient flow of two-layer neural networks can be characterized as a max-margin classifier in a certain non-Hilbertian space. Our work, instead, focuses on deep ResNets in the mean-field regime and derives the generalization analysis framework.

## 3 FROM DISCRETE TO CONTINUOUS RESNET

In this section, we present the problem setting of our deep ResNets for binary classification under the infinite depth and width limit, which allows for parameter evolution of ResNets. Besides, several mild assumptions are introduced for our proof.

## 3.1 PROBLEM SETTING

For an integer $L$, we use the shorthand $[L] = \{1, 2, \ldots, L\}$. Let $\mathcal{X} \subseteq \mathbb{R}^d$ be a compact metric space and $\mathcal{Y} \subseteq \mathbb{R}$. We assume that the training set $\mathcal{D}_n = \{(\boldsymbol{x}_i, y_i)\}_{i=1}^n$ is drawn from an unknown distribution $\mu$ on $\mathcal{X} \times \mathcal{Y}$, and $\mu_X$ is the marginal distribution of $\mu$ over $\mathcal{X}$. The goal of our supervised learning task is to find a hypothesis (i.e., a ResNet used in this work) $f : \mathcal{X} \to \mathcal{Y}$ such that $f(\boldsymbol{x}; \boldsymbol{\Theta})$ parameterized by $\boldsymbol{\Theta}$ is a good approximation of the label $y \in \mathcal{Y}$ corresponding to a new sample $\boldsymbol{x} \in \mathcal{X}$. In this paper, we consider a binary classification task, denoted by minimizing the expected risk, let $\ell_{0-1}(f, y) = \mathbb{1}\{yf < 0\}$,

$$\min_{\boldsymbol{\Theta}} \; L_{0-1}(\boldsymbol{\Theta}) := \mathbb{E}_{(\boldsymbol{x}, y) \sim \mu} \, \ell_{0-1}(f(\boldsymbol{x}; \boldsymbol{\Theta}), y) \,.$$

Note that the $0 - 1$ loss is non-convex and non-smooth, and thus difficult for optimization. One standard way in practice for training is using a *surrogate* loss for empirical risk minimization (ERM), normally convex and smooth (or at least continuous), e.g., the hinge loss, the cross-entropy loss. Interestingly, the squared loss, originally used for regression, can be also applied for classification with good statistical properties in terms of robustness and calibration error, as systematically discussed in Hui & Belkin (2020); Hu et al. (2022). Therefore, we employ the squared loss in ERM for training, let $\ell(f, y) = \frac{1}{2}(y - f)^2$,

$$\min_{\boldsymbol{\Theta}} \; \widehat{L}(\boldsymbol{\Theta}) := \frac{1}{n} \sum_{i=1}^n \ell(f(\boldsymbol{x}_i; \boldsymbol{\Theta}), y_i) = \mathbb{E}_{\boldsymbol{x} \sim \mathcal{D}_n} \ell(f(\boldsymbol{x}; \boldsymbol{\Theta}), y(\boldsymbol{x})) \,, \tag{1}$$

where $\mathcal{D}_n$ is the empirical measure of $\mu_X$ over $\{\boldsymbol{x}_i\}_{i=1}^n$ and note that $y$ is a function of $\boldsymbol{x}$.

We call the probability measure $\rho \in \mathcal{P}^2$ if $\rho$ has the finite second moment, and $\rho \in \mathcal{C}(\mathcal{P}^2; [0, 1])$ if $\rho^s \in \mathcal{P}^2, \forall s \in [0, 1]$. For $\rho_1, \rho_2 \in \mathcal{P}^2$, the 2-Wasserstein distance is denoted by $\mathcal{W}_2(\rho_1, \rho_2)$; and for $\rho_1, \rho_2 \in \mathcal{C}(\mathcal{P}^2; [0, 1])$, we define $\mathcal{W}_2(\rho_1, \rho_2) := \sup_{s \in [0,1]} \mathcal{W}_2(\rho_1^s, \rho_2^s)$.

## 3.2 RESNETS IN THE INFINITE DEPTH AND WIDTH LIMIT

The continuous formulation of ResNets is a recent approach that uses differential equations to model the behavior of the ResNet. This formulation has the advantage of enabling continuous analysis of the ODE, which can make the analysis of ResNets easier (Lu et al., 2020; Ding et al., 2021; 2022; Barboni et al., 2022). We firstly consider the following ResNet (He et al., 2016) of depth $L$ can be formulated as $\boldsymbol{z}_0(\boldsymbol{x}) = \boldsymbol{x} \in \mathbb{R}^d$, and

$$\boldsymbol{z}_{l+1}(\boldsymbol{x}) = \boldsymbol{z}_l(\boldsymbol{x}) + \frac{\alpha}{ML} \sum_{m=1}^M \boldsymbol{\sigma}(\boldsymbol{z}_l(\boldsymbol{x}), \boldsymbol{\theta}_{l,m}) \in \mathbb{R}^d, \quad l \in [L-1] \,,$$

$$f_{\boldsymbol{\Omega}_K, \boldsymbol{\Theta}_{L,M}}(\boldsymbol{x}) = \frac{\beta}{K} \sum_{k=1}^K h(\boldsymbol{z}_L, \boldsymbol{\omega}_k) \in \mathbb{R} \,, \tag{2}$$

where $\boldsymbol{x} \in \mathbb{R}^d$ is the input data, $\alpha, \beta \in \mathbb{R}_+$ are the scaling factors. $\boldsymbol{\Theta}_{L,M} = \{\boldsymbol{\theta}_{l,m} \in \mathbb{R}^{k_\nu}\}_{l=0,m=0}^{L-1,M}$ is the parameters of the ResNet encoder $\boldsymbol{\sigma} : \mathbb{R}^d \to \mathbb{R}^d$ ( activation functions are implicitly included into $\boldsymbol{\sigma}$), and $\boldsymbol{\Omega}_K = \{\boldsymbol{\omega}_k \in \mathbb{R}^{k_\tau}\}_{k=1}^K$ is the parameters of the predictor $h : \mathbb{R}^d \to \mathbb{R}$. We introduce a trainable MLP parametrized by $\boldsymbol{\omega}$ in the end, which is different from the fixed linear predictor in Lu et al. (2020); Ding et al. (2021; 2022). We make the assumptions on the choices of activation function $\boldsymbol{\sigma}$ and predictor $h$ later in Assumption 3.3. Different scaling of $\alpha, \beta$ leads to different training schemes. Note that setting $\alpha = \sqrt{M}, \beta = \sqrt{K}$ corresponds to the standard scaling in the NTK regime (Du et al., 2019b), while setting $\alpha = \beta = 1$ corresponds to the classical mean field analysis (Mei et al., 2018; Rotskoff & Vanden-Eijnden, 2018; Lu et al., 2020; Ding et al., 2022). We will keep $\alpha, \beta$ as a hyperparameter in our theoretical analysis and determine the choice of $\alpha, \beta$ in future discussions. Besides, the scaling $1/L$ is necessary to derive the neural ODE limit (Marion et al., 2022), which has been supported by the empirical observations from Bachlechner et al. (2021); Marion et al. (2023).

We then introduce the infinitely deep and wide ResNet which is known as the mean-field limit of deep ResNet (Lu et al., 2020; Ma et al., 2020; Ding et al., 2022).

**Infinite Depth**    To be specific, we re-parametrize the indices $l \in [L]$ in Eq. (2) with $s = \frac{l}{L} \in [0, 1]$. We view $\boldsymbol{z}$ in Eq. (2) as a function in $s$ that satisfies a coupled ODE, with $1/L$ being the stepsize. Accordingly, we write $\boldsymbol{\theta}_m(s) := \boldsymbol{\theta}_m(l/L) = \boldsymbol{\theta}_{l,m}$, and $\boldsymbol{\Theta}_M(s) = \{\boldsymbol{\theta}_m(s)\}_{m=1}^M$. The continuous limit of Eq. (2) by taking $L \to \infty$ is

$$\frac{\mathrm{d}\boldsymbol{z}(\boldsymbol{x}, s)}{\mathrm{d}s} = \frac{\alpha}{M} \sum_{m=1}^M \boldsymbol{\sigma}(\boldsymbol{z}(\boldsymbol{x}, s), \boldsymbol{\theta}_m(s)) = \alpha \int_{\mathbb{R}^{k_\nu}} \boldsymbol{\sigma}(\boldsymbol{z}(\boldsymbol{x}, s), \boldsymbol{\theta}) \mathrm{d}\nu_M(\boldsymbol{\theta}, s), \quad \boldsymbol{z}(\boldsymbol{x}, 0) = \boldsymbol{x}, \quad (3)$$

where the discrete probability $\nu_M(\boldsymbol{\theta}, s)$ is defined as $\nu_M(\boldsymbol{\theta}, s) := \frac{1}{M} \sum_{i=1}^M \delta_{\boldsymbol{\theta}_m(s)}(\boldsymbol{\theta})$. Accordingly, the empirical risk in Eq. (1) can be written as

$$\widehat{L}(\boldsymbol{\Omega}_K, \boldsymbol{\Theta}_M) := \mathbb{E}_{\boldsymbol{x} \sim \mathcal{D}_n} \ell(f_{\boldsymbol{\Omega}_K, \boldsymbol{\Theta}_M}(\boldsymbol{x}), y(\boldsymbol{x})). \quad (4)$$

**Infinite Width**    The mean-field limit is obtained by considering a ResNet of infinite width, i.e. $M \to \infty$. Denoting the limiting density of $\nu_M(\boldsymbol{\theta}, s)$ by $\nu(\boldsymbol{\theta}, s) \in \mathcal{C}(\mathcal{P}^2; [0, 1])$, Eq. (3) can be written as

$$\frac{\mathrm{d}\boldsymbol{z}(\boldsymbol{x}, s)}{\mathrm{d}s} = \alpha \cdot \int_{\mathbb{R}^{k_\nu}} \boldsymbol{\sigma}(\boldsymbol{z}(\boldsymbol{x}, s), \boldsymbol{\theta}) \mathrm{d}\nu(\boldsymbol{\theta}, s), \quad s \in [0, 1], \quad \boldsymbol{z}(\boldsymbol{x}, 0) = \boldsymbol{x}. \quad (5)$$

We denote the solution of Eq. (5) as $\boldsymbol{Z}_\nu(\boldsymbol{x}, s)$. Besides, we also take the infinite width limit in the final layer, i.e. $K \to \infty$, and then the limiting density of $\boldsymbol{\omega}$ is $\tau(\boldsymbol{\omega})$. The whole network can be written as

$$f_{\tau, \nu}(\boldsymbol{x}) := \beta \cdot \int_{\mathbb{R}^{k_\tau}} h(\boldsymbol{Z}_\nu(\boldsymbol{x}, 1), \boldsymbol{\omega}) \mathrm{d}\tau(\boldsymbol{\omega}), \quad (6)$$

and the empirical loss in Eq. (1) can be defined as:

$$\widehat{L}(\tau, \nu) := \mathbb{E}_{\boldsymbol{x} \sim \mathcal{D}_n} \ell(f_{\tau, \nu}(\boldsymbol{x}), y(\boldsymbol{x})). \quad (7)$$

### 3.2.1    PARAMETER EVOLUTION

In the discrete ResNet (2), consider minimizing the empirical loss $\widehat{L}(\boldsymbol{\Omega}_K, \boldsymbol{\Theta}_{L,M})$ with an infinitesimally small learning rate, the updating process can be characterized by the particle gradient flow, see Definition 2.2 in Chizat & Bach (2018):

$$\frac{\mathrm{d}\boldsymbol{\Omega}_K(t)}{\mathrm{d}t} = -K \nabla_{\boldsymbol{\Omega}_k} \widehat{L}(\boldsymbol{\Omega}_K(t), \boldsymbol{\Theta}_{L,M}(t)), \quad (8)$$

$$\frac{\mathrm{d}\boldsymbol{\Theta}_{L,M}(t)}{\mathrm{d}t} = -LM \nabla_{\boldsymbol{\Theta}_{L,M}} \widehat{L}(\boldsymbol{\Omega}_K(t), \boldsymbol{\Theta}_{L,M}(t)), \quad (9)$$

where $t$ is the rescaled pseudo-time, which amounts to assigning a $\frac{1}{K}$ or $\frac{1}{LM}$ mass to each particle, and is convenient to take the many-particle limit.

In the continuous ResNet (5), we use the gradient flow in the Wasserstein metric to characterize the evolution of $\tau, \nu$ (Chizat & Bach, 2018). The evolution of the final layer distribution $\tau(\boldsymbol{\omega})$ can be characterized as

$$\frac{\partial \tau}{\partial t}(\boldsymbol{\omega}, t) = \nabla_{\boldsymbol{\omega}} \cdot \left( \tau(\boldsymbol{\omega}, t) \nabla_{\boldsymbol{\omega}} \frac{\delta \widehat{L}(\tau, \nu)}{\delta \tau}(\boldsymbol{\omega}, t) \right), \quad t \geq 0, \quad (10)$$

where

$$\frac{\delta \widehat{L}(\tau, \nu)}{\delta \tau}(\boldsymbol{\omega}) = \mathbb{E}_{\boldsymbol{x} \sim \mathcal{D}_n}[\beta \cdot (f_{\tau, \nu}(\boldsymbol{x}) - y(\boldsymbol{x})) \cdot h(\boldsymbol{Z}_\nu(\boldsymbol{x}, 1), \boldsymbol{\omega})]. \quad (11)$$

In addition, the evolution of the ResNet layer $\nu(\boldsymbol{\theta}, s)$ can be characterized as

$$\frac{\partial \nu}{\partial t}(\boldsymbol{\theta}, s, t) = \nabla_{\boldsymbol{\theta}} \cdot \left( \nu(\boldsymbol{\theta}, s, t) \nabla_{\boldsymbol{\theta}} \frac{\delta \widehat{L}(\tau, \nu)}{\delta \nu}(\boldsymbol{\theta}, s, t) \right), \quad t \geq 0. \quad (12)$$

From the results in Lu et al. (2020); Ding et al. (2022; 2021), we can compute the functional derivative as follows:

$$\frac{\delta \widehat{L}(\tau, \nu)}{\delta \nu}(\boldsymbol{\theta}, s) = \mathbb{E}_{\boldsymbol{x} \sim \mathcal{D}_n}[\beta \cdot (f_{\tau, \nu}(\boldsymbol{x}) - y(\boldsymbol{x})) \cdot \boldsymbol{\omega}^\top \frac{\partial \boldsymbol{Z}_\nu(\boldsymbol{x}, 1)}{\partial \boldsymbol{Z}_\nu(\boldsymbol{x}, s)} \frac{\delta \boldsymbol{Z}_\nu(\boldsymbol{x}, s)}{\delta \nu}(\boldsymbol{\theta}, s)] \tag{13}$$

$$= \mathbb{E}_{\boldsymbol{x} \sim \mathcal{D}_n}[\beta \cdot (f_{\tau, \nu}(\boldsymbol{x}) - y(\boldsymbol{x})) \cdot \boldsymbol{p}_\nu^\top(\boldsymbol{x}, s) \cdot \alpha \cdot \boldsymbol{\sigma}(\boldsymbol{Z}_\nu(\boldsymbol{x}, s), \boldsymbol{\theta})], \tag{14}$$

where $\boldsymbol{p}_\nu \in \mathbb{R}^{k_\nu}$, parameterized by $\boldsymbol{x}, s, \nu$, is the solution to the following adjoint ODE, with initial condition dependent on $\tau$:

$$\frac{\mathrm{d}\boldsymbol{p}_\nu^\top}{\mathrm{d}s}(\boldsymbol{x}, s) = -\alpha \cdot \boldsymbol{p}_\nu^\top(\boldsymbol{x}, s) \int_{\mathbb{R}^{k_\nu}} \nabla_{\boldsymbol{z}} \boldsymbol{\sigma}(\boldsymbol{Z}_\nu(\boldsymbol{x}, s), \boldsymbol{\theta}) \mathrm{d}\nu(\boldsymbol{\theta}, s), \tag{15}$$

$$\boldsymbol{p}_\nu^\top(\boldsymbol{x}, 1) = \int_{\mathbb{R}^{k_\tau}} \nabla_{\boldsymbol{z}} h(\boldsymbol{Z}_\nu(\boldsymbol{x}, 1), \boldsymbol{\omega}) \mathrm{d}\tau(\boldsymbol{\omega}). \tag{16}$$

For the linear ODE (15), we can directly obtain the explicit formula, $\boldsymbol{p}_\nu^\top(\boldsymbol{x}, s) = \boldsymbol{p}_\nu^\top(\boldsymbol{x}, 1)\boldsymbol{q}_\nu(\boldsymbol{x}, s)$, where $\boldsymbol{q}_\nu(\boldsymbol{x}, s)$ is the exponentially scaling matrix defined in Eq. (17). The correctness of solution (17) can be verified by taking the gradient w.r.t. $s$ at both sides.

$$\boldsymbol{q}_\nu(\boldsymbol{x}, s) = \exp\left(\alpha \int_s^1 \int_{\mathbb{R}^{k_\nu}} \nabla_{\boldsymbol{z}} \boldsymbol{\sigma}(\boldsymbol{Z}_\nu(\boldsymbol{x}, s'), \boldsymbol{\theta}) \mathrm{d}\nu(\boldsymbol{\theta}, s')\right). \tag{17}$$

### 3.3 ASSUMPTIONS

In the following, we use the upper subscript for ResNet ODE layer $s \in [0, 1]$, and the lower subscript for training time $t \in [0, +\infty)$. For example, $\tau_t(\boldsymbol{\omega}) := \tau(\boldsymbol{\omega}, t)$, and $\nu_t^s(\boldsymbol{\theta}) := \nu(\boldsymbol{\theta}, s, t)$. First, we assume the boundedness and second moment of the dataset by Assumption 3.1.

**Assumption 3.1** (Assumptions on data). *We assume that for $\boldsymbol{x}_i \neq \boldsymbol{x}_j \sim \mu_X$, the following holds with probability 1,*

$$\|\boldsymbol{x}_i\|_2 = 1, |y(\boldsymbol{x}_i)| \leq 1, \langle \boldsymbol{x}_i, \boldsymbol{x}_j \rangle \leq C_{\max} < 1, \forall i, j \in [n].$$

**Remark:** The assumption, i.e., $\boldsymbol{x}_i, \boldsymbol{x}_j$ being not parallel, is attainable and standard in the analysis of neural networks (Du et al., 2019b; Zhu et al., 2022).

Second, we adopt the standard Gaussian initialization for distribution $\tau$ and $\nu$.

**Assumption 3.2** (Assumption on initialization). *The initial distribution $\tau_0, \nu_0$ is standard Gaussian:* $(\tau_0, \nu_0)(\boldsymbol{\omega}, \boldsymbol{\theta}, s) \propto \exp\left(-\frac{\|\boldsymbol{\omega}\|_2^2 + \|\boldsymbol{\theta}\|_2^2}{2}\right), \forall s \in [0, 1].$

Next, we adopt the following assumption on activation $\boldsymbol{\sigma}, h$ in terms of formulation and smoothness. The widely used activation functions, such as Sigmoid, Tanh, satisfy this assumption.

**Assumption 3.3** (Assumptions on activation $\boldsymbol{\sigma}, h$). *Let $\boldsymbol{\theta} := (\boldsymbol{u}, \boldsymbol{w}, b) \in \mathbb{R}^{k_\nu}$, where $\boldsymbol{u}, \boldsymbol{w} \in \mathbb{R}^{k_\nu}, b \in \mathbb{R}$, i.e. $k_\nu = 2d + 1$; $\boldsymbol{\omega} := (a, \boldsymbol{w}, b) \in \mathbb{R}^{k_\tau}$, where $\boldsymbol{w} \in \mathbb{R}^{k_\nu}, a, b \in \mathbb{R}$, i.e. $k_\tau = d + 2$.*

*For any $\boldsymbol{z} \in \mathbb{R}^{k_\nu}$, we assume*

$$\boldsymbol{\sigma}(\boldsymbol{z}, \boldsymbol{\theta}) = \boldsymbol{u}\sigma_0(\boldsymbol{w}^\top \boldsymbol{z} + b), \quad h(\boldsymbol{z}, \boldsymbol{\omega}) = a\sigma_0(\boldsymbol{w}^\top \boldsymbol{z} + b), \quad \sigma_0 : \mathbb{R} \to \mathbb{R}. \tag{18}$$

*In addition, we have the following assumption on $\sigma_0$. $|\sigma_0(x)| \leq C_1 \max(|x|, 1), |\sigma_0'(x)| \leq C_1, |\sigma_0''(x)| \leq C_1$, and let $\mu_i(\sigma_0)$ be the $i$-th Hermite coefficient of $\sigma_0$.*

Based on our description of the evolution of deep ResNets and standard assumptions, we are ready to present our main results on optimization and generalization in the following section.

## 4 MAIN RESULTS

In this section, we derive a quantitative estimation of the convergence rate of optimizing the ResNet. Our main results are three-fold: a) minimum eigenvalue estimation of the Gram matrix during the training dynamics which controls the training speed; b) a quantitative estimation of KL divergence between the weight destruction of trained ResNet with initialization; c) Rademacher complexity generalization guarantees for the trained ResNet.

### 4.1 GRAM MATRIX AND MINIMUM EIGENVALUE

The training dynamics is governed by the Gram matrix of the coordinate tangent vectors to the functional derivatives. In this section, we bound the minimum eigenvalue of the Gram matrix of the gradients through the whole training dynamics, which controls the convergence of gradient flow.

In the lazy training regime (Jacot et al., 2018), the Gram matrix converges pointwisely to the NTK as the width approaches infinity. Hence one only needs to bound the Gram matrix's minimum eigenvalue at initialization, and the global convergence rate under gradient descent is bounded by the minimum eigenvalue of NTK. In our setting under the mean-field regime, we also need similar Gram matrix/matrices to aid our proof on the training dynamics, but we do not rely on their convergence to the NTK when the width approaches infinity. Instead, we consider the Gram matrix of the limiting mean-field model Eq. (6).

For the ResNet parameter distribution $\nu$, we define one Gram matrix $\boldsymbol{G}_1(\tau, \nu) \in \mathbb{R}^{n \times n}$ by

$$\boldsymbol{G}_1(\tau, \nu) = \int_0^1 \boldsymbol{G}_1(\tau, \nu, s)\mathrm{d}s, \quad \boldsymbol{G}_1(\tau, \nu, s) = \mathbb{E}_{\boldsymbol{\theta} \sim \nu(\cdot, s)} \boldsymbol{J}_1(\tau, \nu, \boldsymbol{\theta}, s) \boldsymbol{J}_1(\tau, \nu, \boldsymbol{\theta}, s)^\top, \quad (19)$$

where the row vector of $\boldsymbol{J}_1$ is defined as

$$(\boldsymbol{J}_1(\tau, \nu, \boldsymbol{\theta}, s))_{i,\cdot} = \boldsymbol{p}_\nu^\top(\boldsymbol{x}_i, s) \nabla_{\boldsymbol{\theta}} \boldsymbol{\sigma}(\boldsymbol{Z}_\nu(\boldsymbol{x}_i, s), \boldsymbol{\theta}), \quad 1 \leq i \leq n,$$

where the dependence on $\tau$ on the right side of the equality is from the initial condition $\boldsymbol{p}_\nu^\top(\boldsymbol{x}, 1)$. We also define the Gram matrix for the MLP parameter distribution $\tau$, $\boldsymbol{G}_2(\tau, \nu) \in \mathbb{R}^{n \times n}$ by

$$\boldsymbol{G}_2(\tau, \nu) = \mathbb{E}_{\boldsymbol{\omega} \sim \tau(\cdot)} \boldsymbol{J}_2(\nu, \boldsymbol{\omega}) \boldsymbol{J}_2(\nu, \boldsymbol{\omega})^\top, \quad (20)$$

where the row vector of $\boldsymbol{J}_2$ is defined as

$$(\boldsymbol{J}_2(\nu, \boldsymbol{\omega}))_{i,\cdot} = \nabla_{\boldsymbol{\omega}} h(\boldsymbol{Z}_\nu(\boldsymbol{x}_i, 1), \boldsymbol{\omega}), \quad 1 \leq i \leq n.$$

We characterize the training dynamics of the neural networks by the following theorem (the proof deferred to Appendix C.1), which demonstrates the relationship between the gradient flow of the loss and those of functional derivatives.

**Theorem 4.1.** *The training dynamics of $\widehat{L}(\tau_t, \nu_t)$ can be written as:*

$$\frac{\mathrm{d}\widehat{L}(\tau_t, \nu_t)}{\mathrm{d}t} = -\int_0^1 \int_{\mathbb{R}^{k_\nu}} \left\| \nabla_{\boldsymbol{\theta}} \frac{\delta \widehat{L}(\tau_t, \nu_t)}{\delta \nu_t}(\boldsymbol{\theta}, s) \right\|_2^2 \mathrm{d}\nu_t(\boldsymbol{\theta}, s) - \int_{\mathbb{R}^{k_\nu}} \left\| \nabla_{\boldsymbol{\omega}} \frac{\delta \widehat{L}(\tau_t, \nu_t)}{\delta \tau_t}(\boldsymbol{\omega}) \right\|_2^2 \mathrm{d}\tau_t(\boldsymbol{\omega}).$$

From the definition of functional derivatives $\frac{\delta \widehat{L}(\tau_t, \nu_t)}{\delta \nu_t}(\boldsymbol{\theta}, s)$ and $\frac{\delta \widehat{L}(\tau_t, \nu_t)}{\delta \tau_t}(\boldsymbol{\omega})$, we immediately obtain Proposition 4.2, an extension of Theorem 4.1, which demonstrates that the training dynamics can be controlled by the corresponding Gram matrices.

**Proposition 4.2.** *Let $\boldsymbol{b}_t = (f_{\tau_t, \nu_t}(\boldsymbol{x}_1) - y(\boldsymbol{x}_1), \cdots, f_{\tau_t, \nu_t}(\boldsymbol{x}_n) - y(\boldsymbol{x}_n))$, using the Gram matrix defined in Eq.* (19) *and* (20)*, the training dynamics of $\widehat{L}(\tau_t, \nu_t)$ can be written as:*

$$\frac{\mathrm{d}\widehat{L}(\tau_t, \nu_t)}{\mathrm{d}t} = -\frac{\beta^2}{n^2} \boldsymbol{b}_t^\top (\alpha^2 \boldsymbol{G}_1(\tau_t, \nu_t) + \boldsymbol{G}_2(\tau_t, \nu_t)) \boldsymbol{b}_t.$$

Our analysis mainly relies on the minimum eigenvalue of the Gram matrix, which is commonly used in the analysis of overparameterized neural network (Arora et al., 2019; Chen et al., 2020). The minimum eigenvalue of the Gram matrix controls the convergence rate of the gradient descent.

We remark that the Gram matrix $\boldsymbol{G}_1(\tau_t, \nu_t)$ is always positive semi-definite for any $t \geq 0$, and $\boldsymbol{G}_1(\tau_0, \nu_0) = \boldsymbol{0}_{n \times n}$. Therefore, we only need to bound the minimum eigenvalue of $\boldsymbol{G}_2(\tau_t, \nu_t)$. First, we present such result under initialization, i.e., the lower bound of $\lambda_{\min}(\boldsymbol{G}_2(\tau_0, \nu_0))$ by the following lemma. The proof is deferred to Appendix C.2.

**Lemma 4.3.** *Under Assumption 3.1, 3.2, 3.3, there exist a constant $\Lambda := \Lambda(d)$, only depending on the dimension d, such that $\lambda_{\min}[\boldsymbol{G}(\tau_0, \nu_0)]$ is lower bounded by*

$$\lambda_0 := \lambda_{\min}(\boldsymbol{G}(\tau_0, \nu_0)) \geq \lambda_{\min}(\boldsymbol{G}_2(\tau_0, \nu_0)) \geq \Lambda(d).$$

**Remark:** Using the stability of the ODE model, we derive the KL divergence by virtue of the structure of the ResNet, build the lower bound of $\lambda_{\min}(\boldsymbol{G}_2)$, and prove the global convergence. In fact, our results, e.g., global convergence, and KL divergence can also depend on $\boldsymbol{G}_1$ by taking $\Lambda(t) := \alpha^2 \lambda_{\min}[\boldsymbol{G}_1(t)] + \lambda_{\min}(\boldsymbol{G}_2)$ in Lemma C.5. Due to $\lambda_{\min}[\boldsymbol{G}_1(t)] \geq 0$ for any $t$, we only use $\lambda_{\min}(\boldsymbol{G}_2) \geq \Lambda(d)$ in Lemma 4.3 for proof simplicity. Our model degenerates to a two-layer neural network if the residual part is removed (can be regarded as an identity mapping).

Second, for $\tau, \nu$ different from initialization $\tau_0, \nu_0$, we first prove in the finite time $t < t_{\max}$, we have the minimum eigenvalue is lower bounded $\lambda_{\min}(\boldsymbol{G}_2(\tau_t, \nu_t)) \geq \lambda_0/2$. In the next, we choose a proper scaling of $\alpha, \beta$, such that $t_{\max} = \infty$, so that we obtain a global guarantee.

The proof is deferred to Appendix C.2.

**Lemma 4.4.** *There exists $r_{\max}$, such that, for $\nu \in \mathcal{C}(\mathcal{P}^2; [0,1])$ and $\tau \in \mathcal{P}^2$ satisfying $\max\{\mathcal{W}_2(\nu, \nu_0), \mathcal{W}_2(\tau, \tau_0)\} \leq r_{\max}$, we have $\lambda_{\min}(\boldsymbol{G}_2(\tau, \nu)) \geq \frac{\lambda_0}{2}$.*

**Remark:** The radius is defined as $r_{\max} := \min\left\{\sqrt{d}, \frac{\Lambda(d)}{4nC_{\boldsymbol{G}}(d,\alpha)}\right\}$, where $\Lambda(d)$ is defined in Lemma 4.3 and $C_{\boldsymbol{G}}(d, \alpha)$ is some constant depending on $d$ and $\alpha$, used for the uniform estimation of $\boldsymbol{G}_2(\tau, \nu)$ around its initialization, refer to Lemma C.2. We detail this in Appendix C.3.

**Definition 4.5.** *Define*

$$t_{\max} := \sup\{t_0, \text{ s.t.} \forall t \in [0, t_0], \max\{\mathcal{W}_2(\nu_t, \nu_0), \mathcal{W}_2(\tau_t, \tau_0)\} \leq r_{\max}\},$$

*where $r_{\max}$ is defined in Lemma 4.4.*

## 4.2 KL DIVERGENCE BETWEEN TRAINED NETWORK AND INITIALIZATION

Based on our previous results on the minimum eigenvalue of the Gram matrix, we are ready to prove the global convergence of the empirical loss over the weight distributions $\tau$ and $\nu$ of ResNets, and well control the KL divergence of them before and after training. The proofs in this subsection are deferred to Appendix C.4.

We first present the gradient flow of the KL divergence of the parameter distribution.

**Lemma 4.6.** *The dynamics of the KL divergence $\mathrm{KL}(\tau_t \| \tau_0), \mathrm{KL}(\nu_t \| \nu_0)$ through training can be characterize by*

$$\frac{\mathrm{d}\mathrm{KL}(\tau_t \| \tau_0)}{\mathrm{d}t} := -\int_{\mathbb{R}^{k_\tau}} \left(\nabla_{\boldsymbol{\omega}} \frac{\delta \mathrm{KL}(\tau_t \| \tau_0)}{\delta \tau_t}\right) \cdot \left(\nabla_{\boldsymbol{\omega}} \frac{\delta \widehat{L}(\tau_t, \nu_t)}{\delta \tau_t}\right) \mathrm{d}\tau_t(\boldsymbol{\omega}),$$

$$\frac{\mathrm{d}\mathrm{KL}(\nu_t \| \nu_0)}{\mathrm{d}t} := -\int_{\mathbb{R}^{k_\nu} \times [0,1]} \left(\nabla_{\boldsymbol{\theta}} \frac{\delta \mathrm{KL}(\nu_t^s \| \nu_0^s)}{\delta \nu_t^s}\right) \cdot \left(\nabla_{\boldsymbol{\theta}} \frac{\delta \widehat{L}(\tau_t, \nu_t)}{\delta \nu_t}\right) \mathrm{d}\nu_t(\boldsymbol{\theta}, s).$$

Since the evolution of $\tau_t, \nu_t$ is continuous, we define the notation $t_{\max} > 0$ in the following, such that the minimum eigenvalue of $\boldsymbol{G}_2(\tau_t, \nu_t)$ can be lower bounded for $t < t_{\max}$. In our later proof, we will demonstrate that $t_{\max} = \infty$ can be achieved under proper $\alpha$ and $\beta$.

Combining Lemma 4.4 and Definition 4.5, we immediately obtain that $\lambda_{\min}(\boldsymbol{G}(\tau_t, \nu_t)) \geq \lambda_{\min}(\boldsymbol{G}_2(\tau_t, \nu_t)) \geq \lambda_0/2$ for $t < t_{\max}$.

By choosing certain $\alpha, \beta$, we could prove $t_{\max} = \infty$, which leads to the bound $\mathrm{KL}(\tau_t \| \tau_0)$, and $\mathrm{KL}(\nu_t \| \nu_0)$ uniformly for all $t > 0$. (*c.f.* Theorem 4.7).

**Theorem 4.7.** *Assume the PDE (10) has solution $\tau_t \in \mathcal{P}^2$, and the PDE (12) has solution $\nu_t \in \mathcal{C}(\mathcal{P}^2; [0,1])$. Under Assumption 3.1, 3.2, 3.3, for some constant $C_{\mathrm{KL}}(d, \alpha)$ dependent on $d, \alpha$, taking $\bar{\beta} := \frac{\beta}{n} > \frac{4\sqrt{C_{\mathrm{KL}}(d,\alpha)}}{\Lambda r_{\max}}$, the following results hold for all $t \in [0, \infty)$:*

$$\widehat{L}(\tau_t, \nu_t) \leq e^{-\frac{\beta^2 \Lambda}{2n} t} \widehat{L}(\tau_0, \nu_0), \quad \mathrm{KL}(\tau_t \| \tau_0) \leq \frac{C_{\mathrm{KL}}(d, \alpha)}{\Lambda^2 \bar{\beta}^2}, \quad \mathrm{KL}(\nu_t \| \nu_0) \leq \frac{C_{\mathrm{KL}}(d, \alpha)}{\Lambda^2 \bar{\beta}^2}.$$

We also derive a lower bound for the KL divergence. In Lemma C.10, we have that the average movement of the KL divergence is in the same order as the change in output layers.

### 4.3 RADEMACHER COMPLEXITY BOUND

After our previous estimates on the minimum eigenvalue of the Gram matrix as well as the KL divergence, we are ready to build the generalization bound for such trained mean-field ResNets. The proofs in this subsection are deferred to Appendix C.5.

Before we start the proof, we introduce some basic notations of Rademacher complexity. Let $\mathcal{D}_X = \{x_i\}_{i=1}^n$ be the training dataset, and $\eta_1, \cdots, \eta_n$ be an i.i.d. family of Rademacher variables taking values $\pm 1$ with equal probability. For any function set $\mathcal{H}$, the global Rademacher complexity is defined as $\mathcal{R}_n(\mathcal{H}) := \mathbb{E}\left[\sup_{h \in \mathcal{H}} \frac{1}{n} \sum_{i=1}^n \eta_i h(x_i)\right]$.

Let $\mathcal{F} = \left\{f_{\tau,\nu}(x) = \beta \cdot \int h(Z_\nu(x, 1), \omega) \mathrm{d}\tau(\omega)\right\}$ be the function class of infinite wide infinite depth ResNet defined in Section 3. We consider the following function class of infinite wide infinite depth ResNets whose KL divergence to the initial distribution is upper bounded by some $r > 0$: $\mathcal{F}_{\mathrm{KL}}(r) = \{f_{\tau,\nu} \in \mathcal{F} : \mathrm{KL}(\tau \| \tau_0) \leq r, \mathrm{KL}(\nu \| \nu_0) \leq r\}$. The Rademacher complexity of $\mathcal{F}_{\mathrm{KL}}(r)$ is given by the following lemma.

**Lemma 4.8.** *Under Assumption 3.3, if $r \leq r_0 = O(1/\sqrt{n})$, the Rademacher complexity of $\mathcal{F}_{\mathrm{KL}}(r)$ can be bounded by $\mathcal{R}_n(\mathcal{F}_{\mathrm{KL}}(r)) \lesssim \beta\sqrt{r/n}$, where $\lesssim$ hides the constant dependence on $d, \alpha$.*

Now we consider the generalization error of the 0-1 classification problem,

**Theorem 4.9** (Generalization). *Assume $\tau_y \in \mathcal{C}(\mathcal{P}^2; [0, 1])$ and $\nu_y \in \mathcal{P}^2$ be the ground truth distributions, such that, $y(x) = \mathbb{E}_{\omega \sim \tau_y} h(Z_{\nu_y}(x, 1), \omega)$. Under the Assumption 3.1, 3.2 and 3.3, we set $\beta > \Omega(\sqrt{n})$. For any $\delta > 0$, with probability at least $1 - \delta$, the following bound holds:*

$$\mathbb{E}_{x \sim \mu_X} \ell_{0-1}(f_{\tau_\star, \nu_\star}(x), y(x)) \lesssim 1/\sqrt{n} + 6\sqrt{\log(2/\delta)/2n},$$

*where $\lesssim$ hides the constant dependence on $d, \alpha$.*

**Remark:** Our results of $O(1/\sqrt{n})$ matches the standard generalization error in the NTK regime (Du et al., 2019b). However, in contrast to setting $\alpha = \sqrt{M}, \beta = \sqrt{K}$ in Eq. (2) as the NTK regime, we directly analyze the ResNet in the limiting infinite width depth model in Eq. (6), and select proper choice of $\alpha, \beta$ independent of the width. We also validate our theoretical results by some numerical experiments in Appendix C.6.

## 5 CONCLUSION

In this paper, we build the generalization bound for trained deep results beyond the NTK regime under mild assumptions. Our results demonstrate that the KL divergence between the distribution of parameters after training and initialization of an infinitely width and deep ResNet can be controlled via lower bounding the eigenvalue of the Gram matrix during training. Under some stronger data assumptions, e.g., $k$-sparse parity problem (Suzuki et al., 2023), we may ensure that the limiting distribution of deep ResNet moves far away from its initialization in terms of KL divergence, which cannot be derived under the current setting. We leave it as the future work.

## 6 ACKNOWLEDGEMENT

This work was carried out in the EPFL LIONS group. This work was supported by Hasler Foundation Program: Hasler Responsible AI (project number 21043), the Army Research Office and was accomplished under Grant Number W911NF-24-1-0048, and Swiss National Science Foundation (SNSF) under grant number 200021_205011. Corresponding authors: Fanghui Liu and Yihang Chen.

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

## A    OVERVIEW OF APPENDIX

We give a brief overview of the appendix here.

- **Appendix B**. In Appendix B.1, we prove some lemmas that will be useful. In Appendix B.2, we provide the estimation of the activation function $\boldsymbol{\sigma}$. In Appendix B.3, we provide the prior estimation of $\boldsymbol{Z}_\nu, \boldsymbol{p}_\nu$.

- **Appendix C** In Appendix C.1, we prove the gradient flow of $\frac{\mathrm{d}\widehat{L}}{\mathrm{d}t}$ and $\frac{\mathrm{d}\mathrm{KL}}{\mathrm{d}t}$. In Appendix C.2, we bound the minimal eigenvalue at initialization. In Appendix C.3, we bound the perturbation of minimal eigenvalue. In Appendix C.4, we bound the KL divergence in finite time, and choose scaling parameters to prove the main results in Theorem 4.7. In Appendix C.5, we bound the KL divergence and provided the generalization bound.

- **Appendix C.6** We provide the experimental verification.

## B    USEFUL ESTIMATIONS

### B.1    USEFUL LEMMAS

**Lemma B.1** (2-Wasserstein continuity for functions of quadratic growth, Proposition 1 in Polyanskiy & Wu (2016))**.** *Let $\mu, \nu$ be two probability measures on $\mathbb{R}^d$ with finite second moments, and let $g : \mathbb{R}^d \to \mathbb{R}$ be a $\mathcal{C}_1$ function obeying*

$$\|\nabla g(w)\|_2 \le c_1 \|w\|_2 + c_2, \forall w \in \mathbb{R}^d ,$$

*for some constants $c_1 > 0$ and $c_2 \ge 0$. Then*

$$|\mathbb{E}_{w \sim \mu} g(w) - \mathbb{E}_{w \sim \nu} g(w)| \le (c_1 \sigma + c_2) \mathcal{W}_2(\mu, \nu) ,$$

*where $\sigma^2 = \max\{\mathbb{E}_{w \sim \mu} \|w\|_2^2, \mathbb{E}_{w \sim \nu} \|w\|_2^2\}$*

**Lemma B.2** (Corollary 2.1 in Otto & Villani (2000))**.** *The probability measure $\nu_0(\boldsymbol{\theta}) \propto \exp(-\frac{\|\boldsymbol{\theta}\|_2^2}{2})$ satisfies following Talagrand inequality (in short $T(\frac{1}{2})$) for any $\nu \in \mathcal{P}^2(\mathbb{R}^{k_\nu})$*

$$\mathcal{W}_2^2(\nu, \nu_0) \le 4\mathrm{KL}(\nu \| \nu_0).$$

**Lemma B.3** (Donsker-Varadhan representation (Donsker & Varadhan, 1975))**.** *Let $\mu, \lambda$ be probability measures on a measurable space $(X, \Sigma)$. For any bounded, $\Sigma$-measurable functions $\Phi : X \to \mathbb{R}$:*

$$\int_X \Phi \mathrm{d}\mu \le \mathrm{KL}(\mu \| \lambda) + \log \int_X \exp(\Phi) \mathrm{d}\lambda.$$

### B.2    ESTIMATION OF $\boldsymbol{\sigma}$

**Lemma B.4** (Boundedness of $\boldsymbol{\sigma}(\boldsymbol{z}, \boldsymbol{\theta})$)**.** *Under Assumption 3.3, for $\boldsymbol{x} \in \mathbb{R}^d, \boldsymbol{\theta} \in \mathbb{R}^k$, we have*

$$\|\boldsymbol{\sigma}(\boldsymbol{z}, \boldsymbol{\theta})\|_2 \le C_{\boldsymbol{\sigma}}(\|\boldsymbol{z}\|_2 + 1)(\|\boldsymbol{\theta}\|_2^2 + 1), \tag{21}$$

$$\|\nabla_{\boldsymbol{z}} \boldsymbol{\sigma}(\boldsymbol{z}, \boldsymbol{\theta})\|_F \le C_{\boldsymbol{\sigma}}(\|\boldsymbol{\theta}\|_2^2 + 1), \tag{22}$$

$$\|\nabla_{\boldsymbol{\theta}} \boldsymbol{\sigma}(\boldsymbol{z}, \boldsymbol{\theta})\|_F \le C_{\boldsymbol{\sigma}}(\|\boldsymbol{z}\|_2 + 1)(\|\boldsymbol{\theta}\|_2 + 1), \tag{23}$$

$$\|\Delta_{\boldsymbol{\theta}} \boldsymbol{\sigma}(\boldsymbol{z}, \boldsymbol{\theta})\|_2 \le C_{\boldsymbol{\sigma}}(\|\boldsymbol{z}\|_2^2 + 1)(\|\boldsymbol{\theta}\|_2 + 1), \tag{24}$$

$$\|\nabla_{\boldsymbol{\theta}}(\nabla_{\boldsymbol{\theta}} \boldsymbol{\sigma}(\boldsymbol{z}, \boldsymbol{\theta}) \cdot \boldsymbol{\theta})\|_F \le C_{\boldsymbol{\sigma}}(\|\boldsymbol{\theta}\|_2 + 1)(\|\boldsymbol{z}\|_2 + 1), \tag{25}$$

$$\|\nabla_{\boldsymbol{\theta}} \Delta_{\boldsymbol{\theta}} \boldsymbol{\sigma}(\boldsymbol{z}, \boldsymbol{\theta})\|_F \le C_{\boldsymbol{\sigma}}(\|\boldsymbol{\theta}\|_2 + 1)(\|\boldsymbol{z}\|_2^3 + 1), \tag{26}$$

*where $\Delta$ is the Laplace operator. Let $\boldsymbol{\sigma} = (\sigma_i)_{i=1}^d$, $(\nabla_{\boldsymbol{z}} \boldsymbol{\sigma})_{ij} = \nabla_{z_j} \sigma_i$, $(\nabla_{\boldsymbol{\theta}} \boldsymbol{\sigma})_{ij} = \nabla_{\theta_j} \sigma_i$, $(\Delta_{\boldsymbol{\theta}} \boldsymbol{\sigma})_i = \Delta_{\boldsymbol{\theta}} \sigma_i$, $(\nabla_{\boldsymbol{\theta}} \boldsymbol{\sigma} \cdot \boldsymbol{\theta})_{ij} = (\nabla_{\boldsymbol{\theta}} \boldsymbol{\sigma})_{ij} \theta_j$.*

*Proof of Lemma B.4.* We prove the relations directly in the following:

$$\|\boldsymbol{\sigma}(\boldsymbol{z}, \boldsymbol{\theta})\|_2 = \|\boldsymbol{u}\sigma_0(\boldsymbol{w}^\top \boldsymbol{z} + b)\|_2 \le C_1 \|\boldsymbol{u}\|_2 |\boldsymbol{w}^\top \boldsymbol{z} + b| \le C_1(\|\boldsymbol{z}\|_2 + 1)(\|\boldsymbol{\theta}\|_2^2 + 1), \tag{27}$$

$$\|\nabla_{\boldsymbol{z}} \boldsymbol{\sigma}(\boldsymbol{z}, \boldsymbol{\theta})\|_F \le \|\boldsymbol{u}\|_2 |\boldsymbol{w}\sigma_0'(\boldsymbol{w}^\top \boldsymbol{z} + b)| \le \|\boldsymbol{u}\|_2 \cdot C_1 \|\boldsymbol{w}\|_2 \le C_1(\|\boldsymbol{\theta}\|_2^2 + 1). \tag{28}$$

We can write $\nabla_{\boldsymbol{\theta}}\boldsymbol{\sigma}(\boldsymbol{z},\boldsymbol{\theta}) \in \mathbb{R}^{d \times k}$ by

$$(\nabla_{\boldsymbol{\theta}}\boldsymbol{\sigma}(\boldsymbol{z},\boldsymbol{\theta}))_{ij} = \begin{cases} \sigma_0(\boldsymbol{w}^\top\boldsymbol{z}+b) & j=i, \\ 0, & j \neq i, 1 \leq j \leq d, \\ u_i z_{j-d}\sigma_0'(\boldsymbol{w}^\top\boldsymbol{z}+b) & d+1 \leq j \leq 2d, \\ u_i\sigma_0'(\boldsymbol{w}^\top\boldsymbol{z}+b) & j=2d+1. \end{cases} \tag{29}$$

Therefore,

$$\begin{aligned} \|\nabla_{\boldsymbol{\theta}}\boldsymbol{\sigma}(\boldsymbol{z},\boldsymbol{\theta})\|_F^2 &= \sum_{i=1}^d \left[\sigma_0^2(\boldsymbol{w}^\top\boldsymbol{z}+b) + u_i^2(\sigma_0'(\boldsymbol{w}^\top\boldsymbol{z}+b))^2\left(1+\|\boldsymbol{z}\|_2^2\right)\right] \\ &\leq dC_1^2(\|\boldsymbol{w}\|_2\|\boldsymbol{z}\|_2+b)^2 + C_1^2\|u\|_2^2(1+\|\boldsymbol{z}\|_2^2) \\ &\leq 2dC_1^2(\|\boldsymbol{w}\|_2^2\|\boldsymbol{z}\|_2^2+b^2) + C_1^2\|u\|_2^2(1+\|\boldsymbol{z}\|_2^2) \\ &\leq 2dC_1^2(1+\|\boldsymbol{z}\|_2^2)(\|\boldsymbol{w}\|_2^2 + \|\boldsymbol{u}\|_2^2 + b^2 + 1) \\ &= 2dC_1^2(1+\|\boldsymbol{z}\|_2^2)(1+\|\boldsymbol{\theta}\|_2^2) \leq 2dC_1^2(1+\|\boldsymbol{z}\|)^2(1+\|\boldsymbol{\theta}\|_2)^2. \end{aligned}$$

Therefore,

$$\|\nabla_{\boldsymbol{\theta}}\boldsymbol{\sigma}(\boldsymbol{z},\boldsymbol{\theta})\|_F \leq \sqrt{2d}C_1(1+\|\boldsymbol{z}\|)(1+\|\boldsymbol{\theta}\|_2). \tag{30}$$

For $i \in [d]$

$$|\Delta_{\boldsymbol{\theta}}\boldsymbol{\sigma}(\boldsymbol{z},\boldsymbol{\theta})_i| = \left|u_i(1+\|\boldsymbol{z}\|_2^2)\sigma_0''(\boldsymbol{w}^\top\boldsymbol{z}+b)\right| \leq C_1 \cdot u_i(\|\boldsymbol{z}\|_2^2+1). \tag{31}$$

Therefore,

$$\|\Delta_{\boldsymbol{\theta}}\boldsymbol{\sigma}(\boldsymbol{z},\boldsymbol{\theta})\|_2 \leq C_1(\|\boldsymbol{z}\|_2^2+1) \cdot \sqrt{\sum_{i=1}^d u_i^2} \leq C_1(\|\boldsymbol{z}\|_2^2+1) \cdot (\|\theta\|_2+1). \tag{32}$$

By Eq. (29), we obtain

$$\langle\nabla_{\boldsymbol{\theta}}\boldsymbol{\sigma}(\boldsymbol{z},\boldsymbol{\theta})_{i,\cdot},\boldsymbol{\theta}\rangle = u_i\sigma_0(\boldsymbol{w}^\top\boldsymbol{z}+b) + u_i(\boldsymbol{w}^\top\boldsymbol{z}+b)\sigma_0'(\boldsymbol{w}^\top\boldsymbol{z}+b), \quad 1 \leq i \leq d.$$

Hence,

$$(\nabla_{\boldsymbol{\theta}}\langle\nabla_{\boldsymbol{\theta}}\boldsymbol{\sigma}(\boldsymbol{z},\boldsymbol{\theta})_{i,\cdot},\boldsymbol{\theta}\rangle)_j = \begin{cases} \sigma_0(\boldsymbol{w}^\top\boldsymbol{z}+b) + (\boldsymbol{w}^\top\boldsymbol{z}+b)\sigma_0'(\boldsymbol{w}^\top\boldsymbol{z}+b) & j=i, \\ 0 & j \neq i, 1 \leq j \leq d, \\ u_i z_{j-d}\left(\sigma_0'(y) + (y\sigma_0'(y))'\right)|_{y=\boldsymbol{w}^\top\boldsymbol{z}+b} & d+1 \leq j \leq 2d, \\ u_i\left(\sigma_0'(y) + (y\sigma_0'(y))'\right)|_{y=\boldsymbol{w}^\top\boldsymbol{z}+b} & j=2d+1. \end{cases}$$

By Assumption 3.3, we have

$$\begin{aligned} \|\nabla_{\boldsymbol{\theta}}\langle\nabla_{\boldsymbol{\theta}}\boldsymbol{\sigma}(\boldsymbol{z},\boldsymbol{\theta})_{i,\cdot},\boldsymbol{\theta}\rangle\|_2^2 &= (\sigma_0(\boldsymbol{w}^\top\boldsymbol{z}+b) + (\boldsymbol{w}^\top\boldsymbol{z}+b)\sigma_0'(\boldsymbol{w}^\top\boldsymbol{z}+b))^2 \\ &\quad + (u_i\left(\sigma_0'(y) + (y\sigma_0'(y))'\right)|_{y=\boldsymbol{w}^\top\boldsymbol{z}+b})^2(1+\|\boldsymbol{z}\|_2^2) \\ &\leq [2C_1(\|\boldsymbol{\theta}\|_2+1)(\|\boldsymbol{z}\|_2+1)]^2 + 4u_i^2C_1^2(1+\|\boldsymbol{z}\|_2^2) \end{aligned}$$

Hence,

$$\begin{aligned} \|\nabla_{\boldsymbol{\theta}}\langle\nabla_{\boldsymbol{\theta}}\boldsymbol{\sigma}(\boldsymbol{z},\boldsymbol{\theta}),\boldsymbol{\theta}\rangle\|_F^2 &= \sum_{i=1}^d \|\nabla_{\boldsymbol{\theta}}\langle\nabla_{\boldsymbol{\theta}}\boldsymbol{\sigma}(\boldsymbol{z},\boldsymbol{\theta})_{i,\cdot},\boldsymbol{\theta}\rangle\|_2^2 \\ &\leq 4dC_1^2(\|\boldsymbol{\theta}\|_2+1)^2(\|\boldsymbol{z}\|_2+1)^2 + 4\|u\|_2^2C_1^2(\|\boldsymbol{z}\|_2+1)^2 \\ &\leq (4d+4)C_1^2(\|\boldsymbol{\theta}\|_2+1)^2(\|\boldsymbol{z}\|_2+1)^2 \end{aligned} \tag{33}$$

For the last part, by Eq. (31),

$$\nabla_{\boldsymbol{\theta}}\Delta_{\boldsymbol{\theta}}\boldsymbol{\sigma}(\boldsymbol{z},\boldsymbol{\theta})_{ij} = \begin{cases} (1+\|\boldsymbol{z}\|_2^2)\sigma_0''(\boldsymbol{w}^\top\boldsymbol{z}+b) & j=i \\ 0 & j\neq i, 1\leq j\leq d \\ u_iz_{j-d}(1+\|\boldsymbol{z}\|_2^2)\boldsymbol{\sigma}'''(\boldsymbol{w}^\top\boldsymbol{z}+b) & d+1\leq j\leq 2d \\ u_i(1+\|\boldsymbol{z}\|_2^2)\boldsymbol{\sigma}'''(\boldsymbol{w}^\top\boldsymbol{z}+b) & j=2d+1 \end{cases}$$

Therefore,

$$\|\nabla_{\boldsymbol{\theta}}\Delta_{\boldsymbol{\theta}}\boldsymbol{\sigma}(\boldsymbol{z},\boldsymbol{\theta})\|_F \leq C_1(1+\|\boldsymbol{z}\|_2^2)\sqrt{\sum_{i=1}^d 1 + u_i^2(1+\|\boldsymbol{z}\|_2^2)}$$

$$\leq \sqrt{d}C_1(\|\boldsymbol{z}\|_2^2+1)^{1.5}(\|\boldsymbol{\theta}\|_2+1) \leq 3\sqrt{d}C_1(\|\boldsymbol{\theta}\|_2+1)(\|\boldsymbol{z}\|_2^3+1) \quad (34)$$

The last inequality is from that, for $x > 0$

$$x^3 + 1 = x^3 + \frac{1}{2} + \frac{1}{2} \geq \frac{3}{2^{\frac{2}{3}}}x, \quad x^3 + 1 = 1 + \frac{x^3}{2} + \frac{x^3}{2} \geq \frac{3}{2^{\frac{2}{3}}}x^2,$$

then, we have

$$(1+x^2)^{\frac{3}{2}} \leq (1+x^2)(1+x) = 1 + x + x^2 + x^3 \leq (1+x^3)(1+\frac{2^{\frac{5}{3}}}{3}) < 3(1+x^3).$$

From Eq. (27), Eq. (28), Eq. (30), Eq. (32), Eq. (33), and Eq. (34), taking $C_{\boldsymbol{\sigma}}^1 = 4\sqrt{d}C_1$, the proof is finished. We defer the definition of $C_{\boldsymbol{\sigma}}$ later. $\qquad\square$

**Lemma B.5** (Stability of $\boldsymbol{\sigma}(\boldsymbol{z},\boldsymbol{\theta})$). *Under Assumption 3.3, for $\boldsymbol{x}\in\mathbb{R}^d, \boldsymbol{\theta}\in\mathbb{R}^k$, we have*

$$\|\boldsymbol{\sigma}(\boldsymbol{z}_1,\boldsymbol{\theta}) - \boldsymbol{\sigma}(\boldsymbol{z}_2,\boldsymbol{\theta})\|_2 \leq C_{\boldsymbol{\sigma}}\cdot(\|\boldsymbol{\theta}\|_2^2+1)\|\boldsymbol{z}_1-\boldsymbol{z}_2\|_2 \quad (35)$$

$$\|\nabla_{\boldsymbol{z}}\boldsymbol{\sigma}(\boldsymbol{z}_1,\boldsymbol{\theta}) - \nabla_{\boldsymbol{z}}\boldsymbol{\sigma}(\boldsymbol{z}_2,\boldsymbol{\theta})\|_F \leq C_{\boldsymbol{\sigma}}\cdot(\|\boldsymbol{\theta}\|_2^2+1)\|\boldsymbol{z}_1-\boldsymbol{z}_2\|_2 \quad (36)$$

$$\|\nabla_{\boldsymbol{z}}\boldsymbol{\sigma}(\boldsymbol{z},\boldsymbol{\theta}_1) - \nabla_{\boldsymbol{z}}\boldsymbol{\sigma}(\boldsymbol{z},\boldsymbol{\theta}_2)\|_F \leq C_{\boldsymbol{\sigma}}\cdot(\|\boldsymbol{\theta}_1\|_2+\|\boldsymbol{\theta}_2\|_2+1)\|\boldsymbol{\theta}_1-\boldsymbol{\theta}_2\|_2 \quad (37)$$

$$\|\nabla_{\boldsymbol{\theta}}\boldsymbol{\sigma}(\boldsymbol{z},\boldsymbol{\theta}_1) - \nabla_{\boldsymbol{\theta}}\boldsymbol{\sigma}(\boldsymbol{z},\boldsymbol{\theta}_2)\|_F \leq C_{\boldsymbol{\sigma}}\cdot(\|\boldsymbol{z}\|_2+1)\|\boldsymbol{\theta}_1-\boldsymbol{\theta}_2\|_2 \quad (38)$$

$$\|\nabla_{\boldsymbol{\theta}}\boldsymbol{\sigma}(\boldsymbol{z}_1,\boldsymbol{\theta}) - \nabla_{\boldsymbol{\theta}}\boldsymbol{\sigma}(\boldsymbol{z}_2,\boldsymbol{\theta})\|_F \leq C_{\boldsymbol{\sigma}}\cdot(\|\boldsymbol{\theta}\|_2^2+1)\|\boldsymbol{z}_1-\boldsymbol{z}_2\|_2 \quad (39)$$

*Proof of Lemma B.5.* By the mean-value theorem, we have there exists $\epsilon\in[0,1]$

$$\|\boldsymbol{\sigma}(\boldsymbol{z}_1,\boldsymbol{\theta}) - \boldsymbol{\sigma}(\boldsymbol{z}_2,\boldsymbol{\theta})\|_2 \leq \|\nabla_{\boldsymbol{z}}\boldsymbol{\sigma}(\boldsymbol{z}_1+\epsilon(\boldsymbol{z}_2-\boldsymbol{z}_1),\boldsymbol{\theta})\|_F\|\boldsymbol{z}_1-\boldsymbol{z}_2\|_2$$

$$\leq C_{\boldsymbol{\sigma}}^1\cdot(\|\boldsymbol{\theta}\|_2^2+1)\|\boldsymbol{z}_1-\boldsymbol{z}_2\|_2$$

Denote by $\boldsymbol{\theta} = (\boldsymbol{u},\boldsymbol{w},b)$, we have

$$\|\nabla_{\boldsymbol{z}}\sigma(\boldsymbol{z}_1,\boldsymbol{\theta}) - \nabla_{\boldsymbol{z}}\sigma(\boldsymbol{z}_2,\boldsymbol{\theta})\|_F \leq \|\boldsymbol{u}\boldsymbol{w}^\top(\sigma_0'(\boldsymbol{w}^\top\boldsymbol{z}_1+b) - \sigma_0'(\boldsymbol{w}^\top\boldsymbol{z}_2+b))\|_F$$

$$\leq C_{\boldsymbol{\sigma}}^1\cdot(\|\boldsymbol{\theta}\|_2^2+1)\|\boldsymbol{z}_1-\boldsymbol{z}_2\|_2$$

and

$$\|\nabla_{\boldsymbol{z}}\boldsymbol{\sigma}(\boldsymbol{z},\boldsymbol{\theta}_1) - \nabla_{\boldsymbol{z}}\boldsymbol{\sigma}(\boldsymbol{z},\boldsymbol{\theta}_2)\|_F \leq \|\boldsymbol{u}_1\boldsymbol{w}_1^\top\sigma_0'(\boldsymbol{w}_1^\top\boldsymbol{z}+b_1) - \boldsymbol{u}_2\boldsymbol{w}_2^\top\sigma_0'(\boldsymbol{w}_2^\top\boldsymbol{z}+b_2)\|_F$$

$$\leq C_{\boldsymbol{\sigma}}^1\|\boldsymbol{u}_1\boldsymbol{w}_1^\top - \boldsymbol{u}_2\boldsymbol{w}_2^\top\|_F \leq C_{\boldsymbol{\sigma}}^1\|(\boldsymbol{u}_1-\boldsymbol{u}_2)(\boldsymbol{w}_1-\boldsymbol{w}_2)^\top + (\boldsymbol{u}_1-\boldsymbol{u}_2)\boldsymbol{w}_2^\top + \boldsymbol{u}_1(\boldsymbol{w}_1-\boldsymbol{w}_2)^\top\|_F$$

$$\leq 2C_{\boldsymbol{\sigma}}^1(\|\boldsymbol{\theta}_1\|_2+\|\boldsymbol{\theta}_2\|_2+1)\|\boldsymbol{\theta}_1-\boldsymbol{\theta}_2\|_2$$

In the next, we have

$$(\nabla_{\boldsymbol{\theta}}\boldsymbol{\sigma}(\boldsymbol{z},\boldsymbol{\theta}_1) - \nabla_{\boldsymbol{\theta}}\boldsymbol{\sigma}(\boldsymbol{z},\boldsymbol{\theta}_2))_{ij}$$

$$= \begin{cases} \sigma_0(\boldsymbol{w}_1^\top\boldsymbol{z}+b_1) - \sigma_0(\boldsymbol{w}_2^\top\boldsymbol{z}+b_2) & j=i, \\ 0, & j\neq i, 1\leq j\leq d, \\ u_i^1 z_{j-d}\sigma_0'(\boldsymbol{w}_1^\top\boldsymbol{z}+b_1) - u_i^2 z_{j-d}\sigma_0'(\boldsymbol{w}_2^\top\boldsymbol{z}+b_2) & d+1\leq j\leq 2d, \\ u_i^1\sigma_0'(\boldsymbol{w}_1^\top\boldsymbol{z}+b_1) - u_i^2\sigma_0'(\boldsymbol{w}_2^\top\boldsymbol{z}+b_2) & j=2d+1. \end{cases}$$

and then,

$$(\nabla_{\boldsymbol{\theta}}\boldsymbol{\sigma}(\boldsymbol{z},\boldsymbol{\theta}_1))_{ij} - (\nabla_{\boldsymbol{\theta}}\boldsymbol{\sigma}(\boldsymbol{z},\boldsymbol{\theta}_2))_{ij}| = \begin{cases} C_{\boldsymbol{\sigma}}^1 \cdot \|\boldsymbol{w}_1 - \boldsymbol{w}_2\|_2 \|\boldsymbol{z}\|_2 & j = i, \\ 0, & j \neq i, 1 \leq j \leq d, \\ C_{\boldsymbol{\sigma}}^1 \cdot |u_i^1 - u_i^2| z_{j-d} & d+1 \leq j \leq 2d, \\ C_{\boldsymbol{\sigma}}^1 \cdot |u_i^1 - u_i^2| & j = 2d+1. \end{cases}$$

Therefore,

$$\|\nabla_{\boldsymbol{\theta}}\boldsymbol{\sigma}(\boldsymbol{z},\boldsymbol{\theta}_1) - \nabla_{\boldsymbol{\theta}}\boldsymbol{\sigma}(\boldsymbol{z},\boldsymbol{\theta}_2)\|_F \leq \sqrt{2d}C_{\boldsymbol{\sigma}}^1(\|\boldsymbol{z}\|_2 + 1) \cdot \|\boldsymbol{\theta}_1 - \boldsymbol{\theta}_2\|_2.$$

Similarly, we have

$$(\nabla_{\boldsymbol{\theta}}\boldsymbol{\sigma}(\boldsymbol{z}_1,\boldsymbol{\theta}))_{ij} - (\nabla_{\boldsymbol{\theta}}\boldsymbol{\sigma}(\boldsymbol{z}_2,\boldsymbol{\theta}))_{ij}$$
$$= \begin{cases} \sigma_0(\boldsymbol{w}^\top \boldsymbol{z}_1 + b) - \sigma_0(\boldsymbol{w}^\top \boldsymbol{z}_2 + b) & j = i, \\ 0, & j \neq i, 1 \leq j \leq d, \\ u_i z_{j-d}^1 \sigma_0'(\boldsymbol{w}^\top \boldsymbol{z}_1 + b_1) - u_i z_{j-d}^2 \sigma_0'(\boldsymbol{w}^\top \boldsymbol{z}_2 + b) & d+1 \leq j \leq 2d, \\ u_i \sigma_0'(\boldsymbol{w}^\top \boldsymbol{z}_1 + b) - u_i \sigma_0'(\boldsymbol{w}^\top \boldsymbol{z}_2 + b) & j = 2d+1, \end{cases}$$

and then

$$(\nabla_{\boldsymbol{\theta}}\boldsymbol{\sigma}(\boldsymbol{z},\boldsymbol{\theta}_1))_{ij} - (\nabla_{\boldsymbol{\theta}}\boldsymbol{\sigma}(\boldsymbol{z},\boldsymbol{\theta}_2))_{ij}| = \begin{cases} C_{\boldsymbol{\sigma}}^1 \cdot \|\boldsymbol{w}\|_2 \|\boldsymbol{z}_1 - \boldsymbol{z}_2\|_2 & j = i, \\ 0, & j \neq i, 1 \leq j \leq d, \\ C_{\boldsymbol{\sigma}}^1 \cdot u_i |z_{j-d}^1 - z_{j-d}^2| & d+1 \leq j \leq 2d, \\ C_{\boldsymbol{\sigma}}^1 \cdot u_i \|\boldsymbol{z}_1 - \boldsymbol{z}_2\|_2 \|\boldsymbol{w}\|_2 & j = 2d+1. \end{cases}$$

Therefore,

$$\|\nabla_{\boldsymbol{\theta}}\boldsymbol{\sigma}(\boldsymbol{z}_1,\boldsymbol{\theta}) - \nabla_{\boldsymbol{\theta}}\boldsymbol{\sigma}(\boldsymbol{z}_2,\boldsymbol{\theta})\|_F \leq \sqrt{d}C_{\boldsymbol{\sigma}}^1(\|\boldsymbol{\theta}\|_2^2 + 1)\|\boldsymbol{z}_1 - \boldsymbol{z}_2\|_2$$

taking $C_{\boldsymbol{\sigma}}^2 = \sqrt{2d}C_{\boldsymbol{\sigma}}^1$, the proof is finished. $\qquad\square$

Combined with the estimation in the proofs of Lemma B.4 and Lemma B.5, we let

$$C_{\boldsymbol{\sigma}} := 6d \cdot C_1 \tag{40}$$

### B.3 PRIOR ESTIMATION OF ODE

The Lemma B.6 and Lemma B.7 establishes the boundedness and stability of $\boldsymbol{Z}_\nu$ and $\boldsymbol{p}_\nu$ with respect to $\nu$.

**Lemma B.6** (Boundedness and Stability of $\boldsymbol{Z}_\nu$). *Suppose that Assumption 3.3 holds and that $\boldsymbol{x}$ is in the support of $\mathcal{X}$. Suppose that $\nu_1, \nu_2 \in \mathcal{C}(\mathcal{P}^2; [0,1])$ and $\boldsymbol{Z}_{\nu_1}, \boldsymbol{Z}_{\nu_2}$ are the corresponding unique solutions in Eq. (5).Then the following two bounds are satisfied for all $s \in [0,1]$:*

$$\|\boldsymbol{Z}_{\nu_1}(\boldsymbol{x},s)\|_2 \leq C_{\boldsymbol{Z}}(\|\nu_1\|_\infty^2; \alpha),$$

*and*

$$\|\boldsymbol{Z}_{\nu_1}(\boldsymbol{x},s) - \boldsymbol{Z}_{\nu_2}(\boldsymbol{x},s)\|_2 \leq C_{\boldsymbol{Z}}(\|\nu_1\|_\infty^2, \|\nu_2\|_\infty^2; \alpha) \cdot \mathcal{W}_2(\nu_1, \nu_2),$$

*where $C_{\boldsymbol{Z}}(\|\nu_1\|_\infty^2, \|\nu_2\|_\infty^2; \alpha)$ is a constant depending only on $\|\nu_1\|_\infty^2, \|\nu_2\|_\infty^2$ and $\alpha$, and for $\nu \in \mathcal{C}(\mathcal{P}^2; [0,1])$, we denote by $\|\nu\|_\infty^2 := \sup_{s \in [0,1]} \mathbb{E}_{\boldsymbol{\theta} \sim \nu(\cdot,s)} \|\boldsymbol{\theta}\|_2^2 < \infty$.*

*Proof of Lemma B.6.* We firstly demonstrate that Eq. (5) has a unique $\mathcal{C}_1$ solution and then prove the boundedness of $\boldsymbol{Z}_\nu$ under different probability measures.

By Lemma B.5, we have

$$\left\| \int_{\mathbb{R}^k} (\boldsymbol{\sigma}(\boldsymbol{z}_1,\boldsymbol{\theta}) - \boldsymbol{\sigma}(\boldsymbol{z}_2,\boldsymbol{\theta}))\mathrm{d}\nu_1(\boldsymbol{\theta},s) \right\|_2 \leq C_{\boldsymbol{\sigma}}\|\boldsymbol{z}_1 - \boldsymbol{z}_2\|_2 \int_{\mathbb{R}^k} (\|\boldsymbol{\theta}\|_2^2 + 1)\mathrm{d}\nu_1(\boldsymbol{\theta},s)$$
$$\leq C_{\boldsymbol{\sigma}}\|\boldsymbol{z}_1 - \boldsymbol{z}_2\|_2(\|\nu_1\|_\infty^2 + 1), \tag{41}$$

which implies that $\int_{\mathbb{R}^k} \boldsymbol{\sigma}(\boldsymbol{z}_1, \boldsymbol{\theta}) \mathrm{d}\nu_1(\boldsymbol{\theta}, s)$ is locally Lipschitz. Combining this with the a-priori estimate, the ODE theory implies that Eq. (5) has a unique $\mathcal{C}_1$ solution.

In the next, we aim to prove the boundedness of $\boldsymbol{Z}_\nu$. For any $s \in [0, 1]$, by Eq. (21) in Lemma B.4, we have

$$\left\| \int_{\mathbb{R}^k} \boldsymbol{\sigma}(\boldsymbol{z}, \boldsymbol{\theta}) \mathrm{d}\nu_1(\boldsymbol{\theta}, s) \right\|_2 \leq \int_{\mathbb{R}^k} \|\boldsymbol{\sigma}(\boldsymbol{z}, \boldsymbol{\theta})\|_2 \mathrm{d}\nu_1(\boldsymbol{\theta}, s) \leq C_{\boldsymbol{\sigma}}(\|\boldsymbol{z}\|_2 + 1) \int_{\mathbb{R}^k} (\|\boldsymbol{\theta}\|_2^2 + 1) \mathrm{d}\nu_1(\boldsymbol{\theta}, s) .$$

To prove the boundedness of $\boldsymbol{Z}_{\nu_1}$, using Eq. (5) and Lemma B.4, we have

$$\frac{\mathrm{d}\|\boldsymbol{Z}_{\nu_1}(\boldsymbol{x}, s)\|_2^2}{\mathrm{d}s} = 2\boldsymbol{Z}_{\nu_1}^\top(\boldsymbol{x}, s) \frac{\mathrm{d}\boldsymbol{Z}_{\nu_1}(\boldsymbol{x}, s)}{\mathrm{d}s}$$

$$\leq 2\alpha C_{\boldsymbol{\sigma}}(\|\boldsymbol{Z}_{\nu_1}(\boldsymbol{x}, s)\|_2^2 + \|\boldsymbol{Z}_{\nu_1}(\boldsymbol{x}, s)\|_2) \int_{\mathbb{R}^k} (\|\boldsymbol{\theta}\|_2^2 + 1) \mathrm{d}\nu_1(\boldsymbol{\theta}, s)$$

$$\leq 4\alpha C_{\boldsymbol{\sigma}}(\|\boldsymbol{Z}_{\nu_1}(\boldsymbol{x}, s)\|_2^2 + 1) \int_{\mathbb{R}^k} (\|\boldsymbol{\theta}\|_2^2 + 1) \mathrm{d}\nu_1(\boldsymbol{\theta}, s).$$

By Grönwall's inequality, and $\boldsymbol{Z}_{\nu_1}(\boldsymbol{x}, 0) = \boldsymbol{x}$, we have

$$\|\boldsymbol{Z}_{\nu_1}(\boldsymbol{x}, s)\|_2 \leq \exp\left( 2\alpha C_{\boldsymbol{\sigma}} \left( \int_0^1 \int_{\mathbb{R}^k} \|\boldsymbol{\theta}\|_2^2 \mathrm{d}\nu_1(\boldsymbol{\theta}, s) + 1 \right) \right) (\|\boldsymbol{x}\|_2 + 1)$$

$$\leq \exp(2\alpha C_{\boldsymbol{\sigma}}(\|\nu_1\|_\infty^2 + 1))(\|\boldsymbol{x}\|_2 + 1).$$

By Assumption 3.1, $\|\boldsymbol{x}\|_2 \leq 1$, we thus have a priori estimate of $Z_{\nu_1}$. Let $C_{\boldsymbol{Z}}^1(\|\nu_1\|_\infty^2; \alpha) := 2\exp(2\alpha C_{\boldsymbol{\sigma}}(\|\nu_1\|_\infty^2 + 1))$, we have $\|\boldsymbol{Z}_{\nu_1}(\boldsymbol{x}, s)\|_2 \leq C_{\boldsymbol{Z}}^1(\|\nu_1\|_\infty^2; \alpha)$.

In the next, to estimate the difference under different measures $\nu_1$ and $\nu_2$, define

$$\boldsymbol{\delta}(\boldsymbol{x}, s) = \boldsymbol{Z}_{\nu_1}(\boldsymbol{x}, s) - \boldsymbol{Z}_{\nu_2}(\boldsymbol{x}, s) ,$$

and we can easily obtain

$$\frac{\mathrm{d}\|\boldsymbol{\delta}(\boldsymbol{x}, s)\|_2^2}{\mathrm{d}s} = 2\alpha \left\langle \boldsymbol{\delta}(\boldsymbol{x}, s), \int_{\mathbb{R}^k} \boldsymbol{\sigma}(\boldsymbol{Z}_{\nu_1}(\boldsymbol{x}, s), \boldsymbol{\theta}) \mathrm{d}\nu_1(\boldsymbol{\theta}, s) - \int_{\mathbb{R}^k} \boldsymbol{\sigma}(\boldsymbol{Z}_{\nu_2}(\boldsymbol{x}, s), \boldsymbol{\theta}) \mathrm{d}\nu_2(\boldsymbol{\theta}, s) \right\rangle$$

$$:= 2\alpha \left\langle \boldsymbol{\delta}(\boldsymbol{x}, s), (\mathtt{A}) + (\mathtt{B}) \right\rangle , \tag{42}$$

where, by Eq. (41), we have

$$\|(\mathtt{A})\|_2 := \left\| \int_{\mathbb{R}^k} (\boldsymbol{\sigma}(\boldsymbol{Z}_{\nu_1}(\boldsymbol{x}, s), \boldsymbol{\theta}) - \boldsymbol{\sigma}(\boldsymbol{Z}_{\nu_2}(\boldsymbol{x}, s), \boldsymbol{\theta})) \mathrm{d}\nu_1(\boldsymbol{\theta}, s) \right\|_2 \leq C_{\boldsymbol{\sigma}} \|\boldsymbol{\delta}(\boldsymbol{x}, s)\|_2 (\|\nu_1\|_\infty^2 + 1) ,$$

and

$$(\mathtt{B}) := \int_{\mathbb{R}^k} \boldsymbol{\sigma}(\boldsymbol{Z}_{\nu_2}(\boldsymbol{x}, s), \boldsymbol{\theta}) \mathrm{d}\nu_1(\boldsymbol{\theta}, s) - \int_{\mathbb{R}^k} \boldsymbol{\sigma}(\boldsymbol{Z}_{\nu_2}(\boldsymbol{x}, s), \boldsymbol{\theta}) \mathrm{d}\nu_2(\boldsymbol{\theta}, s) .$$

By Lemma B.1 and $\|\nabla_{\boldsymbol{\theta}} \boldsymbol{\sigma}(\boldsymbol{Z}_{\nu_2}(\boldsymbol{x}, s), \boldsymbol{\theta})\|_F \leq C_{\boldsymbol{\sigma}} \cdot (\|\boldsymbol{Z}_{\nu_2}(\boldsymbol{x}, s), \boldsymbol{\theta})\|_2 + 1)(\|\boldsymbol{\theta}\|_2 + 1)$, we can bound $(\mathtt{B})$ by

$$\|(\mathtt{B})\|_2 \leq C_{\boldsymbol{\sigma}} \cdot (\|\boldsymbol{Z}_{\nu_2}(\boldsymbol{x}, s), \boldsymbol{\theta})\|_2 + 1) \cdot (\|\boldsymbol{\theta}\|_2 + 1) \cdot \mathcal{W}_2(\nu_1^s, \nu_2^s)$$

$$\leq C_{\boldsymbol{\sigma}} \cdot (\|\boldsymbol{Z}_{\nu_2}(\boldsymbol{x}, s), \boldsymbol{\theta})\|_2 + 1) \cdot (\sqrt{\max\{\|\nu_1^s\|_2^2, \|\nu_2^s\|_2^2\}} + 1) \cdot \mathcal{W}_2(\nu_1^s, \nu_2^s)$$

$$\leq C_{\boldsymbol{\sigma}} \cdot (C_{\boldsymbol{Z}}^1(\|\nu_2\|_\infty^2; \alpha) + 1) \cdot (\sqrt{\|\nu_1\|_\infty^2 + \|\nu_2\|_\infty^2} + 1) \mathcal{W}_2(\nu_1, \nu_2).$$

Plugging the estimate of $(\mathtt{A})$ and $(\mathtt{B})$ into Eq. (42), we have

$$\frac{\mathrm{d}\|\boldsymbol{\delta}(\boldsymbol{x}, s)\|_2^2}{\mathrm{d}s} \leq 2\alpha C_{\boldsymbol{\sigma}} \big( \|\boldsymbol{\delta}(\boldsymbol{x}, s)\|_2^2 (\|\nu_1\|_\infty^2 + 1)$$

$$+ \|\boldsymbol{\delta}(\boldsymbol{x}, s)\|_2 (C_{\boldsymbol{Z}}^1(\|\nu_1\|_\infty^2; \alpha) + 1)(\sqrt{\|\nu_1\|_\infty^2 + \|\nu_2\|_\infty^2} + 1) \cdot \mathcal{W}_2(\nu_1, \nu_2))$$

$$\leq 2\alpha C_{\boldsymbol{\sigma}}(\|\boldsymbol{\delta}(\boldsymbol{x}, s)\|_2^2 + \mathcal{W}_2^2(\nu_1, \nu_2))(\sqrt{\|\nu_1\|_\infty^2 + \|\nu_2\|_\infty^2} + 1)^2 (C_{\boldsymbol{Z}}^1(\|\nu_1\|_\infty^2; \alpha) + 1)^2 .$$

Since $\boldsymbol{\delta}(\boldsymbol{x}, 0) = 0$, by Grönwall's inequality, we have, $\forall s \in [0, 1]$,

$$\|\boldsymbol{\delta}(\boldsymbol{x}, s)\|_2 \leq (\exp(\alpha C_{\boldsymbol{\sigma}}) - 1) \cdot (\sqrt{\|\nu_1\|_\infty^2 + \|\nu_2\|_\infty^2} + 1)(C_{\boldsymbol{Z}}^1(\|\nu_1\|_\infty^2; \alpha) + 1) \cdot \mathcal{W}_2(\nu_1, \nu_2),$$

which concludes the proof. $\qquad \square$

**Lemma B.7** (Boundedness and Stability of $\boldsymbol{p}_\nu$). *Suppose that Assumption 3.3 holds and that $\boldsymbol{x}$ is in the support of $\mathcal{X}$. Suppose that $\nu_1, \nu_2 \in \mathcal{C}(\mathcal{P}^2; [0,1])$ and $\boldsymbol{p}_{\nu_1}(\boldsymbol{x}, s)$, $\boldsymbol{p}_{\nu_2}(\boldsymbol{x}, s)$ are defined in Eq. (15). Then the following three bounds are satisfied for all $s \in [0,1]$:*

$$\|\boldsymbol{p}_{\nu_1}(\boldsymbol{x}, s)\|_2 \leq C_{\boldsymbol{p}}(\|\nu_1\|_\infty^2, \|\tau\|_2^2; \alpha), \tag{43}$$

*and*

$$\|\boldsymbol{p}_{\nu_1}(\boldsymbol{x}, s) - \boldsymbol{p}_{\nu_2}(\boldsymbol{x}, s)\|_2 \leq C_{\boldsymbol{p}}(\|\nu_1\|_\infty^2, \|\nu_2\|_\infty^2, \|\tau\|_2^2; \alpha) \cdot \mathcal{W}_2(\nu_1, \nu_2), \tag{44}$$

*where $C_{\boldsymbol{p}}(\|\nu_1\|_\infty^2, \|\nu_2\|_\infty^2, \|\tau\|_2^2; \alpha)$ is a constant depending only on $\|\nu_1\|_\infty^2, \|\nu_2\|_\infty^2, \|\tau\|_2^2$ and $\alpha$, and for $\tau \in \mathcal{P}^2$, we denote by $\|\tau\|_2^2 := \mathbb{E}_{\boldsymbol{\omega} \sim \tau(\cdot)} \|\tau\|_2^2 < \infty$.*

*Proof of Lemma B.7.* For any $s \in [0,1]$, by Eq. (15) and the estimation of $\nabla_{\boldsymbol{z}}\boldsymbol{\sigma}$ in Eq. (22), we have

$$\begin{aligned}
\frac{\mathrm{d}\|\boldsymbol{p}_{\nu_1}(\boldsymbol{x}, s)\|_2^2}{\mathrm{d}s} &= 2\frac{\mathrm{d}\boldsymbol{p}_{\nu_1}^\top(\boldsymbol{x}, s)}{\mathrm{d}s}\boldsymbol{p}_{\nu_1}(\boldsymbol{x}, s) \\
&\leq 2\alpha\|\boldsymbol{p}_{\nu_1}^\top(\boldsymbol{x}, s)\|_2^2 \cdot \left\|\int_{\mathbb{R}^k} \nabla_{\boldsymbol{z}}\boldsymbol{\sigma}(\boldsymbol{Z}_{\nu_1}(\boldsymbol{x}, s), \boldsymbol{\theta})\mathrm{d}\nu_1(\boldsymbol{\theta}, s)\right\|_F \\
&\leq 2\alpha C_{\boldsymbol{\sigma}}\|\boldsymbol{p}_{\nu_1}^\top(\boldsymbol{x}, s)\|_2^2 \int_{\mathbb{R}^k}(\|\boldsymbol{\theta}\|_2^2 + 1)\mathrm{d}\nu_1(\boldsymbol{\theta}, s).
\end{aligned}$$

It follows from the estimation of $\nabla_{\boldsymbol{z}}\boldsymbol{\sigma}$ in Eq. (22),

$$\|\boldsymbol{p}_{\nu_1}^\top(\boldsymbol{x}, 1)\|_2 = \left\|\int_{\mathbb{R}^{k_\tau}} \nabla_{\boldsymbol{z}}h(\boldsymbol{Z}_{\nu_1}(\boldsymbol{x}, 1), \boldsymbol{\omega})\mathrm{d}\tau(\boldsymbol{\omega})\right\|_2 \leq C_{\boldsymbol{\sigma}} \cdot (\|\tau\|_2^2 + 1).$$

Therefore, by the Grönwall's inequality

$$\begin{aligned}
\|\boldsymbol{p}_{\nu_1}(\boldsymbol{x}, s)\|_2 &\leq C_{\boldsymbol{\sigma}} \cdot (\|\tau\|_2^2 + 1) \cdot \exp\left(\alpha C_{\boldsymbol{\sigma}}\int_0^1\int_{\mathbb{R}^k}(\|\boldsymbol{\theta}\|_2^2 + 1)\mathrm{d}\nu_1(\boldsymbol{\theta}, s)\right) \\
&\leq C(\|\nu_1\|_\infty^2, \|\tau\|_2^2; \alpha).
\end{aligned}$$

In the next, we deal with Eq. (44), define

$$\boldsymbol{\delta}_2(\boldsymbol{x}, s) := \boldsymbol{p}_{\nu_1}(\boldsymbol{x}, s) - \boldsymbol{p}_{\nu_2}(\boldsymbol{x}, s),$$

we have (taking $s = 1$)

$$\begin{aligned}
\|\boldsymbol{\delta}_2(\boldsymbol{x}, 1)\|_2 &= \left\|\int_{\mathbb{R}^{k_\tau}} \nabla_{\boldsymbol{z}}h(\boldsymbol{Z}_{\nu_1}(\boldsymbol{x}, 1), \boldsymbol{\omega}) - \nabla_{\boldsymbol{z}}h(\boldsymbol{Z}_{\nu_2}(\boldsymbol{x}, 1), \boldsymbol{\omega})\mathrm{d}\tau(\boldsymbol{\omega})\right\|_2 \\
&\leq C_{\boldsymbol{\sigma}}(\|\tau\|_2^2 + 1) \cdot \|\boldsymbol{Z}_{\nu_1}(\boldsymbol{x}, 1) - \boldsymbol{Z}_{\nu_2}(\boldsymbol{x}, 1)\|_2 \\
&\leq C(\|\nu_1\|_\infty^2, \|\nu_2\|_\infty^2, \|\tau\|_2^2; \alpha) \cdot \mathcal{W}_2(\nu_1, \nu_2).
\end{aligned}$$

The following ODE is satisfied by $\boldsymbol{\delta}_2(\boldsymbol{x}, s)$ by Eq. (15),

$$\frac{\partial\boldsymbol{\delta}_2^\top(\boldsymbol{x}, s)}{\partial s} = -\alpha \cdot \boldsymbol{\delta}_2^\top(\boldsymbol{x}, \boldsymbol{\omega}.s)\int_{\mathbb{R}^k} \nabla_{\boldsymbol{z}}\boldsymbol{\sigma}(\boldsymbol{Z}_{\nu_1}(\boldsymbol{x}, s), \boldsymbol{\theta})\mathrm{d}\nu_1(\boldsymbol{\theta}, s) + \alpha \cdot \boldsymbol{p}_{\nu_2}(\boldsymbol{x}, s)^\top\boldsymbol{D}_{\nu_1, \nu_2}(\boldsymbol{x}, s),$$

with

$$\boldsymbol{D}_{\nu_1, \nu_2}(\boldsymbol{x}, s) := \int_{\mathbb{R}^k} \nabla_{\boldsymbol{z}}\boldsymbol{\sigma}(\boldsymbol{Z}_{\nu_2}(\boldsymbol{x}, s), \boldsymbol{\theta})\mathrm{d}\nu_2(\boldsymbol{\theta}, s) - \int_{\mathbb{R}^k} \nabla_{\boldsymbol{z}}\boldsymbol{\sigma}(\boldsymbol{Z}_{\nu_1}(\boldsymbol{x}, s), \boldsymbol{\theta})\mathrm{d}\nu_1(\boldsymbol{\theta}, s).$$

Furthermore, we also split $\boldsymbol{D}_{\nu_1, \nu_2}(\boldsymbol{x}, s)$ as

$$\|\boldsymbol{D}_{\nu_1, \nu_2}(\boldsymbol{x}, s)\|_F \leq \underbrace{\left\|\int_{\mathbb{R}^k} \nabla_{\boldsymbol{z}}\boldsymbol{\sigma}(\boldsymbol{Z}_{\nu_2}(\boldsymbol{x}, s), \boldsymbol{\theta})\mathrm{d}\nu_2(\boldsymbol{\theta}, s) - \int_{\mathbb{R}^k} \nabla_{\boldsymbol{z}}\boldsymbol{\sigma}(\boldsymbol{Z}_{\nu_2}(\boldsymbol{x}, s), \boldsymbol{\theta})\mathrm{d}\nu_1(\boldsymbol{\theta}, s)\right\|_F}_{(\mathtt{A})}$$

$$+ \underbrace{\left\|\int_{\mathbb{R}^k} \left(\nabla_{\boldsymbol{z}}\boldsymbol{\sigma}(\boldsymbol{Z}_{\nu_2}(\boldsymbol{x}, s), \boldsymbol{\theta}) - \nabla_{\boldsymbol{z}}\boldsymbol{\sigma}(\boldsymbol{Z}_{\nu_1}(\boldsymbol{x}, s), \boldsymbol{\theta})\right)\mathrm{d}\nu_1(\boldsymbol{\theta}, s)\right\|_F}_{(\mathtt{B})}.$$

Clearly, (B) can be estimated by

$$(\text{B}) \leq C_{\boldsymbol{Z}}(\|\nu_1\|_\infty^2, \|\nu_2\|_\infty^2; \alpha) \cdot C_{\boldsymbol{\sigma}} \cdot (\|\nu_1\|_\infty^2 + 1) \cdot \mathcal{W}_2(\nu_1, \nu_2)\,.$$

To estimate (A), denote $\pi_\nu^\star \in \Pi(\nu_1^s, \nu_2^s)$ such that $\mathbb{E}_{(\boldsymbol{\theta}_1, \boldsymbol{\theta}_2) \sim \pi_\nu^\star} \|\boldsymbol{\theta}_1 - \boldsymbol{\theta}_2\|_2^2 = \mathcal{W}_2^2(\nu_1^s, \nu_2^s)$, by Lemma B.5, we then have

$$
\begin{aligned}
(\text{A}) &\leq \mathbb{E}_{(\boldsymbol{\theta}_1, \boldsymbol{\theta}_2) \sim \pi_\nu^\star} \|\nabla_{\boldsymbol{z}} \boldsymbol{\sigma}(\boldsymbol{Z}_{\nu_2}(\boldsymbol{x}, s), \boldsymbol{\theta}_2) - \nabla_{\boldsymbol{z}} \boldsymbol{\sigma}(\boldsymbol{Z}_{\nu_2}(\boldsymbol{x}, s), \boldsymbol{\theta}_1)\|_F \\
&\leq C_{\boldsymbol{\sigma}} \cdot \sqrt{3 \mathbb{E}_{(\boldsymbol{\theta}_1, \boldsymbol{\theta}_2) \sim \pi_\nu^\star} \|\boldsymbol{\theta}_1\|_2^2 + \|\boldsymbol{\theta}_2\|_2^2 + 1} \cdot \sqrt{\mathbb{E}_{(\boldsymbol{\theta}_1, \boldsymbol{\theta}_2) \sim \pi_\nu^\star} \|\boldsymbol{\theta}_1 - \boldsymbol{\theta}_2\|_2^2} \\
&\leq C_{\boldsymbol{\sigma}} \cdot \sqrt{3(\|\nu_1\|_\infty^2 + \|\nu_2\|_\infty^2 + 1)} \cdot \mathcal{W}_2(\nu_1, \nu_0)\,.
\end{aligned}
$$

Combining the estimate of (A) and (B), we have

$$\|\boldsymbol{D}_{\nu_1, \nu_2}(\boldsymbol{x}, s)\|_F \leq C(\|\nu_1\|_\infty^2, \|\nu_2\|_\infty^2; \alpha) \cdot \mathcal{W}_2(\nu_1, \nu_2)\,.$$

Accordingly, we are ready to estimate $\boldsymbol{\delta}_2(\boldsymbol{x}, s)$.

$$
\begin{aligned}
&\frac{\mathrm{d}\|\boldsymbol{\delta}_2(\boldsymbol{x}, s)\|_2^2}{\mathrm{d}s} \\
&= 2\left(-\alpha \cdot \delta^\top(\boldsymbol{x}, s) \int_{\mathbb{R}^k} \nabla_{\boldsymbol{z}} \boldsymbol{\sigma}(\boldsymbol{Z}_{\nu_1}(\boldsymbol{x}, s), \boldsymbol{\theta}) \mathrm{d}\nu_1(\boldsymbol{\theta}, s) + \alpha \cdot \boldsymbol{p}_{\nu_2}(\boldsymbol{x}, s)^\top \boldsymbol{D}_{\nu_1, \nu_2}(\boldsymbol{x}, s)\right) \boldsymbol{\delta}_2(\boldsymbol{x}, s) \\
&\leq 2\alpha\left(\|\boldsymbol{\delta}_2(\boldsymbol{x}, s)\|_2^2 \left(1 + \int_{\mathbb{R}^k} \|\nabla_{\boldsymbol{z}} \boldsymbol{\sigma}(\boldsymbol{Z}_{\nu_1}(\boldsymbol{x}, s), \boldsymbol{\theta})\|_F \mathrm{d}\nu_1(\boldsymbol{\theta}, s)\right) + \|\boldsymbol{p}_{\nu_2}(\boldsymbol{x}, s)^\top \boldsymbol{D}_{\nu_1, \nu_2}(\boldsymbol{x}, s)\|_2^2\right) \\
&\leq \alpha \|\boldsymbol{\delta}_2(\boldsymbol{x}, s)\|_2^2 (1 + C_{\boldsymbol{\sigma}} \cdot (1 + \|\nu_1\|_\infty^2)) + 2\alpha[C(\|\nu_1\|_\infty^2, \|\nu_2\|_\infty^2, \|\tau\|_2^2; \alpha)] \cdot \mathcal{W}_2(\nu_1, \nu_2)^2 \\
&\leq C(\|\nu_1\|_\infty^2, \|\nu_2\|_\infty^2, \|\tau\|_2^2; \alpha) \cdot (\|\boldsymbol{\delta}_2(\boldsymbol{x}, s)\|_2^2 + \mathcal{W}_2(\nu_1, \nu_2)^2)\,.
\end{aligned}
$$

By the Grönwall's inequality, $\|\boldsymbol{\delta}_2(\boldsymbol{x}, s)\|_2 \leq C_{\boldsymbol{p}}(\|\nu_1\|_\infty^2, \|\nu_2\|_\infty^2, \|\tau\|_2^2; \alpha) \cdot \mathcal{W}_2(\nu_1, \nu_2)$, and the proof is finished. $\qquad\square$

## C   MAIN RESULTS

### C.1   GRADIENT FLOW

*Proof of Theorem 4.1.* To prove Theorem 4.1, we need to estimate

$$
\begin{aligned}
\widehat{L}(\tau_t, \nu_t) - \widehat{L}(\tau_{t_0}, \nu_{t_0}) &= \frac{1}{2} \mathbb{E}_{\boldsymbol{x} \sim \mathcal{D}_n}[(\widehat{f}_{\tau_t, \nu_t}(\boldsymbol{x}) - y(\boldsymbol{x}))^2 - (\widehat{f}_{\tau_{t_0}, \nu_{t_0}}(\boldsymbol{x}) - y(\boldsymbol{x}))^2] \\
&= \mathbb{E}_{\boldsymbol{x} \sim \mathcal{D}_n}(\widehat{f}_{\tau_{t_0}, \nu_{t_0}}(\boldsymbol{x}) - y(\boldsymbol{x}))(\widehat{f}_{\tau_t, \nu_t}(\boldsymbol{x}) - \widehat{f}_{\tau_{t_0}, \nu_{t_0}}(\boldsymbol{x})) + o(|\widehat{f}_{\tau_t, \nu_t}(\boldsymbol{x}) - \widehat{f}_{\tau_{t_0}, \nu_{t_0}}(\boldsymbol{x})|),
\end{aligned}
$$

by $(a + \epsilon)^2 - a^2 = 2a\epsilon + o(|\epsilon|)$, where $o(\cdot)$ denotes the higher order of the error term. Then, we estimate $\widehat{f}_{\tau_t, \nu_t}(\boldsymbol{x}) - \widehat{f}_{\tau_{t_0}, \nu_{t_0}}(\boldsymbol{x})$,

$$
\widehat{f}_{\tau_t, \nu_t}(\boldsymbol{x}) - \widehat{f}_{\tau_{t_0}, \nu_{t_0}}(\boldsymbol{x}) = \beta \cdot \int_{\mathbb{R}^{k_\tau}} h(\boldsymbol{Z}_{\nu_t}(\boldsymbol{x}, 1), \boldsymbol{\omega}) \mathrm{d}\tau_t(\boldsymbol{\omega}) - h(\boldsymbol{Z}_{\nu_{t_0}}(\boldsymbol{x}, 1), \boldsymbol{\omega}) \mathrm{d}\tau_{t_0}(\boldsymbol{\omega})
$$

$$
= \beta \cdot \left(\underbrace{\int_{\mathbb{R}^{k_\tau}} h(\boldsymbol{Z}_{\nu_t}(\boldsymbol{x}, 1), \boldsymbol{\omega})(\mathrm{d}\tau_t(\boldsymbol{\omega}) - \mathrm{d}\tau_{t_0}(\boldsymbol{\omega}))}_{(\text{A})} + \underbrace{\int_{\mathbb{R}^{k_\tau}} (h(\boldsymbol{Z}_{\nu_t}(\boldsymbol{x}, 1), \boldsymbol{\omega}) - h(\boldsymbol{Z}_{\nu_{t_0}}(\boldsymbol{x}, 1), \boldsymbol{\omega})) \mathrm{d}\tau_{t_0}(\boldsymbol{\omega})}_{(\text{B})}\right)\,.
$$

We estimate $\boldsymbol{Z}_{\nu_t}(\boldsymbol{x}, s)$ in the following, in which we assume $\boldsymbol{\theta}_t^s \sim \nu_t(\cdot, s), \boldsymbol{\theta}_{t_0}^s \sim \nu_{t_0}(\cdot, s)$ in the expectation. Similar to the derivation in Lu et al. (2020); Ding et al. (2022), we have

$$\frac{1}{\alpha} \cdot \frac{\mathrm{d}(\boldsymbol{Z}_{\nu_t} - \boldsymbol{Z}_{\nu_{t_0}})(\boldsymbol{x}, s)}{\mathrm{d}s} = \mathbb{E}\left(\boldsymbol{\sigma}(\boldsymbol{Z}_{\nu_t}(\boldsymbol{x}, s), \boldsymbol{\theta}_t^s) - \boldsymbol{\sigma}(\boldsymbol{Z}_{\nu_{t_0}}(\boldsymbol{x}, s), \boldsymbol{\theta}_{t_0}^s)\right)$$

$$= \mathbb{E}\left(\boldsymbol{\sigma}(\boldsymbol{Z}_{\nu_t}(\boldsymbol{x}, s), \boldsymbol{\theta}_t^s) - \boldsymbol{\sigma}(\boldsymbol{Z}_{\nu_{t_0}}(\boldsymbol{x}, s), \boldsymbol{\theta}_t^s)\right) + \mathbb{E}\left(\boldsymbol{\sigma}(\boldsymbol{Z}_{\nu_{t_0}}(\boldsymbol{x}, s), \boldsymbol{\theta}_t^s) - \boldsymbol{\sigma}(\boldsymbol{Z}_{\nu_{t_0}}(\boldsymbol{x}, s), \boldsymbol{\theta}_{t_0}^s)\right)$$

$$= \mathbb{E}\, \partial_{\boldsymbol{z}}\boldsymbol{\sigma}(\boldsymbol{Z}_{\nu_{t_0}}(\boldsymbol{x}, s), \boldsymbol{\theta}_t^s)(\boldsymbol{Z}_{\nu_t}(\boldsymbol{x}, s) - \boldsymbol{Z}_{\nu_{t_0}}(\boldsymbol{x}, s)) + \mathbb{E}\, \partial_{\boldsymbol{\theta}}\boldsymbol{\sigma}(\boldsymbol{Z}_{\nu_{t_0}}(\boldsymbol{x}, s), \boldsymbol{\theta}_{t_0}^s)(\boldsymbol{\theta}_t^s - \boldsymbol{\theta}_{t_0}^s) + o(|t - t_0|)$$

$$= \mathbb{E}\, \partial_{\boldsymbol{z}}\boldsymbol{\sigma}(\boldsymbol{Z}_{\nu_{t_0}}(\boldsymbol{x}, s), \boldsymbol{\theta}_{t_0}^s)(\boldsymbol{Z}_{\nu_t}(\boldsymbol{x}, s) - \boldsymbol{Z}_{\nu_{t_0}}(\boldsymbol{x}, s)) + \mathbb{E}\, \partial_{\boldsymbol{\theta}}\boldsymbol{\sigma}(\boldsymbol{Z}_{\nu_{t_0}}(\boldsymbol{x}, s), \boldsymbol{\theta}_{t_0}^s)(\boldsymbol{\theta}_t^s - \boldsymbol{\theta}_{t_0}^s) + o(|t - t_0|)$$

$$= \mathbb{E}\, \partial_{\boldsymbol{z}}\boldsymbol{\sigma}(\boldsymbol{Z}_{\nu_{t_0}}(\boldsymbol{x}, s), \boldsymbol{\theta}_{t_0}^s)(\boldsymbol{Z}_{\nu_t}(\boldsymbol{x}, s) - \boldsymbol{Z}_{\nu_{t_0}}(\boldsymbol{x}, s))$$

$$- \mathbb{E}\, \partial_{\boldsymbol{\theta}}\boldsymbol{\sigma}(\boldsymbol{Z}_{\nu_{t_0}}(\boldsymbol{x}, s), \boldsymbol{\theta}_{t_0}^s)\nabla_{\boldsymbol{\theta}}\frac{\delta\widehat{L}(\tau_{t_0}, \nu_{t_0})}{\mathrm{d}\nu_{t_0}}(\boldsymbol{\theta}_{t_0}^s, s)(t - t_0) + o(|t - t_0|)$$

We therefore have, by the definition of $\boldsymbol{q}_\nu$ in Eq. (17),

$$(\boldsymbol{Z}_{\nu_t} - \boldsymbol{Z}_{\nu_{t_0}})(\boldsymbol{x}, 1)$$

$$= -\int_0^1 \boldsymbol{q}_{v_{t_0}}(\boldsymbol{x}, s) \cdot \mathbb{E}\left(\alpha\partial_{\boldsymbol{\theta}}\boldsymbol{\sigma}(\boldsymbol{Z}_{\nu_{t_0}}(\boldsymbol{x}, s), \boldsymbol{\theta}_{t_0}^s)\nabla_{\boldsymbol{\theta}}\frac{\delta\widehat{L}(\tau_{t_0}, \nu_{t_0})}{\mathrm{d}\nu_{t_0}}(\boldsymbol{\theta}, s)\right) \cdot (t - t_0)\mathrm{d}s + o(|t - t_0|)$$

and then we have $\|(\boldsymbol{Z}_{\nu_t} - \boldsymbol{Z}_{\nu_{t_0}})(\boldsymbol{x}, 1)\|_2 = O(|t - t_0|)$. Using this fact and by the evolution of $\tau_t$ in Eq. (10), we estimate (A),

$$(\mathtt{A}) = \int_{\mathbb{R}^{k_\tau}} h(\boldsymbol{Z}_{\nu_{t_0}}(\boldsymbol{x}, 1), \boldsymbol{\omega})(\mathrm{d}\tau_t(\boldsymbol{\omega}) - \mathrm{d}\tau_{t_0}(\boldsymbol{\omega}))$$

$$+ \int_{\mathbb{R}^{k_\tau}} (h(\boldsymbol{Z}_{\nu_t}(\boldsymbol{x}, 1), \boldsymbol{\omega}) - h(\boldsymbol{Z}_{\nu_{t_0}}(\boldsymbol{x}, 1), \boldsymbol{\omega}))(\mathrm{d}\tau_t(\boldsymbol{\omega}) - \mathrm{d}\tau_{t_0}(\boldsymbol{\omega}))$$

$$= -\int_{\mathbb{R}^{k_\tau}} h(\boldsymbol{Z}_{\nu_{t_0}}(\boldsymbol{x}, 1), \boldsymbol{\omega})\nabla_{\boldsymbol{\omega}}\frac{\delta\widehat{L}(\tau_{t_0}, \nu_{t_0})}{\delta\tau_{t_0}}(\boldsymbol{\omega})(t - t_0)\mathrm{d}\tau_{t_0}(\boldsymbol{\omega}) + o(|t - t_0|)$$

We can also estimate (B), in which $h$ is hidden in the definition of $\boldsymbol{p}_\nu$ in Eq. (17),

$$(\mathtt{B}) = \int_{\mathbb{R}^{k_\tau}} \boldsymbol{p}_{\nu_{t_0}}(\boldsymbol{x}, 1)^\top (\boldsymbol{Z}_{\nu_t} - \boldsymbol{Z}_{\nu_{t_0}})(\boldsymbol{x}, 1)\mathrm{d}\tau_{t_0}(\boldsymbol{\omega}) + o(|t - t_0|)$$

$$= -\int_{\mathbb{R}^{k_\tau}} \boldsymbol{p}_{\nu_{t_0}}(\boldsymbol{x}, s)^\top$$

$$\left(\mathbb{E}\,\alpha\nabla_{\boldsymbol{\theta}}\boldsymbol{\sigma}(\boldsymbol{Z}_{\nu_{t_0}}(\boldsymbol{x}, s), \boldsymbol{\theta}_{t_0}^s)\nabla_{\boldsymbol{\theta}}\frac{\delta\widehat{L}(\tau_{t_0}, \nu_{t_0})}{\mathrm{d}\nu_{t_0}}(\boldsymbol{\theta}_{t_0}^s, s) \cdot (t - t_0)\right)\mathrm{d}\tau_{t_0}(\boldsymbol{\omega}) + o(|t - t_0|)$$

$$= -\int_{\mathbb{R}^{k_\tau} \times \mathbb{R}^{k_\nu} \times [0,1]} \left(\boldsymbol{p}_{\nu_{t_0}}(\boldsymbol{x}, s)^\top \cdot \alpha\nabla_{\boldsymbol{\theta}}\boldsymbol{\sigma}(\boldsymbol{Z}_{\nu_{t_0}}(\boldsymbol{x}, s), \boldsymbol{\theta}) \cdot\right.$$

$$\left.\nabla_{\boldsymbol{\theta}}\frac{\delta\widehat{L}(\tau_{t_0}, \nu_{t_0})}{\delta\nu_{t_0}}(\boldsymbol{\theta}, s)\right)\mathrm{d}\nu_{t_0}(\boldsymbol{\theta}, s) \cdot (t - t_0) + o(|t - t_0|)\,.$$

Combine the estimation of (A) and (B), we have

$$\widehat{L}(\tau_t, \nu_t) - \widehat{L}(\tau_{t_0}, \nu_{t_0}) = \mathbb{E}_{\boldsymbol{x}\sim\mathcal{D}_n}\beta(\widehat{f}_{\tau_{t_0}, \nu_{t_0}}(\boldsymbol{x}) - y(\boldsymbol{x}))((\mathtt{A}) + (\mathtt{B}))$$

$$= -\mathbb{E}_{\boldsymbol{x}\sim\mathcal{D}_n}\beta(\widehat{f}_{\tau_{t_0}, \nu_{t_0}}(\boldsymbol{x}) - y(\boldsymbol{x}))\int_{\mathbb{R}^{k_\tau} \times \mathbb{R}^k \times [0,1]} \mathrm{d}\tau_{t_0}(\boldsymbol{\omega})\mathrm{d}\nu_{t_0}(\boldsymbol{\theta}, s)(t - t_0)$$

$$\cdot \left(\boldsymbol{Z}_{\nu_{t_0}}^\top(\boldsymbol{x}, 1)\nabla_{\boldsymbol{\omega}}\frac{\delta\widehat{L}(\tau_{t_0}, \nu_{t_0})}{\delta\tau_{t_0}}(\boldsymbol{\omega})\right.$$

$$+ \boldsymbol{p}_{\nu_{t_0}}^\top(\boldsymbol{x}, s) \cdot \alpha\nabla_{\boldsymbol{\theta}}\boldsymbol{\sigma}(\boldsymbol{Z}_{\nu_{t_0}}(\boldsymbol{x}, s), \boldsymbol{\theta})\nabla_{\boldsymbol{\theta}}\frac{\delta\widehat{L}(\tau_{t_0}, \nu_{t_0})}{\delta\nu_{t_0}}(\boldsymbol{\theta}, s) + o(|t - t_0|)\right)$$

$$= -\mathbb{E}_{\boldsymbol{\omega}\sim\tau_{t_0}, (\boldsymbol{\theta}, s)\sim\nu_{t_0}}\left(\left\|\nabla_{\boldsymbol{\theta}}\frac{\delta\widehat{L}(\tau, \nu)}{\delta\nu}(\boldsymbol{\theta}, s)\right\|_2^2 + \left\|\nabla_{\boldsymbol{\omega}}\frac{\delta\widehat{L}(\tau, \nu)}{\delta\tau}(\boldsymbol{\omega})\right\|_2^2\right)(t - t_0) + o(|t - t_0|)\,,$$

from the definition of functional gradient in Eq. (11) and Eq. (14). In all, the theorem is proved. $\quad\square$

*Proof of Proposition 4.2.* We expand the function derivative:

$$\int_{\mathbb{R}^{k_\tau} \times \mathbb{R}^{k_\nu} \times [0,1]} \left\| \nabla_{\boldsymbol{\theta}} \frac{\delta \widehat{L}(\tau_t, \nu_t)}{\delta \nu_t}(\boldsymbol{\theta}, s) \right\|_2^2 \mathrm{d}\tau_t(\boldsymbol{\omega}) \mathrm{d}\nu_t(\boldsymbol{\theta}, s)$$

$$= \int_{\mathbb{R}^{k_\tau} \times \mathbb{R}^{k_\nu} \times [0,1]} \frac{\beta^2 \alpha^2}{n^2} \sum_{i,j=1}^n (\widehat{f}_{\tau_t, \nu_t}(\boldsymbol{x}_i) - y(\boldsymbol{x}_i))(\widehat{f}_{\tau_t, \nu_t}(\boldsymbol{x}_j) - y(\boldsymbol{x}_j))$$

$$\cdot \boldsymbol{p}_{\nu_t}^\top(\boldsymbol{x}_i, s) \nabla_{\boldsymbol{\theta}} \boldsymbol{\sigma}(\boldsymbol{Z}_{\nu_t}(\boldsymbol{x}_i, s), \boldsymbol{\theta}) \nabla_{\boldsymbol{\theta}}^\top \boldsymbol{\sigma}(\boldsymbol{Z}_{\nu_t}(\boldsymbol{x}_j, s), \boldsymbol{\theta}) \boldsymbol{p}_{\nu_t}(\boldsymbol{x}_j, s) \mathrm{d}\tau_t(\boldsymbol{\omega}) \mathrm{d}\nu_t(\boldsymbol{\theta}, s)$$

$$= \frac{\alpha^2 \beta^2}{n^2} \boldsymbol{b}_t^\top \boldsymbol{G}_1(\tau_t, \nu_t) \boldsymbol{b}_t \,,$$

and similarly,

$$\int_{\mathbb{R}^{k_\tau}} \left\| \nabla_{\boldsymbol{\omega}} \frac{\delta \widehat{L}(\tau_t, \nu_t)}{\delta \tau_t}(\boldsymbol{\omega}) \right\|_2^2 \mathrm{d}\tau_t(\boldsymbol{\omega}) = \frac{\beta^2}{n^2} \boldsymbol{b}_t^\top \boldsymbol{G}_2(\tau_t, \nu_t) \boldsymbol{b}_t.$$

In all, the lemma is proved. $\qquad\square$

*Proof of Lemma 4.6.* We use the expansion of the gradient flow:

$$\frac{\mathrm{dKL}(\tau_t \| \tau_0)}{\mathrm{d}t} = \int_{\mathbb{R}^{k_\tau}} \frac{\delta \mathrm{KL}(\tau_t \| \tau_0)}{\delta \tau_t} \frac{\mathrm{d}\tau_t}{\mathrm{d}t} \mathrm{d}\boldsymbol{\omega} = \int_{\mathbb{R}^{k_\tau}} \frac{\delta \mathrm{KL}(\tau_t \| \tau_0)}{\delta \tau_t} \nabla \cdot \left( \tau_t(\boldsymbol{\omega}) \nabla \frac{\delta \widehat{L}(\tau_t, \nu_t)}{\delta \tau_t} \right) \mathrm{d}\boldsymbol{\omega}$$

$$= - \int_{\mathbb{R}^{k_\tau}} \tau_t(\boldsymbol{\omega}) \left( \nabla \frac{\delta \mathrm{KL}(\tau_t \| \tau_0)}{\delta \tau_t} \right) \left( \nabla \frac{\delta \widehat{L}(\tau_t, \nu_t)}{\delta \tau_t} \right) \mathrm{d}\boldsymbol{\omega}.$$

Similarly, we have

$$\frac{\mathrm{dKL}(\nu_t^s \| \nu_0^s)}{\mathrm{d}t} = \int_{\mathbb{R}^{k_\nu}} \frac{\delta \mathrm{KL}(\nu_t^s \| \nu_t^0)}{\delta \nu_t^s} \frac{\mathrm{d}\nu_t^s}{\mathrm{d}t} \mathrm{d}\boldsymbol{\theta}$$

$$= \int_{\mathbb{R}^{k_\nu}} \frac{\delta \mathrm{KL}(\nu_t^s \| \nu_t^0)}{\delta \nu_t^s} \nabla_{\boldsymbol{\theta}} \cdot \left( \nu_t^s(\boldsymbol{\theta}) \nabla_{\boldsymbol{\theta}} \frac{\delta \widehat{L}(\tau_t, \nu_t)}{\delta \nu_t}(\boldsymbol{\theta}, s) \right) \mathrm{d}\boldsymbol{\theta}$$

$$= - \int_{\mathbb{R}^{k_\nu}} \nu_t^s(\boldsymbol{\theta}) \left( \nabla_{\boldsymbol{\theta}} \frac{\delta \mathrm{KL}(\nu_t^s \| \nu_t^0)}{\delta \nu_t^s} \right) \left( \nabla_{\boldsymbol{\theta}} \frac{\delta \widehat{L}(\tau_t, \nu_t)}{\delta \nu_t}(\boldsymbol{\theta}, s) \right) \mathrm{d}\boldsymbol{\theta}.$$

Therefore, the proof is completed. $\qquad\square$

## C.2 MINIMUM EIGENVALUE AT INITIALIZATION

*Proof of Lemma 4.3.* In the proof, similar to Assumption 3.3, we assume $\boldsymbol{\theta} = (\boldsymbol{u}, \boldsymbol{w}, b) \in \mathbb{R}^{2d+1}, \boldsymbol{\omega} = (a, \boldsymbol{w}, b) \in \mathbb{R}^{d+2}, \boldsymbol{u}, \boldsymbol{w} \in \mathbb{R}^d, a, b \in \mathbb{R}$. At initialization, we notice that $\nu_0(\boldsymbol{\theta}, s) \propto \exp\left( -\frac{\|\boldsymbol{\theta}\|_2^2}{2} \right)$, and $\tau_0(\boldsymbol{\omega}) \propto \exp\left( -\frac{\|\boldsymbol{\omega}\|_2^2}{2} \right)$ are standard Gaussian. Since the distribution of $\boldsymbol{u}, a$ is symmetric, and independent from other parts of $\boldsymbol{\theta}, \boldsymbol{\omega}$ respectively, we have

$$\frac{\mathrm{d}\boldsymbol{Z}_{\nu_0}(\boldsymbol{x}, s)}{\mathrm{d}s} = \int_{\mathbb{R}^{k_\nu}} \boldsymbol{u}^\top \sigma_0(\boldsymbol{w}^\top \boldsymbol{Z}_{\nu_0}(\boldsymbol{x}, s) + b) \mathrm{d}\nu_0(\boldsymbol{u}, \boldsymbol{w}, b, s)$$

$$= \int_{\mathbb{R}^d} \boldsymbol{u}^\top \mathrm{d}\nu_0(\boldsymbol{u}) \int_{\mathbb{R}^{k_\nu - d}} \sigma_0(\boldsymbol{w}^\top \boldsymbol{Z}_{\nu_0}(\boldsymbol{x}, s) + b) \mathrm{d}\nu_0(\boldsymbol{w}, b, s) = \boldsymbol{0}, \forall s \in [0, 1],$$

$$\frac{\mathrm{d}\boldsymbol{p}_{\nu_0}^\top}{\mathrm{d}s}(\boldsymbol{x}, s) = -\alpha \cdot \boldsymbol{p}_{\nu_0}^\top(\boldsymbol{x}, s) \int_{\mathbb{R}^{k_\nu}} \nabla_{\boldsymbol{z}} \boldsymbol{\sigma}(\boldsymbol{Z}_{\nu_0}(\boldsymbol{x}, s), \boldsymbol{\theta}) \mathrm{d}\nu_0(\boldsymbol{\theta}, s)$$

$$= -\alpha \cdot \boldsymbol{p}_{\nu_0}^\top(\boldsymbol{x}, s) \int_{\mathbb{R}^{k_\nu}} \boldsymbol{u} \boldsymbol{w}^\top \sigma_0'(\boldsymbol{w}^\top \boldsymbol{Z}_{\nu_0}(\boldsymbol{x}, s) + b) \mathrm{d}\nu_0(\boldsymbol{u}, \boldsymbol{w}, b, s) = \boldsymbol{0},$$

$$\boldsymbol{p}_{\nu_0}(\boldsymbol{x}, 1) = \int_{\mathbb{R}^{k_\tau}} \nabla_{\boldsymbol{z}}^\top h(\boldsymbol{Z}_{\nu_0}(\boldsymbol{x}, 1), \boldsymbol{\omega}) \mathrm{d}\tau_0(\boldsymbol{\omega})$$

$$= \int_{\mathbb{R}^{k_\tau}} a \boldsymbol{w} \sigma_0'(\boldsymbol{w}^\top \boldsymbol{Z}_{\nu_0}(\boldsymbol{x}, 1) + b) \tau_0(a, \boldsymbol{w}, b) = \boldsymbol{0}.$$

From the first two equations, we have

$$\boldsymbol{Z}_{\nu_0}(\boldsymbol{x}, s) = \boldsymbol{x}, \forall s \quad \boldsymbol{p}_{\nu_0}(\boldsymbol{x}, s) = \boldsymbol{p}_{\nu_0}(\boldsymbol{x}, 1) = 0$$

By the definition of $\boldsymbol{G}_2(\tau_0, \nu_0)$, we have

$$
\begin{aligned}
\boldsymbol{G}_2(\tau_0, \sigma_0) &= \mathbb{E}_{(a,\boldsymbol{w},b)\sim\mathcal{N}(0,I)} \nabla_{\boldsymbol{\omega}} h(\boldsymbol{X}, \boldsymbol{\omega}) \nabla_{\boldsymbol{\omega}}^{\top} h(\boldsymbol{X}, \boldsymbol{\omega}) \\
&= \mathbb{E}_{(a,\boldsymbol{w},b)\sim\mathcal{N}(0,I)} (\boldsymbol{\sigma}_0((\boldsymbol{X}, \mathbb{1})(\boldsymbol{w}, b)), a\boldsymbol{\sigma}_0'((\boldsymbol{X}, \mathbb{1})(\boldsymbol{w}, b)), \boldsymbol{\sigma}_0'((\boldsymbol{X}, \mathbb{1})(\boldsymbol{w}, b))) \\
&\quad (\boldsymbol{\sigma}_0((\boldsymbol{X}, \mathbb{1})(\boldsymbol{w}, b)), a\boldsymbol{\sigma}_0'((\boldsymbol{X}, \mathbb{1})(\boldsymbol{w}, b)), \boldsymbol{\sigma}_0'((\boldsymbol{X}, \mathbb{1})(\boldsymbol{w}, b)))^{\top}, \\
&\geq \mathbb{E}_{(a,\boldsymbol{w},b)\sim\mathcal{N}(0,I)} \boldsymbol{\sigma}_0((\boldsymbol{X}, \mathbb{1})(\boldsymbol{w}, b)) \boldsymbol{\sigma}_0((\boldsymbol{X}, \mathbb{1})(\boldsymbol{w}, b))^{\top}.
\end{aligned}
$$

Let $\bar{\boldsymbol{x}} = (\boldsymbol{x}, 1)$, by Assumption 3.1, the cosine similarity of $\bar{\boldsymbol{x}}_i$ and $\bar{\boldsymbol{x}}_j$ is no larger than $(1 + C_{\max})/2$. Then we bound $\lambda_{\min}(\boldsymbol{G}^{(2)})$:

$$
\begin{aligned}
\lambda_{\min}(\boldsymbol{G}^{(2)}) &\geq \lambda_{\min}\left(\mathbb{E}_{\boldsymbol{w}\sim\mathcal{N}(0,\mathbb{I}_{d+1})}[\sigma_1(\bar{\boldsymbol{X}}\boldsymbol{w})\sigma_1(\bar{\boldsymbol{X}}\boldsymbol{w})^{\top}]\right) \\
&= \lambda_{\min}\left(\sum_{s=0}^{\infty} \mu_s(\sigma_1)^2 \bigcirc_{i=1}^{s}(\bar{\boldsymbol{X}}\bar{\boldsymbol{X}}^{\top})\right) \quad \text{(Nguyen \& Mondelli, 2020, Lemma D.3)} \\
&\geq \mu_r(\sigma_1)^2 \lambda_{\min}(\bigcirc_{i=1}^{r}\bar{\boldsymbol{X}}\bar{\boldsymbol{X}}^{\top}) \quad \left(\text{taking } r \geq \frac{2\log(2n)}{1 - C_{\max}}\right) \\
&\geq \mu_r(\sigma_1)^2\left(\min_{i\in[n]} \|\bar{\boldsymbol{x}}_i\|_2^{2r} - (n-1)\max_{i\neq j}|\langle\bar{\boldsymbol{x}}_i, \bar{\boldsymbol{x}}_j\rangle|^r\right) \quad \text{[Gershgorin circle theorem]} \\
&\geq \mu_r(\sigma_1)^2\left(1 - (n-1)\left(\frac{1 + C_{\max}}{2}\right)^r\right), \\
&\geq \mu_r(\sigma_1)^2\left(1 - (n-1)\left(1 - \frac{\log(2n)}{r}\right)^r\right) \\
&\geq \mu_r(\sigma_1)^2\left(1 - (n-1)\exp(-\log(2n))\right) \\
&\geq \mu_r(\sigma_1)^2/2,
\end{aligned}
$$

where the last inequality holds by the fact that $\left(1 - \frac{\log(2n)}{r}\right)^r$ is an increasing function of $r$.

$\square$

## C.3 PERTURBATION OF MINIMUM EIGENVALUE

In this section, we analyze the minimum eigenvalue of the Gram matrix.

**Lemma C.1.** *The perturbation of $\boldsymbol{G}_2(\tau, \nu)$ can be upper bounded in the following, for any $i, j \in [n]$,*

$$|\boldsymbol{G}_2(\tau, \nu) - \boldsymbol{G}_2(\tau_0, \nu_0)|_{i,j} \leq C_{\boldsymbol{G}}(\|\tau\|_2^2, \|\nu\|_{\infty}^2; d, \alpha)(\mathcal{W}_2(\tau, \tau_0) + \mathcal{W}_2(\nu, \nu_0)),$$

*where $\boldsymbol{G}_2$ is defined in Section 4.1, and $\tau_0, \nu_0$ satisfies Assumption 3.2.*

*Proof of Lemma C.1.* We deal with $\boldsymbol{G}_2(\tau, \nu)$ in an element-wise way. Let $(\boldsymbol{\omega}, \boldsymbol{\omega}_0) \sim \pi_{\tau}^{\star}$ be the optimal coupling of $\mathcal{W}_2(\tau, \tau_0)$, the difference can be estimated by

$$
\begin{aligned}
&|\boldsymbol{G}_2(\tau, \nu) - \boldsymbol{G}_2(\tau_0, \nu_0)|_{i,j} \\
\leq & \mathbb{E}|\nabla_{\boldsymbol{\omega}} h(\boldsymbol{Z}_{\nu}(\boldsymbol{x}_i, 1), \boldsymbol{\omega}) \nabla_{\boldsymbol{\omega}}^{\top} h(\boldsymbol{Z}_{\nu}(\boldsymbol{x}_j, 1), \boldsymbol{\omega}) - \nabla_{\boldsymbol{\omega}} h(\boldsymbol{Z}_{\nu_0}(\boldsymbol{x}_i, 1), \boldsymbol{\omega}_0) \nabla_{\boldsymbol{\omega}}^{\top} h(\boldsymbol{Z}_{\nu_0}(\boldsymbol{x}_j, 1), \boldsymbol{\omega}_0)| \\
\leq & \underbrace{\mathbb{E}|\nabla_{\boldsymbol{\omega}} h(\boldsymbol{Z}_{\nu}(\boldsymbol{x}_i, 1), \boldsymbol{\omega})(\nabla_{\boldsymbol{\omega}} h(\boldsymbol{Z}_{\nu}(\boldsymbol{x}_j, 1), \boldsymbol{\omega}) - \nabla_{\boldsymbol{\omega}} h(\boldsymbol{Z}_{\nu_0}(\boldsymbol{x}_j, 1), \boldsymbol{\omega}_0))^{\top}|}_{(\text{A})} \\
&+ \underbrace{\mathbb{E}|(\nabla_{\boldsymbol{\omega}} h(\boldsymbol{Z}_{\nu}(\boldsymbol{x}_i, 1), \boldsymbol{\omega}) - \nabla_{\boldsymbol{\omega}} h(\boldsymbol{Z}_{\nu_0}(\boldsymbol{x}_i, 1), \boldsymbol{\omega}_0))\nabla_{\boldsymbol{\omega}}^{\top} h(\boldsymbol{Z}_{\nu_0}(\boldsymbol{x}_j, 1), \boldsymbol{\omega}_0)|}_{(\text{B})}.
\end{aligned}
$$

We then estimate (A) and (B) separately. The term (A) involves

$$\|\nabla_{\boldsymbol{\omega}} h(\boldsymbol{Z}_\nu(\boldsymbol{x}, 1), \boldsymbol{\omega}) - \nabla_{\boldsymbol{\omega}} h(\boldsymbol{Z}_{\nu_0}(\boldsymbol{x}, 1), \boldsymbol{\omega}_0)\|_2$$
$$\leq \|\nabla_{\boldsymbol{\omega}} h(\boldsymbol{Z}_\nu(\boldsymbol{x}, 1), \boldsymbol{\omega}) - \nabla_{\boldsymbol{\omega}} h(\boldsymbol{Z}_\nu(\boldsymbol{x}, 1), \boldsymbol{\omega}_0)\|_2 + \|\nabla_{\boldsymbol{\omega}} h(\boldsymbol{Z}_\nu(\boldsymbol{x}, 1), \boldsymbol{\omega}_0) - \nabla_{\boldsymbol{\omega}} h(\boldsymbol{Z}_{\nu_0}(\boldsymbol{x}, 1), \boldsymbol{\omega}_0)\|_2$$
$$\leq C_{\boldsymbol{\sigma}} \cdot (\|\boldsymbol{Z}_\nu(\boldsymbol{x}, 1)\|_2 + 1) \cdot \|\boldsymbol{\omega} - \boldsymbol{\omega}_0\|_2 + C_{\boldsymbol{\sigma}} \cdot (\|\boldsymbol{\omega}_0\|_2^2 + 1) \cdot \|\boldsymbol{Z}_\nu(\boldsymbol{x}, 1) - \boldsymbol{Z}_{\nu_0}(\boldsymbol{x}, 1)\|_2$$
$$\leq C_{\boldsymbol{\sigma}} \cdot ((C_{\boldsymbol{Z}}(\|\nu\|_\infty^2; \alpha) + 1) \cdot \|\boldsymbol{\omega} - \boldsymbol{\omega}_0\|_2 + (\|\boldsymbol{\omega}_0\|_2^2 + 1) \cdot C_{\boldsymbol{Z}}(\|\nu\|_\infty^2, \|\nu_0\|_\infty^2; \alpha) \mathcal{W}_2(\nu, \nu_0)),$$

where we use Lemmas B.5 and B.6 in our proof. Besides, the term (B) involves

$$\|\nabla_{\boldsymbol{\omega}} h(\boldsymbol{Z}_\nu(\boldsymbol{x}, 1), \boldsymbol{\omega})\|_2 \leq C_{\boldsymbol{\sigma}} \cdot (\|\boldsymbol{Z}_\nu(\boldsymbol{x}, 1)\|_2 + 1) \cdot (\|\boldsymbol{\omega}\|_2 + 1)$$
$$\leq C_{\boldsymbol{\sigma}} \cdot (C_{\boldsymbol{Z}}(\|\nu\|_\infty^2; \alpha) + 1) \cdot (\|\boldsymbol{\omega}\|_2 + 1).$$

Therefore, by $\mathbb{E}\|\boldsymbol{\omega}_0\|_2^4 = 3k_\tau = 3(d+2)$.

$$\text{(A)} + \text{(B)} \leq C(\|\nu\|_\infty^2, \|\nu_0\|_\infty^2; \alpha) \mathbb{E}(\|\boldsymbol{\omega} - \boldsymbol{\omega}_0\|_2 + (\|\boldsymbol{\omega}_0\|_2^2 + 1) \mathcal{W}_2(\nu, \nu_0))(\|\boldsymbol{\omega}\|_2 + \|\boldsymbol{\omega}_0\|_2 + 2)$$
$$\leq C(\|\nu\|_\infty^2, \|\nu_0\|_\infty^2; \alpha) \mathbb{E}(\|\boldsymbol{\omega} - \boldsymbol{\omega}_0\|_2 + (\|\boldsymbol{\omega}_0\|_2^2 + 1) \mathcal{W}_2(\nu, \nu_0))(\|\boldsymbol{\omega}\|_2^2 + \|\boldsymbol{\omega}_0\|_2^2 + 4)$$
$$\leq C(\|\tau\|_2^2, \|\tau_0\|_2^2, \|\nu\|_\infty^2, \|\nu_0\|_\infty^2; d, \alpha)(\mathbb{E}\|\boldsymbol{\omega} - \boldsymbol{\omega}_0\|_2(\|\boldsymbol{\omega}\|_2^2 + \|\boldsymbol{\omega}_0\|_2^2 + 4) + \mathcal{W}_2(\nu, \nu_0))$$
$$\leq C(\|\tau\|_2^2, \|\tau_0\|_2^2, \|\nu\|_\infty^2, \|\nu_0\|_\infty^2; d, \alpha)[(\mathbb{E}\|\boldsymbol{\omega} - \boldsymbol{\omega}_0\|_2^2)^{\frac{1}{2}} (\mathbb{E}(\|\boldsymbol{\omega}\|_2^2 + \|\boldsymbol{\omega}_0\|_2^2 + 4)^2)^{\frac{1}{2}} + \mathcal{W}_2(\nu, \nu_0)]$$
$$\leq C(\|\tau\|_2^2, \|\tau_0\|_2^2, \|\nu\|_\infty^2, \|\nu_0\|_\infty^2; d, \alpha)(\mathcal{W}_2(\tau, \tau_0) + \mathcal{W}_2(\nu, \nu_0)),$$

since $\mathbb{E}\|\boldsymbol{\omega} - \boldsymbol{\omega}_0\|_2^2 = (\mathcal{W}_2(\tau, \tau_0))^2$, by the definition of optimal coupling. Since $\|\tau_0\|_2^2 = d + 2, \|\nu_0\|_\infty^2 = 2d + 1$, we can drop dependence of $C$ on $\|\tau_0\|_2^2, \|\nu_0\|_\infty^2$ and replace them by $d$. In all, the lemma is proved. Specifically, we could set

$$C_{\boldsymbol{G}}(\|\tau\|_2^2, \|\nu\|_\infty^2; d, \alpha)$$
$$:= 16(d+1)C_{\boldsymbol{\sigma}}^2(C_{\boldsymbol{Z}}(\|\nu\|_\infty^2; \alpha) + 1) + C_{\boldsymbol{Z}}(\|\nu\|_\infty^2, 2d + 1; \alpha))^2(\|\tau\|_2^2 + d + 1)$$

$\square$

**Lemma C.2.** *If $\nu \in \mathcal{C}(\mathcal{P}^2; [0, 1]), \tau \in \mathcal{P}^2, \mathcal{W}_2(\nu, \nu_0) \leq \sqrt{d}$, and $\mathcal{W}_2(\tau, \tau_0) \leq \sqrt{d}$, we have $\forall i, j \in [n]$,*

$$|\boldsymbol{G}_2(\tau, \nu) - \boldsymbol{G}_2(\tau_0, \nu_0)|_{i,j} \leq C_{\boldsymbol{G}}(d, \alpha) \cdot (\mathcal{W}_2(\nu_t, \nu_0) + \mathcal{W}_2(\tau_t, \tau_0))$$

*Proof of Lemma C.2.* For any $s \in [0, 1]$, let $(\boldsymbol{\theta}^s, \boldsymbol{\theta}_0^s) \sim \pi_{\nu^s}^\star$ be the optimal coupling of $\mathcal{W}_2(\nu^s, \nu_0^s)$.

$$\|\nu^s\|_2^2 = \mathbb{E}\|\boldsymbol{\theta}^s\|_2^2 \leq 2\mathbb{E}(\|\boldsymbol{\theta}^s - \boldsymbol{\theta}_0^s\|_2^2 + \|\boldsymbol{\theta}_0^s\|_2^2) = 2\mathcal{W}_2^2(\nu^s, \nu_0^s) + 2(2d + 1) \leq 6d + 2.$$

where the last inequality holds, since $\mathcal{W}_2(\nu^s, \nu_0^s) \leq \mathcal{W}_2(\nu, \nu_0), \forall s \in [0, 1]$.

We also let $(\boldsymbol{\omega}, \boldsymbol{\omega}_0) \sim \pi_\tau^\star$ be the optimal coupling of $\mathcal{W}_2(\tau, \tau_0)$.

$$\|\tau\|_2^2 = \mathbb{E}\|\boldsymbol{\omega}\|_2^2 \leq 2\mathbb{E}(\|\boldsymbol{\omega} - \boldsymbol{\omega}_0\|_2^2 + \|\boldsymbol{\omega}_0\|_2^2) = 2\mathcal{W}_2^2(\tau, \tau_0) + 2(d + 2) \leq 4(d + 1).$$

Therefore, $\|\nu\|_\infty \leq \sqrt{6d + 2}, \|\tau\|_2 \leq 2\sqrt{d + 1}$.

By Lemma C.1, replacing $\|\tau\|_2^2, \|\nu\|_\infty^2$ with their upper bound w.r.t. $d$ in the definition of $C_{\boldsymbol{G}}(\|\tau\|_2^2, \|\nu\|_\infty^2; d, \alpha)$, there exist $C_{\boldsymbol{G}}(d, \alpha)$ satisfying Lemma C.2. $\square$

For ease of description, we restate Lemma 4.4 with more details here.

**Lemma C.3.** *If $\nu \in \mathcal{C}(\mathcal{P}^2; [0, 1]), \tau \in \mathcal{P}^2, \mathcal{W}_2(\nu, \nu_0) \leq r$ and $\mathcal{W}_2(\tau, \tau_0) \leq r$, we have*

$$\lambda_{\min}(\boldsymbol{G}_2(\tau, \nu)) \geq \frac{\Lambda}{2}, \text{with } r := r_{\max}(d, \alpha) = \min\left\{\sqrt{d}, \frac{\Lambda}{4nC_{\boldsymbol{G}}(d, \alpha)}\right\}.$$

*where $\Lambda$ is defined in Lemma 4.3, and $C_{\boldsymbol{G}}(d, \alpha)$ is defined in Lemma C.2.*

*Proof of Lemma C.3.* By Lemma C.2, let

$$r := \min\left\{\sqrt{d}, \frac{\Lambda}{4nC_{\boldsymbol{G}}(d, \alpha)}\right\},$$

By Lemma 4.3, we have $\forall i, j \in [n]$,

$$|\boldsymbol{G}_2(\tau, \nu) - \boldsymbol{G}_2(\tau_0, \nu_0)|_{i,j} \leq C_{\boldsymbol{G}}(d; \alpha) \cdot (\mathcal{W}_2(\nu, \nu_0) + \mathcal{W}_2(\tau, \tau_0)) \leq \frac{\Lambda}{2n}.$$

By the standard matrix perturbation bounds, we have

$$\begin{aligned}
\lambda_{\min}(\boldsymbol{G}_2(\tau, \nu)) &\geq \lambda_{\min}(\boldsymbol{G}_2(\tau_0, \nu_0)) - \|\boldsymbol{G}_2(\tau, \nu) - \boldsymbol{G}_2(\tau_0, \nu_0)\|_2 \\
&\geq \lambda_{\min}(\boldsymbol{G}_2(\tau_0, \nu_0)) - n\|\boldsymbol{G}_2(\tau, \nu) - \boldsymbol{G}_2(\tau_0, \nu_0)\|_{\infty, \infty} \\
&\geq \frac{\Lambda}{2}.
\end{aligned}$$

$\square$

### C.4 ESTIMATION OF KL DIVERGENCE.

Inspired by Lemma C.3, we propose the following definition:

**Definition C.4.** *Define*

$$t_{\max} := \sup\{t_0, \text{s.t.} \forall t \in [0, t_0], \max\{\mathcal{W}_2(\nu_t, \nu_0), \mathcal{W}_2(\tau_t, \tau_0)\} \leq r_{\max}\},$$

*where $r_{\max}$ is defined in Lemma 4.4.*

By Lemma C.2 and Lemma C.3, for $t \leq t_{\max}$, we have $\max\{\|\nu_t\|_\infty^2, \|\tau_t\|_2^2\} = O(d)$, and $\lambda_{\min}(\boldsymbol{G}_2(\tau_t, \nu_t)) \geq \frac{\Lambda}{2}$.

We first prove the linear convergence of empirical loss under finite time.

**Lemma C.5.** *Assume the PDE* (10) *has solution $\tau_t \in \mathcal{P}^2$, and the PDE* (12) *has solution $\nu_t \in \mathcal{C}(\mathcal{P}^2; [0, 1])$. Under Assumption 3.1, 3.2, 3.3, for all $t \in [0, t_{\max})$, we have*

$$\widehat{L}(\tau_t, \nu_t) \leq e^{-\frac{\beta^2 \Lambda}{2n} t}\widehat{L}(\tau_0, \nu_0), \quad \mathrm{KL}(\tau_t\|\tau_0) \leq \frac{C_{\mathrm{KL}}(d, \alpha)}{\Lambda^2 \bar{\beta}^2}, \quad \mathrm{KL}(\nu_t\|\nu_0) \leq \frac{C_{\mathrm{KL}}(d, \alpha)}{\Lambda^2 \bar{\beta}^2}. \quad (45)$$

*Proof of Lemma C.5.* Please see Lemma C.6, Lemma C.7, and Lemma C.8. $\square$

**Lemma C.6.** *Assume $\tau_t, \nu_t$ is the solution to PDE* (10) *and* (12)*, we have for $t < t_{\max}$,*

$$\widehat{L}(\tau_t, \nu_t) \leq e^{-\frac{\beta^2 \Lambda}{2n} t}\widehat{L}(\tau_0, \nu_0),$$

*where $\Lambda$ is defined in Lemma 4.3.*

*Proof of Lemma C.6.* By Lemma C.3, for $t < t_{\max}$, $\lambda_{\min}(\boldsymbol{G}(\tau_t, \nu_t)) \geq \frac{\Lambda}{2}$,

$$\frac{\partial \widehat{L}(\nu_t, \tau_t)}{\partial t} = -\frac{\beta^2}{n^2}\boldsymbol{b}_t^\top (\alpha^2 \boldsymbol{G}_1(\tau_t, \nu_t) + \boldsymbol{G}_2(\tau_t, \nu_t))\boldsymbol{b}_t \leq -\frac{\beta^2 \Lambda}{2n}\widehat{L}(\tau_t, \nu_t) \leq -\frac{\beta^2 \Lambda}{2n}\widehat{L}(\tau_0, \nu_0).$$

Therefore, we have

$$\widehat{L}(\tau_t, \nu_t) \leq e^{-\frac{\beta^2 \Lambda}{2n} t}\widehat{L}(\tau_0, \nu_0).$$

$\square$

**Lemma C.7.** *Assume the PDE* (12) *has solution $\nu_t \in \mathcal{C}(\mathcal{P}^2; [0, 1])$, and the PDE* (10) *has solution $\tau_t \in \mathcal{P}^2$. Under Assumption 3.3, 3.1, 3.2, then for all $t \in [0, t_{\max})$, the following results hold,*

$$\mathrm{KL}(\nu_t^s\|\nu_0^s) \leq \frac{1}{\Lambda^2 \bar{\beta}^2}C_{\mathrm{KL}}(d, \alpha), \quad \forall s \in [0, 1].$$

*Proof of Lemma C.7.* By Gaussian initialization of $\nu_0^s$, $\log \nu_0^s(\boldsymbol{\theta}) = -\frac{\|\boldsymbol{\theta}\|_2^2}{2} + C$, and we thus have $\nabla_{\boldsymbol{\theta}} \frac{\partial \mathrm{KL}(\nu_t^s\|\nu_0^s)}{\partial \nu_t^s} = \nabla_{\boldsymbol{\theta}} \log \nu_t^s + \boldsymbol{\theta}$. Combining this with Lemma 4.6 and Eq. (14), we have

$$\begin{aligned}
\frac{\partial \mathrm{KL}(\nu_t^s\|\nu_0^s)}{\partial t} = &-\mathbb{E}_{\boldsymbol{x} \sim \mathcal{D}_n} \beta \cdot (f_{\tau_t, \nu_t}(\boldsymbol{x}) - y(\boldsymbol{x})) \\
&\cdot \alpha \int_{\mathbb{R}^{k_\tau}} \nabla_{\boldsymbol{\theta}}(\boldsymbol{p}_{\nu_t}^\top(\boldsymbol{x}, s)\boldsymbol{\sigma}(\boldsymbol{Z}_{\nu_t}(\boldsymbol{x}, s), \boldsymbol{\theta})) \cdot (\nabla_{\boldsymbol{\theta}} \log \nu_t^s + \boldsymbol{\theta})\mathrm{d}\nu_t^s(\boldsymbol{\theta}).
\end{aligned}$$

Define

$$J_t^s(\boldsymbol{x}, \boldsymbol{\theta}) := -\alpha \cdot \boldsymbol{p}_{\nu_t}(\boldsymbol{x}, s)^\top \Big( \nabla_{\boldsymbol{\theta}} \boldsymbol{\sigma}(\boldsymbol{Z}_{\nu_t}(\boldsymbol{x}, s), \boldsymbol{\theta}) \cdot \boldsymbol{\theta} - \Delta_{\boldsymbol{\theta}} \boldsymbol{\sigma}(\boldsymbol{Z}_{\nu_t}(\boldsymbol{x}, s), \boldsymbol{\theta}) \Big).$$

By the definition of $J_t^s(\boldsymbol{x}, \boldsymbol{\theta})$, and integration by parts, we have

$$\frac{\partial \mathrm{KL}(\nu_t^s || \nu_0^s)}{\partial t} = \beta \cdot \mathbb{E}_{\boldsymbol{x} \sim \mathcal{D}_n}[(f_{\tau_t, \nu_t}(\boldsymbol{x}) - y(\boldsymbol{x})) \mathbb{E}_{\boldsymbol{\theta} \sim \nu_t^s} J_t^s(\boldsymbol{x}, \boldsymbol{\theta})].$$

We can obtain the gradient of $J_t^s$ w.r.t. $\boldsymbol{\theta}$,

$$\nabla_{\boldsymbol{\theta}} J_t^s(\boldsymbol{x}, \boldsymbol{\theta}) = -\alpha \cdot \boldsymbol{p}_{\nu_t}(\boldsymbol{x}, s)^\top \Big( \nabla_{\boldsymbol{\theta}}(\nabla_{\boldsymbol{\theta}} \boldsymbol{\sigma}(\boldsymbol{Z}_{\nu_t}(\boldsymbol{x}, s), \boldsymbol{\theta}) \cdot \boldsymbol{\theta}) - \nabla_{\boldsymbol{\theta}} \Delta_{\boldsymbol{\theta}} \boldsymbol{\sigma}(\boldsymbol{Z}_{\nu_t}(\boldsymbol{x}, s), \boldsymbol{\theta}) \Big).$$

Therefore, by Lemma B.4, and the estimate $\|\boldsymbol{Z}_{\nu_t}(\boldsymbol{x}, s)\|_2 \le C_{\boldsymbol{Z}}(\|\nu_t\|_\infty^2; \alpha)$ from Lemma B.6, we have

$$\|\nabla(\nabla_{\boldsymbol{\theta}} \boldsymbol{\sigma}(\boldsymbol{Z}_{\nu_t}(\boldsymbol{x}, s), \boldsymbol{\theta}) \cdot \boldsymbol{\theta})\|_F \le C_{\boldsymbol{\sigma}} \cdot (\|\boldsymbol{\theta}\|_2 + 1) \cdot (C_{\boldsymbol{Z}}(\|\nu_t\|_\infty^2; \alpha) + 1)$$
$$\|\nabla_{\boldsymbol{\theta}} \Delta_{\boldsymbol{\theta}} \boldsymbol{\sigma}(\boldsymbol{Z}_{\nu_t}(\boldsymbol{x}, s), \boldsymbol{\theta})\|_F \le C_{\boldsymbol{\sigma}} \cdot (\|\boldsymbol{\theta}\|_2 + 1) \cdot (C_{\boldsymbol{Z}}(\|\nu_t\|_\infty^2; \alpha)^3 + 1).$$

Therefore, we can estimate $\nabla_{\boldsymbol{\theta}} J_t^s(\boldsymbol{x}, \boldsymbol{\theta})$,

$$\begin{aligned}
\|\nabla_{\boldsymbol{\theta}} J_t^s(\boldsymbol{x}, \boldsymbol{\theta})\|_2 &\le 2C_{\boldsymbol{\sigma}} \cdot (C_{\boldsymbol{Z}}(\|\nu_t\|_\infty^2; \alpha)^3 + 1)(\|\boldsymbol{\theta}\|_2 + 1)\|\boldsymbol{p}_{\nu_t}(\boldsymbol{x}, s)\|_2 \\
&\le 2C_{\boldsymbol{\sigma}} \cdot (C_{\boldsymbol{Z}}(\|\nu_t\|_\infty^2; \alpha)^3 + 1) \cdot C_{\boldsymbol{p}}(\|\nu_t\|_\infty^2, \|\tau_t\|_2^2; \alpha) \cdot (\|\boldsymbol{\theta}\|_2 + 1) \\
&\le C(\|\tau_t\|_2^2, \|\nu_t\|_\infty^2; \alpha) \cdot (\|\boldsymbol{\theta}\|_2 + 1).
\end{aligned}$$

By Lemma B.1 and Lemma B.2

$$\begin{aligned}
\mathbb{E}_{\nu_t^s} J_t^s(\boldsymbol{x}, \boldsymbol{\theta}) - \mathbb{E}_{\nu_0^s} J_0^s(\boldsymbol{x}, \boldsymbol{\theta}_0) &\le C(\|\nu_t\|_\infty^2, \|\tau_t\|_2^2; \alpha) \mathcal{W}_2(\nu_t^s, \nu_0^s) \\
&\le C(\|\nu_t\|_\infty^2, \|\tau_t\|_2^2; \alpha) \sqrt{\mathrm{KL}(\nu_t^s || \nu_0^s)}.
\end{aligned}$$

For $t \in [0, t_{\max})$, by Definition C.4, we have $\|\nu_t\|_\infty^2, \|\tau_t\|_2^2 = O(d)$. We have

$$\mathbb{E}_{\nu_t^s} J_t^s(\boldsymbol{x}, \boldsymbol{\theta}) - \mathbb{E}_{\nu_0^s} J_0^s(\boldsymbol{x}, \boldsymbol{\theta}_0) \le C(d, \alpha) \sqrt{\mathrm{KL}(\nu_t^s || \nu_0^s)}.$$

Since $\boldsymbol{p}_{\nu_0}(\boldsymbol{x}, s) = \boldsymbol{0}$,

$$\mathbb{E}_{\nu_0^s} J_0^s(\boldsymbol{x}, \boldsymbol{\theta}) = \boldsymbol{0}.$$

Therefore,

$$\begin{aligned}
\frac{\partial \mathrm{KL}(\nu_t^s || \nu_0^s)}{\partial t} &= \beta \cdot \mathbb{E}_{x \sim \mathcal{D}_n}(f_{\tau_t, \nu_t}(\boldsymbol{x}) - y(\boldsymbol{x})) \mathbb{E}_{\nu_t^s} J_t^s(\boldsymbol{x}, \boldsymbol{\theta}) \\
&= \beta \cdot \mathbb{E}_{x \sim \mathcal{D}_n}(f_{\tau_t, \nu_t}(\boldsymbol{x}) - y(\boldsymbol{x})) \mathbb{E}_{\nu_t^s}(J_t^s(\boldsymbol{x}, \boldsymbol{\theta}) - J_0^s(\boldsymbol{x}, \boldsymbol{\theta})) \\
&\le \beta \cdot C(d, \alpha) \sqrt{\mathrm{KL}(\nu_t^s || \nu_0^s)} \mathbb{E}_{x \sim \mathcal{D}_n}(f_{\tau_t, \nu_t}(\boldsymbol{x}) - y(\boldsymbol{x})) \\
&\le \beta \cdot C(d, \alpha) \sqrt{\mathrm{KL}(\nu_t^s || \nu_0^s)} \sqrt{\mathbb{E}(f_{\tau_t, \nu_t}(\boldsymbol{x}) - y(\boldsymbol{x}))^2} \\
&= \beta \cdot C(d, \alpha) \sqrt{\mathrm{KL}(\nu_t^s || \nu_0^s)} \sqrt{\widehat{L}(\tau_t, \nu_t)},
\end{aligned}$$

where the last inequality holds owing to the Jesen's inequality.

By the relation $\mathrm{d} 2\sqrt{x} = \mathrm{d}x / \sqrt{x}$,

$$\mathrm{d} \left( 2\sqrt{\mathrm{KL}(\nu_t^s || \nu_0^s)} \right) \le \beta \cdot C(d, \alpha) \sqrt{L(\nu_t, \tau_t)} \mathrm{d}t.$$

We have for $t \in [0, t_{\max})$,

$$\widehat{L}(\tau_t, \nu_t) \le e^{-\frac{\beta^2 \Lambda}{2n} t} \widehat{L}(\tau_0, \nu_0).$$

Hence,

$$2\sqrt{\mathrm{KL}(\nu_t^s\|\nu_0^s)} \le \beta \cdot C(d,\alpha) \int_0^t \sqrt{\widehat{L}(\nu_{t_0},\tau_{t_0})}\mathrm{d}t_0 \le \frac{4C(d,\alpha)}{\Lambda\bar{\beta}}\sqrt{\widehat{L}(\tau_0,\nu_0)}\,.$$

Since $\tau_0(\boldsymbol{u})$ is standard normal distribution, we have

$$f_{\tau_0,\nu_0}(\boldsymbol{x}) = \beta \cdot \boldsymbol{a}^\top \int_{\mathbb{R}^{k_\tau}\times\mathbb{R}^{k_\tau}\times\mathbb{R}} \boldsymbol{u}^\top \boldsymbol{\sigma}_0(\boldsymbol{w}^\top \boldsymbol{Z}_{\nu_0}(\boldsymbol{x},1)+b)\mathrm{d}\tau_0(\boldsymbol{u},\boldsymbol{w},b) = 0\,,$$

and $|y(\boldsymbol{x})| \le 1$ (Assumption 3.1), we have $\widehat{L}(\tau_0,\nu_0) \le 1$. Therefore, we obtain

$$\mathrm{KL}(\nu_t^s\|\nu_0^s) \le \frac{C_{\mathrm{KL}}(d,\alpha)}{\Lambda^2\bar{\beta}^2}\,,$$

where $C_{\mathrm{KL}}$ is a constant dependent only on $d,\alpha$. $\qquad\square$

**Lemma C.8.** *Assume the PDE* (12) *has solution* $\tau_t \in \mathcal{P}^2$, *and the PDE* (10) *has solution* $\tau_t \in \mathcal{P}^2$. *Under Assumption 3.3, 3.1, 3.2, then for all* $t \in [0,t_{\max})$, *the following results hold:*

$$\mathrm{KL}(\tau_t\|\tau_0) \le \frac{1}{\Lambda^2\bar{\beta}^2}C_{\mathrm{KL}}(d,\alpha).$$

*Proof of Lemma C.8.* By Gaussian initialization of $\tau_0$, $\log\tau_0(\boldsymbol{\omega}) = -\frac{\|\boldsymbol{\omega}\|_2^2}{2} + C$, and we have $\nabla_{\boldsymbol{\omega}}\frac{\partial\mathrm{KL}(\tau_t\|\tau_0)}{\partial\tau_t} = \nabla_{\boldsymbol{\omega}}\log\tau_t + \boldsymbol{\omega}$. Therefore, by Lemma 4.6 and Eq. (14), we have

$$\frac{\partial\mathrm{KL}(\tau_t\|\tau_0)}{\partial t}$$
$$= -\beta \cdot \int_{\mathbb{R}^{k_\tau}} (\mathbb{E}_{\boldsymbol{x}\sim\mathcal{D}_n}(f_{\tau_t,\nu_t}(\boldsymbol{x})-y(\boldsymbol{x}))\nabla_{\boldsymbol{\omega}}h(\boldsymbol{Z}_{\nu_t}(\boldsymbol{x},1),\boldsymbol{\omega}))\cdot(\nabla_{\boldsymbol{\omega}}\log\tau_t+\boldsymbol{\omega})\mathrm{d}\nu_t^s(\boldsymbol{\theta}).$$

Define

$$\tilde{\mathbf{u}}_t(\boldsymbol{\theta}) := \mathbb{E}_{\boldsymbol{x}\sim\mathcal{D}_n}[(f_{\tau_t,\nu_t}(\boldsymbol{x})-y(\boldsymbol{x}))\nabla_{\boldsymbol{\omega}}h(\boldsymbol{Z}_\nu(\boldsymbol{x},1),\boldsymbol{\omega})],$$

we have

$$\frac{\partial\mathrm{KL}(\tau_t\|\tau_0)}{\partial t} = -\alpha \cdot \int_{\mathbb{R}^{k_\tau}} \tau_t\tilde{\mathbf{u}}_t\cdot(\nabla_{\boldsymbol{\omega}}\log\tau_t+\boldsymbol{\omega})\mathrm{d}\boldsymbol{\omega} = -\alpha \cdot \int_{\mathbb{R}^{k_\tau}} \tau_t[\tilde{\mathbf{u}}_t\cdot\boldsymbol{\omega}-\nabla_{\boldsymbol{\omega}}\cdot\tilde{v}_t^s]\mathrm{d}\boldsymbol{\omega}.$$

We also define

$$I_t(\boldsymbol{x},\boldsymbol{\omega}) := -\Big(\nabla_{\boldsymbol{\omega}}h(\boldsymbol{Z}_{\nu_t}(\boldsymbol{x},1),\boldsymbol{\omega})\cdot\boldsymbol{\omega}-\Delta_{\boldsymbol{\omega}}h(\boldsymbol{Z}_{\nu_t}(\boldsymbol{x},1),\boldsymbol{\omega})\Big).$$

By the definition of $I_t(\boldsymbol{x},\boldsymbol{\omega})$,

$$\frac{\partial\mathrm{KL}(\tau_t\|\tau_0)}{\partial t} = \beta \cdot \mathbb{E}_{\boldsymbol{x}\sim\mathcal{D}_n}[(f_{\tau_t,\nu_t}(\boldsymbol{x})-y(\boldsymbol{x}))\mathbb{E}_{\boldsymbol{\omega}\sim\tau_t}I_t(\boldsymbol{x},\boldsymbol{\omega})].$$

Similar to the estimate $J_t^s$, we have the estimation of $I_t$ as

$$\mathbb{E}_{\tau_t}I_t(\boldsymbol{x},\boldsymbol{\omega}) - \mathbb{E}_{\tau_0}I_0(\boldsymbol{x},\boldsymbol{\omega}) \le C(\|\tau_t\|_2^2,\|\nu_t\|_\infty^2;\alpha)\sqrt{\mathrm{KL}(\tau_t\|\tau_0)}.$$

and we can have, by setting $\boldsymbol{\omega} = (a,\boldsymbol{w},b)$, we have

$$\mathbb{E}_{\tau_0}I_0(\boldsymbol{x},\boldsymbol{\omega}) = \mathbb{E}_{\tau_0}(-\nabla_{\boldsymbol{\omega}}h(\boldsymbol{x},\boldsymbol{\omega})\cdot\boldsymbol{\omega}+\Delta_{\boldsymbol{\omega}}h(\boldsymbol{x},\boldsymbol{\omega}))$$
$$= \mathbb{E}_{(a,\boldsymbol{w},b)\sim\mathcal{N}(0,I)}(-a\sigma_0(\boldsymbol{w}^\top\boldsymbol{x}+b)-a(\nabla_{\boldsymbol{w}}\sigma_0(\boldsymbol{w}^\top\boldsymbol{x}+b))\boldsymbol{w}-ab\sigma_0'(\boldsymbol{w}^\top\boldsymbol{x}+b))$$
$$+ \mathbb{E}_{(a,\boldsymbol{w},b)\sim\mathcal{N}(0,I)}(\Delta_{\boldsymbol{w}}\sigma_0(\boldsymbol{w}^\top\boldsymbol{x}+b)+a\sigma_0''(\boldsymbol{w}^\top\boldsymbol{x}+b)) = \mathbf{0}.$$

Therefore, we obtain the KL divergence in a similar fashion.

$$\mathrm{KL}(\tau_t\|\tau_0) \le \frac{C_{\mathrm{KL}}(d,\alpha)}{\Lambda^2\bar{\beta}^2}.$$

$\qquad\square$

**Lemma C.9** (Lower bound on the KL divergence). *For any $\tau, \tau' \in \mathcal{P}^2$, $\nu, \nu' \in \mathcal{C}(\mathcal{P}^2; [0,1])$, if $\tau', \nu'$ satisfy the Talagrand inequality $T(\frac{1}{2})$, (ref Lemma B.2) We have the lower bound for the KL divergence of $\tau, \tau'$ and $\nu, \nu'$, such that for constant $C_{\text{low}}(\|\tau\|_2^2, \|\tau'\|_2^2, \|\nu\|_\infty^2, \|\nu'\|_\infty^2; \alpha)$,*

$$\sqrt{\text{KL}(\tau\|\tau')} + \sqrt{\text{KL}(\nu\|\nu')} \geq \frac{\mathbb{E}_{\boldsymbol{\omega}\sim\tau} h(\boldsymbol{Z}_\nu(\boldsymbol{x}, 1), \boldsymbol{\omega}) - \mathbb{E}_{\boldsymbol{\omega}'\sim\tau'} h(\boldsymbol{Z}_{\nu'}(\boldsymbol{x}, 1), \boldsymbol{\omega}')}{C_{\text{low}}(\|\tau\|_2^2, \|\tau'\|_2^2, \|\nu\|_\infty^2, \|\nu'\|_\infty^2; \alpha)}.$$

.

*Proof of Lemma C.9.* We have the following estimmaation

$$\mathbb{E}_{\boldsymbol{\omega}\sim\tau} h(\boldsymbol{Z}_\nu(\boldsymbol{x}, 1), \boldsymbol{\omega}) - \mathbb{E}_{\boldsymbol{\omega}'\sim\tau'} h(\boldsymbol{Z}_{\nu'}(\boldsymbol{x}, 1), \boldsymbol{\omega}')$$
$$= \underbrace{(\mathbb{E}_{\boldsymbol{\omega}\sim\tau} h(\boldsymbol{Z}_\nu(\boldsymbol{x}, 1), \boldsymbol{\omega}) - \mathbb{E}_{\boldsymbol{\omega}'\sim\tau'} h(\boldsymbol{Z}_\nu(\boldsymbol{x}, 1), \boldsymbol{\omega}'))}_{(A)}$$
$$+ \underbrace{(\mathbb{E}_{\boldsymbol{\omega}'\sim\tau'} h(\boldsymbol{Z}_\nu(\boldsymbol{x}, 1), \boldsymbol{\omega}) - \mathbb{E}_{\boldsymbol{\omega}'\sim\tau'} h(\boldsymbol{Z}_{\nu'}(\boldsymbol{x}, 1), \boldsymbol{\omega}'))}_{(B)},$$

By Lemma B.4, we have

$$\|\nabla_{\boldsymbol{\omega}} h(\boldsymbol{Z}_\nu(\boldsymbol{x}, 1), \boldsymbol{\omega})\|_2 \leq C_{\boldsymbol{\sigma}} \cdot (\|\boldsymbol{Z}_\nu(\boldsymbol{x}, 1)\|_2 + 1) \cdot (\|\boldsymbol{\omega}\|_2 + 1),$$

and by Lemma B.1, Lemma B.6 and Lemma B.2, we have

$$(A) \leq C_{\boldsymbol{\sigma}} \cdot (\|\boldsymbol{Z}_\nu(\boldsymbol{x}, 1)\|_2 + 1) \cdot \max\{\|\tau\|_2^2, \|\tau'\|_2^2\} \cdot \mathcal{W}_2(\tau, \tau')$$
$$\leq C_{\boldsymbol{\sigma}} \cdot (C_{\boldsymbol{Z}}(\|\nu_1\|_\infty^2; \alpha) + 1) \cdot \max\{\|\tau\|_2^2, \|\tau'\|_2^2\} \mathcal{W}_2(\tau, \tau').$$
$$\leq 2C_{\boldsymbol{\sigma}} \cdot (C_{\boldsymbol{Z}}(\|\nu_1\|_\infty^2; \alpha) + 1) \cdot \max\{\|\tau\|_2^2, \|\tau'\|_2^2\} \sqrt{\text{KL}(\tau\|\tau')}.$$

Besides, by Lemma B.5 and Lemma B.2, we have

$$(B) \leq \mathbb{E}_{\boldsymbol{\omega}'\sim\tau'} C_{\boldsymbol{\sigma}}(\|\boldsymbol{\omega}\|_2^2 + 1) \cdot (\|\boldsymbol{Z}_\nu(\boldsymbol{x}, 1) - \boldsymbol{Z}_{\nu'}(\boldsymbol{x}, 1)\|_2)$$
$$\leq (\|\tau'\|_2^2 + 1) \cdot C_{\boldsymbol{Z}}(\|\nu_1\|_\infty^2, \|\nu_2\|_\infty^2; \alpha) \cdot \mathcal{W}_2(\nu_1, \nu_2)$$
$$\leq 2(\|\tau'\|_2^2 + 1) \cdot C_{\boldsymbol{Z}}(\|\nu_1\|_\infty^2, \|\nu_2\|_\infty^2; \alpha) \cdot \sqrt{\text{KL}(\nu\|\nu')}.$$

We let $C_{\text{low}}$ be

$$C_{\text{low}}(\|\tau\|_2^2, \|\tau'\|_2^2, \|\nu\|_\infty^2, \|\nu'\|_\infty^2; \alpha)$$
$$= \min\{2C_{\boldsymbol{\sigma}} \cdot (C_{\boldsymbol{Z}}(\|\nu_1\|_\infty^2; \alpha) + 1) \cdot \max\{\|\tau\|_2^2, \|\tau'\|_2^2\}, 2(\|\tau'\|_2^2 + 1) \cdot C_{\boldsymbol{Z}}(\|\nu_1\|_\infty^2, \|\nu_2\|_\infty^2; \alpha)\}$$

Therefore, we have

$$\sqrt{\text{KL}(\tau\|\tau')} + \sqrt{\text{KL}(\nu\|\nu')} \geq \frac{\mathbb{E}_{\boldsymbol{\omega}\sim\tau} h(\boldsymbol{Z}_\nu(\boldsymbol{x}, 1), \boldsymbol{\omega}) - \mathbb{E}_{\boldsymbol{\omega}'\sim\tau'} h(\boldsymbol{Z}_{\nu'}(\boldsymbol{x}, 1), \boldsymbol{\omega}')}{C_{\text{low}}(\|\tau\|_2^2, \|\tau'\|_2^2, \|\nu\|_\infty^2, \|\nu'\|_\infty^2; \alpha)}$$

Since $\|\tau\|_2^2, \|\tau'\|_2^2, \|\nu\|_\infty^2, \|\nu'\|_\infty^2 = O(d)$, we have that the average movement of the KL divergence is on the same order as the change in output value.

$\square$

**Lemma C.10.** *Assume the PDE* (12) *has solution $\tau_t \in \mathcal{P}^2$, and the PDE* (10) *has solution $\tau_t \in \mathcal{P}^2$. Under Assumption 3.3, 3.1, 3.2, then for all $t \in [0, t_{\max})$, We have the lower bound for the KL divergence of $\tau_t$ and $\nu_t$, we have for constant $C_{\text{low}}(d; \alpha)$, such that*

$$\sqrt{\text{KL}(\tau_t\|\tau_0)} + \sqrt{\text{KL}(\nu_t\|\nu_0)} \geq \frac{\mathbb{E}_{\boldsymbol{\omega}\sim\tau_t} h(\boldsymbol{Z}_{\nu_t}(\boldsymbol{x}, 1), \boldsymbol{\omega})}{C_{\text{low}}(d; \alpha)}.$$

*Proof of Lemma C.10.* By the definition of $r_{\max} \leq \sqrt{d}$, and the proof of Lemma C.2, we have $\|\tau_t\|_2^2, \|\nu_t\|_\infty^2 = O(d)$, and we can directly obtain $\|\tau_0\|_2^2 = d + 2, \|\nu_0\|_\infty^2 = 2d + 1$, and $\mathbb{E}_{\boldsymbol{\omega}_0\sim\tau_0} h(\boldsymbol{Z}_{\nu_0}(\boldsymbol{x}, 1), \boldsymbol{\omega}_0) = 0$. Besides, Gaussian initialization satisfies $T(\frac{1}{2})$ condition in Lemma B.2. We have, by Lemma C.9,

$$\sqrt{\text{KL}(\tau_t\|\tau_0)} + \sqrt{\text{KL}(\nu_t\|\nu_0)} \geq \frac{\mathbb{E}_{\boldsymbol{\omega}\sim\tau_t} h(\boldsymbol{Z}_{\nu_t}(\boldsymbol{x}, 1), \boldsymbol{\omega})}{C_{\text{low}}(d; \alpha)},$$

where $C_{\text{low}}(d; \alpha)$ is a constant depending on $d, \alpha$ derived from $C_{\text{low}}(\|\tau\|_2^2, \|\tau'\|_2^2, \|\nu\|_\infty^2, \|\nu'\|_\infty^2; \alpha)$.

$\square$

**Lemma C.11.** *Under the Assumptions in Lemma C.7, let $\bar{\beta} \geq \frac{4\sqrt{C_{\mathrm{KL}}(d,\alpha)}}{\Lambda r_{\max}}$, we have $t_{\max} = \infty$.*

*Proof of Lemma C.11.* Otherwise, we have the following inequality, for $\forall t < t_{\max}$:

$$W_2(\nu_t^s, \nu_0^s) \leq 2\sqrt{\mathrm{KL}(\nu_t^s \| \nu_t^0)} \leq \frac{2}{\Lambda\bar{\beta}}\sqrt{C_{\mathrm{KL}}(d,\alpha)}, \quad \forall s \in [0,1].$$

Therefore,

$$\mathcal{W}_2(\nu_t, \nu_0) \leq \frac{2}{\Lambda\bar{\beta}}\sqrt{C_{\mathrm{KL}}(d,\alpha)}.$$

According to the definition of $t_{\max}$, we have $\mathcal{W}_2(\nu_t, \nu_0) \leq r_{\max}$. Let

$$\bar{\beta} \geq \frac{4\sqrt{C_{\mathrm{KL}}(d,\alpha)}}{\Lambda r_{\max}}$$

we have $\mathcal{W}_2(\nu_t, \nu_0) \leq r_{\max}/2, \forall t \in [0, t_{\max})$, which contradict to the definition of $t_{\max}$ in Definition C.4. $\qquad\square$

*Proof of Theorem 4.7.* Combine the results of Lemma C.11, Lemma C.8, and Lemma C.7, we prove the theorem. $\qquad\square$

## C.5 RADEMACHER COMPLEXITY

*Proof of Lemma 4.8.* Let $\gamma$ be a parameter whose value will be determined later in the proof. Let $\eta_i, 1 \leq i \leq n$ be the i.i.d. Rademacher random variables,

$$
\begin{aligned}
\mathcal{R}_n(\mathcal{F}_{\mathrm{KL}}(r)) &= \frac{\beta}{\gamma} \cdot \mathbb{E}_\eta \left( \sup_{\tau : \mathrm{KL}(\tau\|\tau_0) \leq r, \nu : \mathrm{KL}(\nu\|\nu_0) \leq r} \mathbb{E}_\tau \left( \frac{\gamma}{n} \sum_{i=1}^n \eta_i h(\boldsymbol{Z}_\nu(\boldsymbol{x}_i, 1), \boldsymbol{\omega}) \right) \right) \\
&\leq \frac{\beta}{\gamma} \cdot \left( r + \mathbb{E}_\eta \sup_{\nu : \mathrm{KL}(\nu\|\nu_0) \leq r} \log \mathbb{E}_{\tau_0} \exp \left( \frac{\gamma}{n} \sum_{i=1}^n \eta_i h(\boldsymbol{Z}_\nu(\boldsymbol{x}_i, 1), \boldsymbol{\omega}) \right) \right) \\
&\leq \frac{\beta}{\gamma} \cdot \left( r + \mathbb{E}_\eta \log \mathbb{E}_{\tau_0} \exp \left( \frac{\gamma}{n} \sup_{\nu : \mathrm{KL}(\nu\|\nu_0) \leq r} \sum_{i=1}^n \eta_i h(\boldsymbol{Z}_\nu(\boldsymbol{x}_i, 1), \boldsymbol{\omega}) \right) \right) \\
&\leq \frac{\beta}{\gamma} \cdot \left( r + \log \mathbb{E}_{\tau_0} \mathbb{E}_\eta \exp \left( \frac{\gamma}{n} \sup_{\nu : \mathrm{KL}(\nu\|\nu_0) \leq r} \sum_{i=1}^n \eta_i h(\boldsymbol{Z}_\nu(\boldsymbol{x}_i, 1), \boldsymbol{\omega}) \right) \right),
\end{aligned}
$$

where the first inequality is followed by the Donsker-Varadhan representation of KL-divergence in Lemma B.3. The second inequality follows from the increasing function $\log(\cdot), \exp(\cdot)$, and $\sup_y \mathbb{E}_x f(x, y) \leq \mathbb{E}_x \sup_y f(x, y)$, for general variable $x, y$ and function $f$. The third inequality follows from $\log(\cdot)$'s convexity.

By Assumption 3.3, where $\boldsymbol{\omega} = (a, \boldsymbol{w}, b)$, we have

$$|h(z_1, \boldsymbol{\omega}) - h(z_2, \boldsymbol{\omega})| \leq C_1 \cdot \|z_1 - z_2\|_2 \cdot a\|\boldsymbol{w}\|_2.$$

We further estimate

$$
\begin{aligned}
&\left| \sum_{i=1}^n \eta_i h(\boldsymbol{Z}_\nu(\boldsymbol{x}_i, 1), \boldsymbol{\omega}) - \sum_{i=1}^n \eta_i h(\boldsymbol{Z}_{\nu_0}(\boldsymbol{x}_i, 1), \boldsymbol{\omega}) \right| \\
&\leq C_1 n \cdot a\|\boldsymbol{w}\|_2 \cdot \|\boldsymbol{Z}_\nu(\boldsymbol{x}_i, 1) - \boldsymbol{Z}_{\nu_0}(\boldsymbol{x}_i, 1)\|_2 \\
&\leq C_1 n \cdot a\|\boldsymbol{w}\|_2 \cdot C_{\boldsymbol{Z}}(\|\nu\|_\infty^2, \|\nu_0\|_\infty^2; \alpha) \cdot \mathcal{W}_2(\nu, \nu_0) \\
&\leq C_1 n \cdot a\|\boldsymbol{w}\|_2 \cdot C_{\boldsymbol{Z}}(\|\nu\|_\infty^2, \|\nu_0\|_\infty^2; \alpha) \cdot 2\sqrt{r}
\end{aligned}
$$

given $\mathrm{KL}(\nu\|\nu_0) \leq r$. Further, we have $\|\nu_0\|_\infty^2 = 2d + 1$, and we have $\|\nu\|_\infty^2 \leq 6d + 2$.

In the following, we use $C_d := 2C_1 \cdot C_{\mathbf{Z}}(6d + 2, 2d + 1; \alpha)$ to denote the constant.

$$\mathcal{R}_n(\mathcal{F}_{\mathrm{KL}}(r)) \le \frac{\beta}{\gamma} \cdot \left( r + \log \mathbb{E}_{\tau_0} \mathbb{E}_\eta \exp \left( \frac{\gamma}{n} \sum_{i=1}^n \eta_i h(\mathbf{Z}_{\nu_0}(\mathbf{x}_i, 1), \boldsymbol{\omega}) + \gamma \cdot C_d \sqrt{r} \cdot a \|\mathbf{w}\|_2 \right) \right)$$

$$= \frac{\beta}{\gamma} \cdot \left( r + \log \mathbb{E}_{\tau_0} \exp \left( \gamma \cdot C_d \sqrt{r} \cdot a \|\mathbf{w}\|_2 \right) \mathbb{E}_\eta \exp \left( \frac{\gamma}{n} \sum_{i=1}^n \eta_i h(\mathbf{Z}_{\nu_0}(\mathbf{x}_i, 1), \boldsymbol{\omega}) \right) \right)$$

$$\le \frac{\beta}{\gamma} \cdot \left( r + \log \mathbb{E}_{\tau_0} \exp \left( \gamma \cdot C_d \sqrt{r} \cdot a \|\mathbf{w}\|_2 + \frac{\gamma^2}{2n^2} \sum_{i=1}^n h^2(\mathbf{Z}_{\nu_0}(\mathbf{x}_i, 1), \boldsymbol{\omega}) \right) \right)$$

$$\le \frac{\beta}{\gamma} \cdot \left( r + \frac{1}{2} \log \mathbb{E}_{\tau_0} \exp \left( 2\gamma \cdot C_d \sqrt{r} \cdot a \|\mathbf{w}\|_2 \right) + \frac{1}{2} \log \mathbb{E}_{\tau_0} \exp \left( \frac{\gamma^2}{n^2} \sum_{i=1}^n h^2(\mathbf{Z}_{\nu_0}(\mathbf{x}_i, 1), \boldsymbol{\omega}) \right) \right)$$

where the first inequality follows from the previous bound; and the second inequality follows from the tail bound: $\mathbb{E}_\eta \exp(\sum_{i=1}^n \alpha_i \eta_i) \le \exp(\frac{1}{2} \sum_{i=1}^n \alpha_i^2)$; and the last inequality follows from the Cauchy ineqaulity.

Still, we set the decomposition $\boldsymbol{\omega} = (a, \mathbf{w}, b) \in \mathbb{R}^{d+2}$, we have $|h(\mathbf{Z}_{\nu_0}(\mathbf{x}_i, 1), \boldsymbol{\omega})| = |a\sigma_0(\mathbf{w}^\top \mathbf{Z}_{\nu_0}(\mathbf{x}_i, 1) + b)| \le |a|C_1$. We have

$$\mathcal{R}(\mathcal{F}_{\mathrm{KL}}(r))$$

$$\le \frac{\beta}{\gamma} \cdot \left( r + \frac{1}{2} \log \mathbb{E}_{(a,\mathbf{w},b) \sim \mathcal{N}(0,I)} \exp \left( 2\gamma \cdot C_d \sqrt{r} \cdot a \|\mathbf{w}\|_2 \right) + \frac{1}{2} \log \mathbb{E}_{a \sim \mathcal{N}(0,1)} \exp \left( \frac{\gamma^2 a^2 C_1^2}{n} \right) \right)$$

We remark that

$$\log \mathbb{E}_{t \sim \mathcal{N}(0,1)} \exp(Ct^2) = -\frac{1}{2} \log(1 - 2C) \le 2C$$

$$\log \mathbb{E}_{t \sim \mathcal{N}(0,1)} \exp(Ct) = \exp(C^2/2).$$

where the first inequality holds for $C \le 1/4$.

Therefore, setting $\gamma = \sqrt{nr}/C_1$, we have

$$\frac{1}{2} \log \mathbb{E}_{a \sim \mathcal{N}(0,1)} \exp \left( \frac{\gamma^2 a^2 C_1^2}{n} \right) = -\frac{1}{4} \log(1 - 2r) \le r,$$

$$\frac{1}{2} \log \mathbb{E}_{(a,\mathbf{w},b) \sim \mathcal{N}(0,I)} \exp \left( 2\gamma \cdot C_d \sqrt{r} \cdot a \|\mathbf{w}\|_2 \right) = \frac{1}{2} \log \mathbb{E}_{\boldsymbol{\omega} \sim \mathcal{N}(0,\boldsymbol{I})} \exp(2\gamma^2 C_d^2 r \|\mathbf{w}\|_2^2)$$

$$\le -\frac{1}{4} \log(1 - 4nr^2 (C_d/C_1)^2) \le 2nr^2 (C_d/C_1)^2.$$

where the inequality holds iff $r \le \frac{1}{4}$ and $nr^2(C_d/C_1)^2 \le \frac{1}{8}$. We set $r_0 = \min\{1/4, 1/(4\sqrt{n}) \cdot C_1/C_d\}$, we have for $\forall r \le r_0$,

$$\mathcal{R}(\mathcal{F}_{\mathrm{KL}}(r)) \le \beta/\gamma(r + 2nr^2(C_d/C_1)^2 + r) \le \beta \cdot \sqrt{r/n} \cdot 2(C_1 + C_d).$$

where $\lesssim$ hides constant. $\qquad \square$

**Theorem C.12** (Rademacher complexity). *For any $\delta > 0$, with probability at least $1 - \delta$, the following bound holds $\forall f_{\tau,\nu} \in \mathcal{F}_{\mathrm{KL}}(r)$:*

$$\mathbb{E}_{\mathbf{x} \sim \mu_X} \ell_{0-1}(f_{\tau,\nu}(\mathbf{x}), y(\mathbf{x})) \le 4\mathcal{R}_n(\mathcal{F}_{\mathrm{KL}}(r)) + 6\sqrt{\log(2/\delta)/2n} + \sqrt{\mathbb{E}_{\mathcal{D}_n}(f_{\tau,\nu}(\mathbf{x}) - y(\mathbf{x}))^2}$$

*Proof of Theorem C.12.* We introduce the additional loss function

$$\bar{\ell}(f, y) = \max\{\min\{1 - 2yf, 1\}, 0\}.$$

By definition, we have $\bar{\ell}$ is 2-Lipschitz in the first argument, and

$$\ell_{0-1}(f, y) \le \bar{\ell}(f, y) \le |f - y|,$$

for any $f \in \mathbb{R}$ and $y \in \{\pm 1\}$. Using the standard properties of Rademacher complexity, we have that with probability at least $1 - \delta$, for all $f \in \mathcal{F}_{\mathrm{KL}}(r)$

$$\mathbb{E}_{\mu_X} \bar{\ell}(f, y) \leq \mathbb{E}_{\mathcal{D}_n} \bar{\ell}(f, y) + 4\mathcal{R}_n(\mathcal{F}_{\mathrm{KL}}(r)) + 6\sqrt{\frac{\log(2/\delta)}{2n}}.$$

Therefore, we have

$$\mathbb{E}_{\mu_X} \ell_{0-1}(f, y) \leq \mathbb{E}_{\mu_X} \bar{\ell}(f, y) \leq \sqrt{\mathbb{E}_{\mathcal{D}_n}(f - y)^2} + 4\mathcal{R}_n(\mathcal{F}_{\mathrm{KL}}(r)) + 6\sqrt{\frac{\log(2/\delta)}{2n}}$$

$\square$

**Lemma C.13.** *Let $\tau_y \in \mathcal{C}(\mathcal{P}^2; [0,1])$ and $\nu_y \in \mathcal{P}^2$ be the ground truth distributions, such that,*

$$y(\boldsymbol{x}) := \mathbb{E}_{\boldsymbol{\omega} \sim \tau_y} h(\boldsymbol{Z}_{\nu_y}(\boldsymbol{x}, 1), \boldsymbol{\omega}).$$

*Then, we have the bound for the KL divergence, for $\tau_\star, \nu_\star$ satisfying $\widehat{L}(\tau_\star, \nu_\star) = 0$,*

$$\max\{\mathrm{KL}(\tau_\star\|\tau_0), \mathrm{KL}(\nu_\star\|\nu_0)\} \leq \beta^{-2}(\chi^2(\tau_y\|\tau_0) + \chi^2(\nu_y\|\nu_0)).$$

*Proof of Lemma C.13.* We assume that $\{\tau_\star^\lambda, \nu_\star^\lambda\}$ is the solution to the following minimization problem:

$$\{\tau_\star^\lambda, \nu_\star^\lambda\} = \arg\min_{\tau, \nu} \widehat{L}(\tau, \nu) + \lambda(\mathrm{KL}(\tau\|\tau_0) + \mathrm{KL}(\nu\|\nu_0)).$$

Consider the mixture distribution $\widehat{\tau}, \widehat{\nu}$ be defined as

$$(\widehat{\tau}, \widehat{\nu}) = \frac{\beta - 1}{\beta}(\tau_0, \nu_0) + \frac{1}{\beta}(\tau_y, \nu_y),$$

and we have

$$\widehat{L}(\widehat{\tau}, \widehat{\nu}) = \mathbb{E}_{x \sim \mathcal{D}_n} \left( \frac{\beta - 1}{\beta} \cdot \beta \mathbb{E}_{\boldsymbol{\omega} \sim \tau_0} h(\boldsymbol{Z}_{\nu_0}(\boldsymbol{x}, 1), \boldsymbol{\omega}) + \frac{1}{\beta} \cdot \beta \mathbb{E}_{\boldsymbol{\omega} \sim \tau_y} h(\boldsymbol{Z}_{\nu_y}(\boldsymbol{x}, 1), \boldsymbol{\omega}) - y(\boldsymbol{x}) \right)$$

$$= \mathbb{E}_{x \sim \mathcal{D}_n} \left( 0 + \frac{1}{\beta} \cdot \beta \cdot y - y \right)^2 = 0,$$

and by the definition of $\tau_\star^\lambda, \nu_\star^\lambda$, we obtain

$$\widehat{L}(\tau_\star^\lambda, \nu_\star^\lambda) + \lambda(\mathrm{KL}(\tau_\star^\lambda\|\tau_0) + \mathrm{KL}(\nu_\star^\lambda\|\nu_0)) \leq \widehat{L}(\widehat{\tau}, \widehat{\nu}) + \lambda(\mathrm{KL}(\widehat{\tau}\|\tau_0) + \mathrm{KL}(\widehat{\nu}\|\nu_0)).$$

This leads to

$$\widehat{L}(\tau_\star^\lambda, \nu_\star^\lambda) \leq \lambda(\mathrm{KL}(\widehat{\tau}\|\tau_0) + \mathrm{KL}(\widehat{\nu}\|\nu_0))$$
$$\mathrm{KL}(\tau_\star^\lambda\|\tau_0) + \mathrm{KL}(\nu_\star^\lambda\|\nu_0) \leq \mathrm{KL}(\widehat{\tau}\|\tau_0) + \mathrm{KL}(\widehat{\nu}\|\nu_0).$$

Taking $\lambda \to 0$, we have $\tau_\star^\lambda \to \tau_\star$, and $\nu_\star^\lambda \to \nu_\star$. Accordingly, we have

$$\widehat{L}(\tau_\star, \nu_\star) = 0$$
$$\mathrm{KL}(\tau_\star\|\tau_0) + \mathrm{KL}(\nu_\star\|\nu_0) \leq \mathrm{KL}(\widehat{\tau}\|\tau_0) + \mathrm{KL}(\widehat{\nu}\|\nu_0)$$

We can explicitly compute the KL divergence such that

$$\mathrm{KL}(\widehat{\tau}\|\tau_0) \leq \chi^2(\widehat{\tau}\|\tau_0) = \int \left( \frac{\beta - 1}{\beta} + \frac{\tau_y(\boldsymbol{\omega})}{\beta\tau_0(\boldsymbol{\omega})} - 1 \right)^2 \mathrm{d}\boldsymbol{\omega} = \beta^{-2}\chi^2(\tau_y\|\tau_0),$$

and similarly, we have

$$\mathrm{KL}(\widehat{\nu}\|\nu_0) \leq \beta^{-2}\chi^2(\nu_y\|\nu_0).$$

Finally, we conclude the proof. $\square$

*Proof of Theorem 4.9.* By Theorem C.12, for $r > 0$, for any $\delta > 0$, with probability at least $1 - \delta$, the following bound holds $\forall f_{\tau,\nu} \in \mathcal{F}_{\mathrm{KL}}(r)$:

$$\mathbb{E}_{\boldsymbol{x} \sim \mu_X} \ell_{0-1}(f_{\tau,\nu}(\boldsymbol{x}), y(\boldsymbol{x})) \leq 4\mathcal{R}_n(\mathcal{F}_{\mathrm{KL}}(r)) + 6\sqrt{\log(2/\delta)/2n} + \sqrt{\mathbb{E}_{\mathcal{D}_n}(f_{\tau,\nu}(\boldsymbol{x}) - y(\boldsymbol{x}))^2} \,.$$

By Lemma C.13, and the definition of $r_0$ in Lemma 4.8, we set $\beta$ such that

$$\beta^{-2}(\chi^2(\tau_y \| \tau_0) + \chi^2(\nu_y \| \nu_0)) \leq r_0 \,,$$

i.e.,

$$\beta \geq \sqrt{\frac{\chi^2(\tau_y \| \tau_0) + \chi^2(\nu_y \| \nu_0)}{r_0}} = O(\sqrt{n}) \,,$$

and

$$f_{\tau_\star,\nu_\star} \in \mathcal{F}_{\mathrm{KL}}(\beta^{-2}(\chi^2(\tau_y \| \tau_0) + \chi^2(\nu_y \| \nu_0))) \,.$$

Therefore, we apply the Rademacher complexity bound in Lemma 4.8, and we have

$$\mathcal{R}_n(\mathcal{F}_{\mathrm{KL}}(\beta^{-2}(\chi^2(\tau_y \| \tau_0) + \chi^2(\nu_y \| \nu_0)))) \lesssim \beta \sqrt{\frac{\beta^{-2}(\chi^2(\tau_y \| \tau_0) + \chi^2(\nu_y \| \nu_0))}{n}} = O(1/\sqrt{n}) \,.$$

where $\beta$ cancels out. By Lemma C.13, we have $\widehat{L}(\tau_\star, \nu_\star) = 0$. Finally,

$$\mathbb{E}_{\boldsymbol{x} \sim \mu_X} \ell_{0-1}(f_{\tau,\nu}(\boldsymbol{x}), y(\boldsymbol{x})) \lesssim O(1/\sqrt{n}) + 6\sqrt{\log(2/\delta)/2n}.$$

$\square$

## C.6 EXPERIMENTS

We validate our findings on the toy dataset "Two Spirals", where the data dimension $d = 2$. We use a neural ODE model (Poli et al., 2021) to approximate the infinite depth ResNets, where we take the discretization $L = 10$. The neural ODE model and the output layer are both parametrized by a two-layer network with the tanh activation function, and the hidden dimension is $M = K = 20$. The parameters of the ResNet encoder and the output layer are jointly trained by Adam optimizer with an initial learning rate 0.01. We perform full-batch training for 1,000 steps on the training dataset of size $n_{\mathrm{train}}$, and test the resulting model on the test dataset of size $n_{\mathrm{test}} = 1024$ by the 0-1 classification loss. We run experiments over 3 seeds and report the mean. We fit the results (after logarithm) by ordinary least squares, and obtain the slope is $-1.02$ with p-value $10^{-5}$, as shown in Figure 1. That means, the obtained rate is $\mathcal{O}(1/n)$, which is faster than our derived $\mathcal{O}(1/\sqrt{n})$ rate. We hope we could use some localized schemes, e.g., local Rademacher complexity, to close the gap.

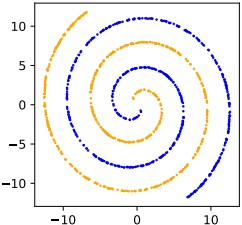 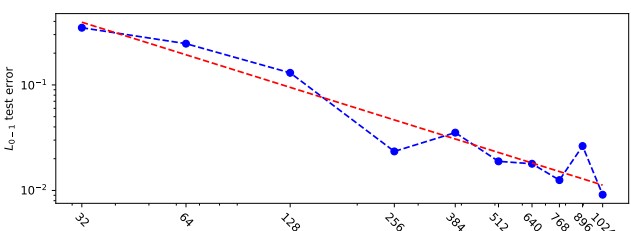

Figure 1: **Left**: "Two Spirals" datasets. **Right**: $L_{0-1}$ test error v.s. the training dataset size $n_{\mathrm{train}}$ (blue), OLS fitted line (red) which is close to the $\mathcal{O}(1/n)$ rate with p-value $10^{-5}$.

