# OpenReview forum: "Generalization of Scaled Deep ResNets in the Mean-Field Regime"
_ICLR.cc/2024/Conference — ICLR 2024 spotlight_

### Official Review · Reviewer_8ogw · 2023-10-15

**Soundness:** 3 good
**Presentation:** 3 good
**Contribution:** 2 fair
**Rating:** 5
**Confidence:** 3

**Summary:**

The manuscript studies convergence rate and generalization in infinitely deep and wide ResNets within the mean-field scaling. Utilizing this limit, they recast the neural network's dynamics as a dynamics over weight probabilities per layer. They obtain the analog of the lowest NTK eigenvalue at initialization, and show that it remains of that scale up to some range (r) in Wasserstein-2 distance. This establishes a lower bound on the convergence rate in this r-region. They then turn to bound the KL divergence in this r-region (which they can also characterize by a maximal time t_0 for which the Wasserstein-2 distance remains below r). Finally, they characterize the generalization gap using a Rademacher complexity bound where they use the fact that at init the complexity is zero and afterwards it can be bounded by the KL divergence. All in all, they claim to obtain that within the r-region the generalization gap goes as 1/\sqrt{n}.

**Strengths:**

The paper studies convergence rate and generalization in a challenging setting, a deep ResNet with no assumption on the data-set measure.

It is a technically challenging work which requires skill and effort.

**Weaknesses:**

Regarding weaknesses, I found several of those.

1. Perhaps this is a necessity for using such learning-theory tools, but I found the use of r-regions somewhat misleading. Fixing the r-region, it seems to be that, in one way or another, one is limiting the training time. Thus qualitatively, one is limiting the number of data points as training for a very short time on infinitely many data points is not much different than training for such a time on a finite large number of data points. If so what is the meaning of taking large n here if we don't give the network enough time to learn these points?

2. Similarly confusing for me was the claim that $R_n(F) = \beta \sqrt{r/n}$ when \beta scales larger than $\sqrt{n}$. If so it seems that $R_n(F)=O(n^0)$ and not $1/\sqrt{n}$.

3. The authors' results do not reflect the complexity of the task of the dataset measure. While this can also be viewed as a positive point, I find it hard to believe that one can provide tight or even order-of-magnitude estimates on actual neural networks (in the regime where they generalize well) by putting so little data in. While this is not a criticism of the specific work under consideration, it does weaken its broader appeal in my mind.

4. With relation to point number 3, some numerical experiments (to the very least on toy data sets) showing how accurate these bounds are would be useful.

**Questions:**

1. Can the authors clarify what their results mean in practical terms? Fixing hyper-parameters how do they expect r to scale t_max and n? How do they expect the training error to decrease with r?

2. Can the authors provide some supporting experiments showing how tight are their bounds?

---

> ### Author Response · Authors · 2023-11-21
> **Official Response to Reviewer 8ogw**
>
> > **Q1** What is the meaning of taking large $n$ here if we don't give the network enough time to learn these points?
>
> **A1**: In Theorem 4.8, we prove that when $\beta$ is large enough, for any $t>0$, the KL divergence is bounded by a constant order for **infinite** time. Therefore, our results have no constraint on the training time.
>
> > **Q2** $R_n(F) = \beta \sqrt{r/n}$ when $\beta$ scales larger than $\sqrt{n}$. If so it seems that $R_n(F)=O(n^0)$ and not $1/\sqrt{n}$.
>
> **A2**:  We have made the proof clearer in our updated version. Concretely, in proving the generalization bound, we take $r = O(1/\beta^2)$ in our result and thus $R_n(F) = \beta \sqrt{r/n}$ scales $1/\sqrt{n}$ as $\beta \sqrt{r}$ cancels out.
>
>
> > **Q3** The authors' results do not reflect the complexity of the task of the dataset measure.
>
> **A3**: We follow data assumptions on datasets as used in previous mean-field paper, such as "A mean field view of the landscape of two-layer neural networks". We remove the minimum eigenvalue assumption by explicitly estimating the lower bound, which is independent of $n$.
> We agree with the reviewer that a stronger data assumption would allow us for certain results under some specific settings, e.g., $k$-sparse parity problem [1].
>
> > **Q4**: Experiments:
>
> **A4**: According to your suggestion, we have already added experiments in the updated paper, see Sec. 4.5.
>
> Here we briefly introduce here. We validate our findings on the toy datasets [Two Spirals](https://i.imgur.com/FTjqjuV.jpg), where the data dimension $d=2$. We use a neural ODE model ([torchdyn](https://github.com/DiffEqML/torchdyn)) to approximate the infinite depth ResNets, we take the discretization $L=10$. The neural ODE model and the output layer are both parametrized by a two-layer network with the tanh activation function, and the hidden dimension is $M=K=20$. The parameters of ResNet encoder and the output layer are jointly trained by Adam optimizer with an initial learning rate $0.01$. We perform full-batch training for 1,000 steps on the training dataset of size $n_{\rm train}$, and test the resulting model on the test dataset of size $n_{\rm test}=1024$ by the 0-1 classification loss. We run experiments over 3 seeds and report the mean. We fit the results (after logarithm) by ordinary least squares, and obtain the slope is $-1.02$ with p-value $10^{-5}$, as shown in [this figure](https://i.imgur.com/ZJ0eAAJ.jpg).
>
> ---
>
>
> [1] Suzuki, Taiji, et al. "Feature learning via mean-field langevin dynamics: classifying sparse parities and beyond." NeurIPS 2023.

---

> ### Comment · Reviewer_8ogw · 2023-11-22
> **Response to Rebuttal**
>
> I thank the authors for their concise reply. In particular their clarification regarding the proof and the extra experiment which, despite being quite simple, illustrate the capabilities of the approach.
>
> I want to comment, that I was actually looking for **more** assumptions or "input-from" the data and not less as it is difficult to be both very general and tight.
>
> I leave my score as is.

---

> ### Author Response · Authors · 2023-11-22
> **Official Response to Reviewer 8ogw**
>
> We appreciate your time and effort in reviewing our paper. We agree with the reviewer that adopting a stronger data assumption could attain better outcomes in particular scenarios. For example, in the $k$-sparse parity problem for a special two-layer neural network [Suzuki, 1], it is possible to obtain a constant upper bound of the KL divergence to some max-margin classifier (not the training dynamics $\tau_t,\nu_t$ or the limit $\tau_\star$, $\nu_\star$ in our paper). However, this setting cannot be applied to our infinite width/depth ResNet due to the requirement of an explicit formulation of the “target” distribution in this special problem. Facing "minimal assumptions but a moderate bound", and "very strong assumptions but a tight bound" dilemma, we currently adopt the former option and hope this issue can be resolved under the latter option in the future work.

---

### Official Review · Reviewer_x2zh · 2023-10-30

**Soundness:** 3 good
**Presentation:** 4 excellent
**Contribution:** 4 excellent
**Rating:** 8
**Confidence:** 4

**Summary:**

The paper studies the optimization and generalization properties of infinitely wide and deep residual networks in the mean-field limit, where the weights are described by a probability density. The network is trained by gradient flow over the empirical risk. The paper shows two main results: the convergence to the global minimum of the empirical risk, and a generalization bound for the trained network. The convergence result is obtained by lower-bounding the smallest eigenvalue of the Gram matrix of the gradients. The generalization bound is obtained by bounding the KL divergence between the initial weight distribution and the weight distribution after training, then studying the Rademacher complexity of the family of models in a ball around initialization for the KL divergence.

**Strengths:**

The paper provides a very interesting take on the important question of jointly understanding the training dynamics and generalization properties of deep neural networks. The final result is strong since it shows a bound on the population risk of trained deep networks under reasonably mild assumptions. The paper seems technically solid, although I have not read the proofs in full details.

Furthermore, the paper is easy to read and well-presented. The paper presents all the results in a unified setting, which is enjoyable. I also liked that the three main contributions are well-separated throughout the paper (in the explanations, the mathematical statements and the proofs). This allows the community to easily build on proof techniques presented in the paper.

To my knowledge, the idea of bounding the Rademacher complexity through a bound on the KL divergence wrt the initialization is novel.

All in all, this makes this paper a very interesting contribution for the community.

**Weaknesses:**

I have two main questions for the authors. [Update on Nov 22: the authors have answered to all the questions, except the first one, see thread at the top of the page].
1. The proof of global convergence relies on lower-bounding the smallest eigenvalue of the Gram matrix of the gradients of the two-layer neural network in the end of the architecture. There is no guarantee on the Gram matrix of the gradients of the infinite-depth ResNet part of the architecture. **For this reason, I am wondering if the proof would still hold if the ResNet part of the architecture was fixed to its initial value or even altogether removed.** If this is the case, I think this should be stated more clearly, and in my opinion questions a bit the narrative of the paper which gives an important focus on the ResNet part of the architecture, whereas I do not fully understand at this point the role it plays in the proof.
2. The generalization bound is proven in the case where the radius $r$ of the KL ball decreases in $O(1/n)$. The paper proves that, with the appropriate scaling of $\beta$, the trained network remains in this ball, and thus the generalization bound holds for the trained network. This means that the distribution of the weights is required to be increasingly close to the distribution at initialization as the sample size increases. **For this reason, I am wondering if the regime where the generalization bound holds can really be though of as feature learning.** While this is not a problem in itself, it questions in my opinion the statement in the conclusion of the paper that “this is the first paper to build a generalization bound for deep neural networks in the so-called feature learning regime”. In my opinion, either more caution in the phrasing or more explanations would be highly welcome.

I would be willing to raise my score if the remarks are answered by the authors, and the narrative of the paper changed accordingly if appropriate.

A less important remark regards Theorem 4.11. The rate of convergence of the empirical risk in Lemma 4.7 actually depends on $n$ through $\beta$ (which appears both in the exponential and implicitly in the loss at initialization, since the output of the network at initialization is proportional to $\beta$). I don’t think this dependence is taken into account by the authors in Theorem 4.11, or at least it is not stated since the authors state that the empirical risk decays as $O(e^{-t})$. While I believe that the stated theorem is correct, the result could probably be sharpened (since the rate of convergence is actually $O(e^{-nt})$, although the magnitude of the loss at initialization also depends on $n$ and would have to be taken into account). In any case, a proof would be beneficial.

**Literature review**

There are several papers which show global convergence of deep residual networks and which are not cited. In particular, the beginning of Section 4 suggests that this paper is the first one to prove a rate of convergence with mild assumptions. There have actually been other papers proposing this, see e.g. https://arxiv.org/abs/1910.02934, https://arxiv.org/abs/2204.07261, https://arxiv.org/abs/2309.01213. Please rephrase accordingly the beginning of Section 4. Also, I believe Barboni et al. do not “assume that the model parameters are close to the initialization”, they prove it to be the case in their setting, which is a very different statement.

**Minor remarks that did not influence the score**

1. page 2, second paragraph: $\tau$ et $\nu$ should be swapped.
2. First paragraph of Section 2.1 is not very clear.
3. page 3: “nerual” -> neural.
4. page 3: in your ResNet model, you assume a scaling in $1/L$. While this is necessary to obtain an ODE limit and usually assumed in the corresponding literature, it departs from the actual practice. I think it would be interesting to mention this. See https://arxiv.org/abs/2105.12245, https://arxiv.org/abs/2206.06929 for a related discussion.
5. page 4: the sentence “We introduce a trainable MLP …” is not very clear. What do you mean by “zero initialization of the network’s prediction”?
6. page 5, equations (11), (14), (15): $D$ should be $D_n$.
7. pages 8 and 9: who is $\Lambda$? Should it be $\Lambda(d)$?
8. page 9, after Theorem 4.10: $r_0$ is defined but was already defined in the Theorem.
9. There are duplicated entries in the references, eg Chizat and Bach, and (probably) Ding, Chen, Li and Wright. Please double-check this.
10. Lemma B.2: $N$ should be $n$? This typo is present in several places throughout the proof.
11. Lemma B.7: I think the infinite norm is undefined. It is possible to infer what it means from context, but please define it.

**Questions:**

In your model, the output $y$ is a deterministic function of the input $x$. Could this assumption be relaxed (eg, adding noise to $y$)?

Also see weaknesses.

---

> ### Author Response · Authors · 2023-11-21
> **Official Response to Reviewer x2zh (Part I)**
>
> Thank you for reviewing our paper and providing insightful comments. We address the concerns below:
>
> > **Q1** The role of ResNet and the Gram matrix $G_1$ and $G_2$ in the analysis
>
> **A1:** Please see the common response.
>
> > **Q2** The phrase "feature learning" might be an overclaim.
>
> **A2:** We agree with the reviewer that the stationary distribution after training will be close to its initialization as suggested by our theory.
> Our results provide an upper bound of the KL divergence, which allows for a decent movement in terms of distribution, but we cannot provide a lower bound to the KL divergence to ensure a constant order distance.
> According to the reviewer's suggestion, we have rephrased the expression to "beyond NTK regime" to avoid any confusion.
>
> >**Q3** Convergence of the empirical loss.
>
> **A3:** Thanks for pointing it out. We have already updated Theorem 4.10, such that it is able to describe the limiting distribution more accurately.
>
> > **Q4** The beginning of Section 4 suggests that this paper is the first one to prove a rate of convergence with mild assumptions, and missing related works.
>
> **A4:** Thanks for pointing out these related works. We have already rephrased our expression for a better understanding in the revised version. We summarize the difference here.
>
> [1] obtains an algorithm-dependent bound for the overparameterized deep residual networks. [2] proves linear convergence of gradient descent to a global optimum for the training of deep residual networks with constant layer width.
> Both of these two works focus on the finite width setting, instead of the infinite mean-field limit in our work. The employed techniques are accordingly different.
> [3] discusses the implicit regularization of ResNets converging to ODEs by virtue of the PL condition that is different from our KL divergence technique.
> Besides, the generalization properties are not characterized in [3].
> We are thankful to the reviewer for pointing out these references. Notice that [3] was posted on arXiv in September 2023.
>
> Regarding the work from "Barboni et al." [4], we agree with the reviewer that they do not assume that the model parameters are close to the initialization. We also do not assume this in our work.
> In Theorem 2 of [4], the condition of radius $R$  is relatively difficult to verify, since the order of $\lambda(\cdot)$ function in the $N$-universality assumption (Assumption 2) is hard to estimate.
>
> ---
>
> [1] Frei, Spencer, Yuan Cao, and Quanquan Gu. "Algorithm-dependent generalization bounds for overparameterized deep residual networks." Advances in neural information processing systems 32 (2019).
>
> [2] Cont, Rama, Alain Rossier, and RenYuan Xu. "Convergence and Implicit Regularization Properties of Gradient Descent for Deep Residual Networks." arXiv preprint arXiv:2204.07261 (2022).
>
> [3] Marion, Pierre, et al. "Implicit regularization of deep residual networks towards neural ODEs." arXiv preprint arXiv:2309.01213 (2023).
>
> [4] Barboni et al. "On global convergence of ResNets: From finite to infinite width using linear parameterization." NeurIPS 2022.

---

> ### Author Response · Authors · 2023-11-21
> **Official Response to Reviewer x2zh (Part II)**
>
> > **Q5**  A scaling in $1/L$ is necessary to obtain an ODE limit, but it departs from practice.
>
> **A5**. Thanks for pointing it out. We have added these references to the paper. [1,2] discuss the influence of scaling on the ResNets training. [1] points out the $1/L$ scaling is inconsistent with the practical observation, which should be $1/L^{0.7}$ instead, which will lead to different infinite depth limit such as rough path. We leave this as future work. [2] derive a limiting neural SDE for the $1/\sqrt{L}$ scaling, and discuss the stability around the critical scaling $1/\sqrt{L}$ and $1/L$.  Different scaling leads to different continuous limits varying from Ito process to rough differential equation. We believe that our approach not only serves effectively in the Neural ODE regime but also plays a crucial role in establishing generalization bounds in other continuous limits, for our proof lies in the stability exhibited by the continuous model under consideration. We thank the reviewer for pointing this out, we'll leave this for future work.
>
> In this paper, we still adopt the $1/L$ scaling, following the traditional neural ODE community. One reason is that [1] is not the complete story, the $1/L$ scaling is also observed in practice, such as Section 5.3 in [3], which observes the approximately $1/L$ scaling in the residual connected networks (ReZero); and Section 5 in [4] also validates the $1/L$ scaling both in theory and experiment.
>
> [1] Cohen, Alain-Sam, et al. "Scaling properties of deep residual networks." International Conference on Machine Learning. PMLR, 2021.
>
> [2] Marion, Pierre, et al. "Scaling resnets in the large-depth regime." arXiv preprint arXiv:2206.06929 (2022).
>
> [3] Bachlechner, Thomas, et al. "Rezero is all you need: Fast convergence at large depth." Uncertainty in Artificial Intelligence. PMLR, 2021.
>
> [4] Marion, Pierre, et al. "Implicit regularization of deep residual networks towards neural ODEs." arXiv preprint arXiv:2309.01213 (2023).
>
> ---
>
>
> > **Q6**: Typos
>
> **A6** We are thankful to the reviewer for the attentive study of our work. We have fixed the typos that the reviewer pointed out. Specifically,
> - (1) The sentence, “zero initialization of the network’s prediction”, means that the initial prediction of the limiting network is zero, which will be useful in the later proofs. We notice that it is confusing here, and we delete this sentence.
> - (2 )$\Lambda$ is $\Lambda(d)$, we omit the dependence on $d$ for simplicity.
> - (3) We have changed Chizat and Bach's duplicated entries. For Ding, et. al, there are actually two different papers: "*On the Global Convergence of Gradient Descent for multi-layer ResNets in the mean-field Regime*" and "*Overparameterization of Deep ResNet: Zero Loss and Mean-field Analysis*". Therefore, there is no duplication here.
> - (4) We have added the definition of the distribution’s infinite norm in Lemma B.7 and B.8.

---

> > ### Comment · Reviewer_x2zh · 2023-11-22
> >
> > Thank you for your detailed answer. All the questions are answered appropriately, except the question on the smallest eigenvalue of the Gram matrix $G_1$ corresponding to the ResNet part, where I am still not convinced. I am replying to this part under the main comment at the top of the page.
> >
> > Despite this reservation, I still think this is a very interesting paper for the community, and I have updated my score accordingly.

---

> ### Author Response · Authors · 2023-11-22
> **Official Response to Reviewer x2zh**
>
> Thanks for your support on this work on increasing the score. We agree with the reviewer that our model degenerates to the two-layer NN setting if we set the residual part as an identity mapping. According to the reviewer's suggestion, we clearly mentioned this in the remark after Lemma 4.3  in our updated paper.

---

### Official Review · Reviewer_JCPH · 2023-10-31

**Soundness:** 3 good
**Presentation:** 4 excellent
**Contribution:** 3 good
**Rating:** 8
**Confidence:** 4

**Summary:**

The authors study the convergence of the gradient flow for infinitely wide and deep ResNets under a mean-field regime utilizing the technique developed by [Lu et al. (2020)] and [Ding et al. (2021, 2022)]. Specifically, they show the global convergence based on the estimation of the smallest eigenvalue of (varying) NTK matrix. Moreover, the generalization analysis using Rademacher complexity for the trained network is also provided.

**Strengths:**

- This work studies the mean-field regime of the neural network, which has attracted attention due to the presence of feature learning. The optimization analysis with the convergence rate is quite challenging especially when not using the entropy regularization. This work nicely combines the mean-field analysis of infinitely deep and wide ResNet ([Lu et al. (2020)], [Ding et al. (2021, 2022)]) and the estimation of the smallest eigenvalue of NTK matrix, and derives the convergence rate. I think this work is interesting and suggests a new way of analyzing the deep ResNets under a mean-field regime.

- The paper is well organized. The presentation is very clear and easy to follow.

**Weaknesses:**

- In fact, the mathematical method developed is quite interesting. However, I am not completely convinced whether the optimization dynamics considered in this paper indicates feature learning. The convergence analysis is based on the positivity of $G_2$ instead of $G_1$. Here $G_2$ is the Gram matrix corresponding to the last layer. Therefore, I suspect that the convergence property is fully relying on the optimization of the last layer. That is, it may not be possible to distinguish between the proposed dynamics and the optimization of the last layer only. Essentially, feature learning is caused by training the input layer rather than the output.

- The theory of the paper does not seem to work when the last layer is frozen in contrast to many work for two-layer mean-field neural networks. This concern is related to the one above.

- Missing reference. The following work is also closely relevant to the paper.

Z. Chen, E. Vanden-Eijnden, and J. Bruna. On feature learning in neural networks with global convergence guarantees.

**Questions:**

It would be nice if the authors could prove that feature learning certainly appears in the proposed setup.

---

> ### Author Response · Authors · 2023-11-21
> **Official Response to Reviewer JCPH**
>
> Thank you for reviewing our paper and providing insightful comments. We address the concerns below:
>
> > **Q1** The role of ResNet and the Gram matrix $G_1$ and $G_2$ in the analysis
>
> **A1** Please see the common response. **A1**.
>
> > **Q2** The phrase "feature learning" might be an overclaim.
>
> **A2** We agree with the reviewer that the stationary distribution after training will be close to its initialization as suggested by our theory.
> Our results provide an upper bound of the KL divergence, which allows for a decent movement in terms of distribution, We also provide a lower bound in [Lemma C.9](https://i.imgur.com/02lhb0X.png) that the average movement of the KL divergence is in the same order as the change in output layers. We notice that the upper and lower bound both scales $O(1/\beta^2)$, and therefore is tight. However, in the generalization analysis where $\beta$ is dependent on $n$, such a tight bound is therefore also dependent on $n$. According to the reviewer's suggestion, we have rephrased the expression to "beyond NTK regime" to avoid any confusion.
>
> >**Q3** Convergence of the empirical loss.
>
> **A3** Thanks for pointing it out. We have already updated Theorem 4.10, such that it is able to describe the limiting distribution more accurately.
>
> > **Q4** The beginning of Section 4 suggests that this paper is the first one to prove a rate of convergence with mild assumptions, and missing related works.
>
> **A4**. Thanks for pointing out these related works. We have already rephrased our expression for a better understanding in the revised version.
>
> We summarize the difference here. [1] obtains an algorithm-dependent bound for the overparameterized deep residual networks. [2] proves linear convergence of gradient descent to a global optimum for the training of deep residual networks with constant layer width.
> Both of these two works focus on the finite width setting, instead of the infinite mean-field limit in our work. The employed techniques are accordingly different.
> [3] discusses the implicit regularization of ResNets converging to ODEs by virtue of the PL condition that is different from our KL divergence technique.
> Besides, the generalization properties are not characterized in [3].
> We are thankful to the reviewer for pointing out these references. Notice that [3] was posted on arXiv in September 2023.
>
> Regarding the work from "Barboni et al." [4], we agree with the reviewer that they do not assume that the model parameters are close to the initialization. We also do not assume this in our work.
> In Theorem 2 of [4], the condition of radius $R$  is relatively difficult to verify, since the order of $\lambda(\cdot)$ function in the $N$-universality assumption (Assumption 2) is hard to estimate.
>
> [1] Frei, Spencer, Yuan Cao, and Quanquan Gu. "Algorithm-dependent generalization bounds for overparameterized deep residual networks." Advances in neural information processing systems 32 (2019).
>
> [2] Cont, Rama, Alain Rossier, and RenYuan Xu. "Convergence and Implicit Regularization Properties of Gradient Descent for Deep Residual Networks." arXiv preprint arXiv:2204.07261 (2022).
>
> [3] Marion, Pierre, et al. "Implicit regularization of deep residual networks towards neural ODEs." arXiv preprint arXiv:2309.01213 (2023).
>
> [4] Barboni et al. "On global convergence of ResNets: From finite to infinite width using linear parameterization." NeurIPS 2022.v

---

### Author Response · Authors · 2023-11-21
**Common Response**

We are thankful to the reviewers for their effort and time to review this paper. We address below one concern that was common among reviewers and then we address the concerns of each reviewer individually.

> **Common Questions** The role of ResNet and the Gram matrix $G_1$ and $G_2$ in the analysis

**A**: We thank the reviewers for pointing out the confusing point here.
Using the stability of the ODE model, we derive the KL divergence by virtue of the structure of the ResNet, build the lower bound of $\lambda_{\min}(G_2)$, and prove the global convergence.
In fact, our results, e.g., global convergence, and KL divergence can also depend on $G_1$ by taking $\Lambda(t) := \alpha^2 \lambda_{\min}[G_1(t)] + \lambda_{\min}(G_2)$ in Lemma 4.7.
Due to $\lambda_{\min}[ G_1(t)] \geq 0$ for any $t$, we only use $\lambda_{\min}(G_2) \geq \Lambda(d)$ in Lemma 4.3 for proof simplicity.
We have added a remark in Lemma 4.7 for clarification.

---

> ### Comment · Reviewer_x2zh · 2023-11-22
>
> Thanks to the authors for their answer to a common question of the reviewers. I agree with the authors that the KL divergence is bounded both for the residual part and the last fully connected part. However, the Gram matrix lower bound is only proven for the last fully connected part (since the lower bound by $0$ for $G_1$ is trivial since a Gram matrix is always positive semidefinite). Thus, I think that the global convergence result also holds if the residual part is fixed to its initial value (which is an identity mapping in the infinite width limit). If this remark is correct, I would strongly encourage the authors to add it in the paper for clarity.

---

### Meta-Review · Area_Chair_GhVc · 2023-12-14

**Metareview:**

This paper analyzes optimization and generalization properties of ResNet in a mean field regime where the both width and depth are taken limit to infinity. Thanks to the minimum eigenvalue evaluation of the Gram matrix, the convergence of the gradient descent is guaranteed. In addition to that, the Rademacher complexity bound is also given by utilizing the KL-divergence bound. Some numerical experiments are also given.

The analyses presented in this paper are interesting and solid. The convergence of GD in mean field ResNet and its connection to generalization error bound are novel. One concern is that it is not entirely clear that the analysis in this paper really characterize feature learning. However, in summary, this paper is well written and novel result is shown. Hence, I recommend acceptance for this paper.

**Justification For Why Not Higher Score:**

Although this paper gives an interesting result, its applicability is still restrictive. For example, the analysis in the paper does not show the benefit of feature learning, and it is still not clear whether depth helps its generalization.

**Justification For Why Not Lower Score:**

The result is certain and the technical novelty of this paper is actually good. In particular, the global optimality of GD in the deep ResNet is an interesting result.

---

### Decision · Program_Chairs · 2024-01-16

Accept (spotlight)